# Second-Order Neural ODE Optimizer

**Guan-Horng Liu, Tianrong Chen, Evangelos A. Theodorou**
Georgia Institute of Technology, USA
{ghliu, tianrong.chen, evangelos.theodorou}@gatech.edu

## Abstract

We propose a novel second-order optimization framework for training the emerging deep continuous-time models, specifically the Neural Ordinary Differential Equations (Neural ODEs). Since their training already involves expensive gradient computation by solving a backward ODE, deriving efficient second-order methods becomes highly nontrivial. Nevertheless, inspired by the recent Optimal Control (OC) interpretation of training deep networks, we show that a specific continuous-time OC methodology, called *Differential Programming*, can be adopted to derive backward ODEs for higher-order derivatives at the same $\mathcal{O}(1)$ memory cost. We further explore a low-rank representation of the second-order derivatives and show that it leads to efficient preconditioned updates with the aid of Kronecker-based factorization. The resulting method – named **SNOpt** – converges much faster than first-order baselines in wall-clock time, and the improvement remains consistent across various applications, *e.g.* image classification, generative flow, and time-series prediction. Our framework also enables direct architecture optimization, such as the integration time of Neural ODEs, with second-order feedback policies, strengthening the OC perspective as a principled tool of analyzing optimization in deep learning. Our code is available at https://github.com/ghliu/snopt.

## 1 Introduction

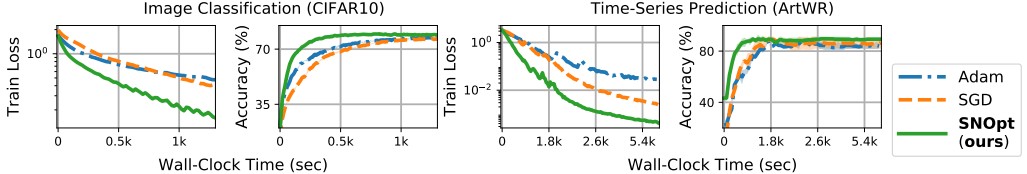

Figure 1: Our second-order method (SNOpt; green solid curves) achieves superior convergence compared to first-order methods (SGD, Adam) on various Neural-ODE applications.

Neural ODEs (Chen et al., 2018) have received tremendous attention over recent years. Inspired by taking the continuous limit of the "discrete" residual transformation, $\mathbf{x}_{k+1} = \mathbf{x}_k + \epsilon F(\mathbf{x}_k, \theta)$, they propose to directly parameterize the vector field of an ODE as a deep neural network (DNN), *i.e.*

$$\frac{\mathrm{d}\mathbf{x}(t)}{\mathrm{d}t} = F(t, \mathbf{x}(t), \theta), \quad \mathbf{x}(t_0) = \boldsymbol{x}_{t_0}, \tag{1}$$

where $\mathbf{x}(t) \in \mathbb{R}^m$ and $F(\cdot, \cdot, \theta)$ is a DNN parameterized by $\theta \in \mathbb{R}^n$. This provides a powerful paradigm connecting modern machine learning to classical differential equations (Weinan, 2017) and has since then achieved promising results on time series analysis (Rubanova et al., 2019; Kidger et al., 2020b), reversible generative flow (Grathwohl et al., 2018; Nguyen et al., 2019), image classification (Zhuang et al., 2020, 2021), and manifold learning (Lou et al., 2020; Mathieu & Nickel, 2020).

Due to the continuous-time representation, Neural ODEs feature a distinct optimization process (see Fig. 2) compared to their discrete-time counterparts, which also poses new challenges. First, the

35th Conference on Neural Information Processing Systems (NeurIPS 2021)

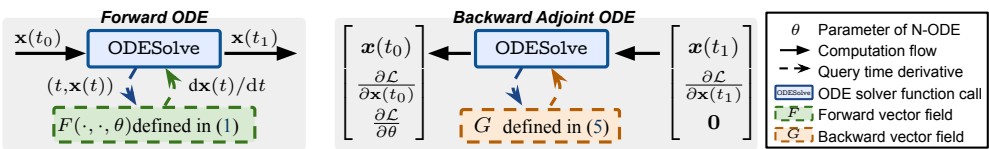

Figure 2: Neural ODE features distinct training process: Both forward and backward passes parameterize vector fields so that any generic ODE solver (which can be non-differentiable) can query time derivatives, *e.g.* $\frac{d\mathbf{x}(t)}{dt}$, to solve the ODEs (1, 5). In this work, we extend it to second-order training.

forward pass of Neural ODEs involves solving (1) with a black-box ODE solver. Depending on how its numerical integration is set up, the propagation may be refined to arbitrarily small step sizes and become prohibitively expensive to solve without any regularization (Ghosh et al., 2020; Finlay et al., 2020). On the other hand, to prevent Back-propagating through the entire ODE solver, the gradients are typically obtained by solving a *backward adjoint ODE* using the Adjoint Sensitivity Method (ASM; Pontryagin et al. (1962)). While this can be achieved at a favorable $\mathcal{O}(1)$ memory, it further increases the runtime and can suffer from inaccurate integration (Gholami et al., 2019). For these reasons, Neural ODEs often take notoriously longer time to train, limiting their applications to relatively small or synthetic datasets (Massaroli et al., 2020) until very recently (Zhuang et al., 2021).

To improve the convergence rate of training, it is natural to consider higher-order optimization. While efficient second-order methods have been proposed for discrete models (Ba et al., 2016; George et al., 2018), it remains unclear how to extend these successes to Neural ODEs, given their distinct computation processes. Indeed, limited discussions in this regard only note that one may repeat the backward adjoint process recursively to obtain higher-order derivatives (Chen et al., 2018). This is, unfortunately, impractical as the recursion will accumulate the aforementioned integration errors and scale the per-iteration runtime linearly. As such, second-order methods for Neural ODEs are seldom considered in practice, nor have they been rigorously explored from an optimization standpoint.

In this work, we show that efficient second-order optimization is in fact viable for Neural ODEs. Our method is inspired by the emerging Optimal Control perspective (Weinan et al., 2018; Liu & Theodorou, 2019), which treats the parameter $\theta$ as a control variable, so that the training process, *i.e.* optimizing $\theta$ w.r.t. some objective, can be interpreted as an Optimal Control Programming (OCP). Specifically, we show that a continuous-time OCP methodology, called *Differential Programming*, provides analytic second-order derivatives by solving a set of coupled matrix ODEs. Interestingly, these matrix ODEs can be augmented to the backward adjoint ODE and solved simultaneously. In other words, a single backward pass is sufficient to compute all derivatives, including the original ASM-based gradient, the newly-derived second-order matrices, or even higher-order tensors. Further, these higher-order computations enjoy the same $\mathcal{O}(1)$ memory and a comparable runtime to first-order methods by adopting Kronecker factorization (Martens & Grosse, 2015). The resulting method – called **SNOpt** – admits superior convergence in wall-clock time (Fig. 1), and the improvement remains consistent across image classification, continuous normalizing flow, and time-series prediction.

Our OCP framework also facilitates progressive training of the network architecture. Specifically, we study an example of jointly optimizing the "integration time" of Neural ODEs, in analogy to the "depth" of discrete DNNs. While analytic gradients w.r.t. this architectural parameter have been derived under the ASM framework, they were often evaluated on limited synthetic datasets (Massaroli et al., 2020). In the context of OCP, however, free-horizon optimization is a well-studied problem for practical applications with a priori unknown terminal time (Sun et al., 2015; De Marchi & Gerdts, 2019). In this work, we show that these principles can be applied to Neural ODEs, yielding a novel second-order *feedback policy* that adapts the integration time throughout training. On training CIFAR10, this further leads to a 20% runtime reduction, yet without hindering test-time accuracy.

In summary, we present the following contributions.

- We propose a novel computational framework for computing higher-order derivatives of deep continuous-time models, with a rigorous analysis using continuous-time Optimal Control theory.

- We propose an efficient second-order method, **SNOpt**, that achieves superior convergence (in wall-clock time) over first-order methods in training Neural ODEs, while retaining constant memory complexity. These improvements remain consistent across various applications.

- To show that our framework also enables direct architecture optimization, we derive a second-order feedback policy for adapting the integration horizon and show it further reduces the runtime.

## 2 Preliminaries

**Notation.** We use roman and italic type to represent a variable $\mathbf{x}(t)$ and its realization $x(t)$ given an ODE. `ODESolve` denotes a function call that solves an initial value problem given an initial condition, start and end integration time, and vector field, *i.e.* `ODESolve`$(\mathbf{x}(t_0), t_0, t_1, F)$ where $\frac{\mathrm{d}\mathbf{x}(t)}{\mathrm{d}t} = F$.

**Forward and backward computations of Neural ODEs.** Given an initial condition $\mathbf{x}(t_0)$ and integration interval $[t_0, t_1]$, Neural ODEs concern the following optimization over an objective $\mathcal{L}$,

$$\min_\theta \mathcal{L}(\mathbf{x}(t_1)), \quad \text{where } \mathbf{x}(t_1) = \mathbf{x}(t_0) + \int_{t_0}^{t_1} F(t, \mathbf{x}(t), \theta) \, \mathrm{d}t \tag{2}$$

is the solution of the ODE (1) and can be solved by calling a black-box ODE solver, *i.e.* $\mathbf{x}(t_1) =$ `ODESolve`$(\mathbf{x}(t_0), t_0, t_1, F)$. The use of `ODESolve` allows us to adopt higher-order numerical methods, *e.g.* adaptive Runge-Kutta (Press et al., 2007), which give more accurate integration compared with *e.g.* vanilla Euler discretization in residual-based discrete models. To obtain the gradient $\frac{\partial \mathcal{L}}{\partial \theta}$ of Neural ODE, one may naively Back-propagate through `ODESolve`. This, even if it could be made possible, leads to unsatisfactory memory complexity since the computation graph can grow arbitrarily large for adaptive ODE solvers. Instead, Chen et al. (2018) proposed to apply the Adjoint Sensitivity Method (ASM), which states that the gradient can be obtained through the following integration.

$$\frac{\partial \mathcal{L}}{\partial \theta} = -\int_{t_1}^{t_0} \mathbf{a}(t)^\mathsf{T} \frac{\partial F(t, \mathbf{x}(t), \theta)}{\partial \theta} \, \mathrm{d}t \,, \tag{3}$$

where $\mathbf{a}(t) \in \mathbb{R}^m$ is referred to the *adjoint* state whose dynamics obey a *backward adjoint ODE*,

$$-\frac{\mathrm{d}\mathbf{a}(t)}{\mathrm{d}t} = \mathbf{a}(t)^\mathsf{T} \frac{\partial F(t, \mathbf{x}(t), \theta)}{\partial \mathbf{x}(t)}, \quad \mathbf{a}(t_1) = \frac{\partial \mathcal{L}}{\partial \mathbf{x}(t_1)} \,. \tag{4}$$

Equations (3, 4) present two coupled ODEs that can be viewed as the continuous-time expression of the Back-propagation (LeCun et al., 1988). Algorithmically, they can be solved through another call of `ODESolve` (see Fig. 2) with an augmented dynamics $G$, *i.e.*

$$\begin{bmatrix} \mathbf{x}(t_0) \\ \mathbf{a}(t_0) \\ \partial\mathcal{L}/\partial\theta \end{bmatrix} = \texttt{ODESolve(} \begin{bmatrix} \mathbf{x}(t_1) \\ \mathbf{a}(t_1) \\ \mathbf{0} \end{bmatrix}, t_1, t_0, G), \text{ where } G\left(t, \begin{bmatrix} \mathbf{x}(t) \\ \mathbf{a}(t) \\ \cdot \end{bmatrix}, \theta\right) := \begin{bmatrix} F(t, \mathbf{x}(t), \theta) \\ -\mathbf{a}(t)^\mathsf{T} \frac{\partial F}{\partial \mathbf{x}} \\ -\mathbf{a}(t)^\mathsf{T} \frac{\partial F}{\partial \theta} \end{bmatrix} \tag{5}$$

augments the original dynamics $F$ in (1) with the adjoint ODEs (3, 4). Notice that this computation (5) depends only on $(\mathbf{x}(t_1), \mathbf{a}(t_1))$. This differs from naive Back-propagation, which requires storing intermediate states along the entire computation graph of forward `ODESolve`. While the latter requires $\mathcal{O}(\widetilde{T})$ memory cost,[1] the computation in (5) only consumes constant $\mathcal{O}(1)$ memory cost.

Chen et al. (2018) noted that if we further encapsulate (5) by $\frac{\partial}{\partial\theta}\mathcal{L} = \texttt{grad}(\mathcal{L}, \theta)$, one may compute higher-order derivatives by recursively calling $\frac{\partial^n \mathcal{L}}{\partial\theta^n} = \texttt{grad}(\frac{\partial^{n-1}\mathcal{L}}{\partial\theta^{n-1}}, \theta)$, starting from $n{=}1$. This can scale unfavorably due to its recursive dependence and accumulated integration errors. Indeed, Table 1

Table 1: Numerical errors between ground-truth and adjoint derivatives using different `ODESolve` on CIFAR10.

|  | rk4 | implicit adams | dopri5 |
|---|---|---|---|
| $\frac{\partial\mathcal{L}}{\partial\theta}$ | $7.63\times10^{-5}$ | $2.11\times10^{-3}$ | $3.44\times10^{-4}$ |
| $\frac{\partial^2\mathcal{L}}{\partial\theta^2}$ | $6.84\times10^{-3}$ | $2.50\times10^{-1}$ | $41.10$ |

suggests that the errors of second-order derivatives, $\frac{\partial^2 \mathcal{L}}{\partial\theta^2}$, obtained from the recursive adjoint procedure can be 2-6 orders of magnitude larger than the ones from the first-order adjoint, $\frac{\partial\mathcal{L}}{\partial\theta}$. In the next section, we will present a novel optimization framework that computes these higher-order derivatives *without* any recursion (Section 3.1) and discuss how it can be implemented efficiently (Section 3.2).

## 3 Approach

### 3.1 Dynamics of Higher-order Derivatives using Continuous-time Optimal Control Theory

OCP perspective is a recently emerging methodology for analyzing optimization of discrete DNNs. Central to its interpretation is to treat the layer propagation of a DNN as discrete-time dynamics, so

---

[1] $\widetilde{T}$ is the number of the adaptive steps used to solve (1), as an analogy of the "depth" of Neural ODEs.

that the training process, *i.e.* finding an *optimal parameter* of *a DNN*, can be understood like an OCP, which searches for an *optimal control* subjected to *a dynamical constraint*. This perspective has provided useful insights on characterizing the optimization process (Hu et al., 2019) and enhancing principled algorithmic design (Liu et al., 2021a). We leave a complete discussion in Appendix A.1.

Lifting this OCP perspective from discrete DNNs to Neural ODEs requires special treatments from continuous-time OCP theory (Todorov, 2016). Nevertheless, we highlight that training Neural ODEs and solving continuous-time OCP are fundamentally intertwined since these models, by construction, represent continuous-time dynamical systems. Indeed, the ASM used for deriving (3, 4) originates from the celebrated Pontryagin's principle (Pontryagin et al., 1962), which is an optimality condition to OCP. Hence, OCP analysis is not only motivated but principled from an optimization standpoint.

We begin by first transforming (2) to a form that is easier to adopt the continuous-time OCP analysis.

$$\min_{\theta} \left[ \Phi(\mathbf{x}_{t_1}) + \int_{t_0}^{t_1} \ell(t, \mathbf{x}_t, \mathbf{u}_t) \mathrm{d}t \right] \quad \text{subjected to} \begin{cases} \frac{\mathrm{d}\mathbf{x}_t}{\mathrm{d}t} = F(t, \mathbf{x}_t, \mathbf{u}_t), & \mathbf{x}_{t_0} = \boldsymbol{x}_{t_0} \\ \frac{\mathrm{d}\mathbf{u}_t}{\mathrm{d}t} = \mathbf{0}, & \mathbf{u}_{t_0} = \theta \end{cases}, \quad (6)$$

where $\mathbf{x}(t) \equiv \mathbf{x}_t$, and etc. It should be clear that (6) describes (2) without loss of generality by having $(\Phi, \ell) := (\mathcal{L}, 0)$. These functions are known as the terminal and intermediate costs in standard OCP. In training Neural ODEs, $\ell$ can be used to describe either the weight decay, *i.e.* $\ell \propto \|\mathbf{u}_t\|$, or more complex regularization (Finlay et al., 2020). The time-invariant ODE imposed for $\mathbf{u}_t$ makes the ODE of $\mathbf{x}_t$ equivalent to (1). Problem (6) shall be understood as a particular type of OCP that searches for an optimal initial condition $\theta$ of a time-invariant control $\mathbf{u}_t$. Despite seemly superfluous, this is a necessary transformation that enables rigorous OCP analysis for the original training process (2), and it has also appeared in other control-related analyses (Zhong et al., 2020; Chalvidal et al., 2021).

Next, define the accumulated loss from any time $t \in [t_0, t_1]$ to the integration end time $t_1$ as

$$Q(t, \mathbf{x}_t, \mathbf{u}_t) := \Phi(\mathbf{x}_{t_1}) + \int_t^{t_1} \ell(\tau, \mathbf{x}_\tau, \mathbf{u}_\tau) \, \mathrm{d}\tau, \quad (7)$$

which is also known in OCP as the *cost-to-go* function. Recall that our goal is to compute higher-order derivatives w.r.t. the parameter $\theta$ of Neural ODEs. Under the new OCP representation (6), the first-order derivative $\frac{\partial \mathcal{L}}{\partial \theta}$ is identical to $\frac{\partial Q(t_0, \mathbf{x}_{t_0}, \mathbf{u}_{t_0})}{\partial \mathbf{u}_{t_0}}$. This is because $Q(t_0, \mathbf{x}_{t_0}, \mathbf{u}_{t_0})$ accumulates all sources of losses between $[t_0, t_1]$ (hence it sufficiently describes $\mathcal{L}$) and $\mathbf{u}_{t_0} = \theta$ by construction. Likewise, the second-order derivatives can be captured by the Hessian $\frac{\partial^2 Q(t_0, \mathbf{x}_{t_0}, \mathbf{u}_{t_0})}{\partial \mathbf{u}_{t_0} \partial \mathbf{u}_{t_0}} = \frac{\partial^2 \mathcal{L}}{\partial \theta \partial \theta} \equiv \mathcal{L}_{\theta\theta}$. In other words, we are only interested in obtaining the derivatives of $Q$ at the integration start time $t_0$.

To obtain these derivatives, notice that we can rewrite (7) as

$$0 = \ell(t, \mathbf{x}_t, \mathbf{u}_t) + \frac{\mathrm{d}Q(t, \mathbf{x}_t, \mathbf{u}_t)}{\mathrm{d}t}, \quad Q(t_1, \mathbf{x}_{t_1}) = \Phi(\mathbf{x}_{t_1}), \quad (8)$$

since the definition of $Q$ implies that $Q(t, \mathbf{x}_t, \mathbf{u}_t) = \ell(t, \mathbf{x}_t, \mathbf{u}_t)\mathrm{d}t + Q(t + \mathrm{d}t, \mathbf{x}_{t+\mathrm{d}t}, \mathbf{u}_{t+\mathrm{d}t})$. We now state our main result, which provides a local characterization of (8) with a set of coupled ODEs expanded along a solution path. These ODEs can be used to obtain all second-order derivatives at $t_0$.

**Theorem 1** (Second-order Differential Programming). *Consider a solution path $(\bar{\boldsymbol{x}}_t, \bar{\boldsymbol{u}}_t)$ that solves the ODEs in (6). Then the first and second-order derivatives of $Q(t, \mathbf{x}_t, \mathbf{u}_t)$, expanded locally around this solution path, obey the following backward ODEs:*

$$-\frac{\mathrm{d}Q_{\bar{\boldsymbol{x}}}}{\mathrm{d}t} = \ell_{\bar{\boldsymbol{x}}} + F_{\bar{\boldsymbol{x}}}^\mathsf{T} Q_{\bar{\boldsymbol{x}}}, \qquad\qquad -\frac{\mathrm{d}Q_{\bar{\boldsymbol{u}}}}{\mathrm{d}t} = \ell_{\bar{\boldsymbol{u}}} + F_{\bar{\boldsymbol{u}}}^\mathsf{T} Q_{\bar{\boldsymbol{x}}}, \qquad (9a)$$

$$-\frac{\mathrm{d}Q_{\bar{\boldsymbol{x}}\bar{\boldsymbol{x}}}}{\mathrm{d}t} = \ell_{\bar{\boldsymbol{x}}\bar{\boldsymbol{x}}} + F_{\bar{\boldsymbol{x}}}^\mathsf{T} Q_{\bar{\boldsymbol{x}}\bar{\boldsymbol{x}}} + Q_{\bar{\boldsymbol{x}}\bar{\boldsymbol{x}}} F_{\bar{\boldsymbol{x}}}, \quad -\frac{\mathrm{d}Q_{\bar{\boldsymbol{x}}\bar{\boldsymbol{u}}}}{\mathrm{d}t} = \ell_{\bar{\boldsymbol{x}}\bar{\boldsymbol{u}}} + Q_{\bar{\boldsymbol{x}}\bar{\boldsymbol{x}}} F_{\bar{\boldsymbol{u}}} + F_{\bar{\boldsymbol{x}}}^\mathsf{T} Q_{\bar{\boldsymbol{x}}\bar{\boldsymbol{u}}}, \quad (9b)$$

$$-\frac{\mathrm{d}Q_{\bar{\boldsymbol{u}}\bar{\boldsymbol{u}}}}{\mathrm{d}t} = \ell_{\bar{\boldsymbol{u}}\bar{\boldsymbol{u}}} + F_{\bar{\boldsymbol{u}}}^\mathsf{T} Q_{\bar{\boldsymbol{x}}\bar{\boldsymbol{u}}} + Q_{\bar{\boldsymbol{u}}\bar{\boldsymbol{x}}} F_{\bar{\boldsymbol{u}}}, \quad -\frac{\mathrm{d}Q_{\bar{\boldsymbol{u}}\bar{\boldsymbol{x}}}}{\mathrm{d}t} = \ell_{\bar{\boldsymbol{u}}\bar{\boldsymbol{x}}} + F_{\bar{\boldsymbol{u}}}^\mathsf{T} Q_{\bar{\boldsymbol{x}}\bar{\boldsymbol{x}}} + Q_{\bar{\boldsymbol{u}}\bar{\boldsymbol{x}}} F_{\bar{\boldsymbol{x}}}, \quad (9c)$$

*where $F_{\bar{\boldsymbol{x}}}(t) \equiv \frac{\partial F}{\partial \mathbf{x}_t}|_{(\bar{\boldsymbol{x}}_t, \bar{\boldsymbol{u}}_t)}$, $Q_{\bar{\boldsymbol{x}}\bar{\boldsymbol{x}}}(t) \equiv \frac{\partial^2 Q}{\partial \mathbf{x}_t \partial \mathbf{x}_t}|_{(\bar{\boldsymbol{x}}_t, \bar{\boldsymbol{u}}_t)}$, and etc. All terms in (9) are time-varying vector-valued or matrix-valued functions expanded at $(\bar{\boldsymbol{x}}_t, \bar{\boldsymbol{u}}_t)$. The terminal condition is given by*

$$Q_{\bar{\boldsymbol{x}}}(t_1) = \Phi_{\bar{\boldsymbol{x}}}, \quad Q_{\bar{\boldsymbol{x}}\bar{\boldsymbol{x}}}(t_1) = \Phi_{\bar{\boldsymbol{x}}\bar{\boldsymbol{x}}}, \quad \text{and} \quad Q_{\bar{\boldsymbol{u}}}(t_1) = Q_{\bar{\boldsymbol{u}}\bar{\boldsymbol{u}}}(t_1) = Q_{\bar{\boldsymbol{u}}\bar{\boldsymbol{x}}}(t_1) = Q_{\bar{\boldsymbol{x}}\bar{\boldsymbol{u}}}(t_1) = \mathbf{0}.$$

The proof (see Appendix A.2) relies on rewriting (8) with *differential states*, $\delta \mathbf{x}_t := \mathbf{x}_t - \bar{\mathbf{x}}_t$, which view the deviation from $\bar{\mathbf{x}}_t$ as an *optimizing variable* (hence the name "*Differential Programming*"). It can be shown that $\delta \mathbf{x}_t$ follows a linear ODE expanded along the solution path. Theorem 1 has several important implications. First, the ODEs in (9a) recover the original ASM computation (3,4), as one can readily verify that $Q_{\bar{\boldsymbol{x}}}(t) \equiv \mathbf{a}(t)$ follows the same backward ODE in (4) and the solution of the second ODE in (9a), $Q_{\bar{\boldsymbol{u}}}(t_0) = -\int_{t_1}^{t_0} F_{\bar{\boldsymbol{u}}}{}^\mathsf{T} Q_{\bar{\boldsymbol{x}}} \mathrm{d}t$, gives the exact gradient in (3). Meanwhile, solving the coupled matrix ODEs presented in (9b, 9c) yields the desired second-order matrix, $Q_{\bar{\boldsymbol{u}}\bar{\boldsymbol{u}}}(t_0) \equiv \mathcal{L}_{\theta\theta}$, for preconditioning the update. Finally, one can derive the dynamics of other higher-order tensors using the same Differential Programming methodology by simply expanding (8) beyond the second order. We leave some discussions in this regard in Appendix A.2.

## 3.2 Efficient Second-order Preconditioned Update

Theorem 1 provides an attractive computational framework that does not require recursive computation (as mentioned in Section 2) to obtain higher-order derivatives. It suggests that we can obtain first and second-order derivatives all at once with a single function call of `ODESolve`:

$$
\begin{aligned}
[\boldsymbol{x}_{t_0}, Q_{\bar{\boldsymbol{x}}}(t_0), Q_{\bar{\boldsymbol{u}}}(t_0), Q_{\bar{\boldsymbol{x}}\bar{\boldsymbol{x}}}(t_0), Q_{\bar{\boldsymbol{u}}\bar{\boldsymbol{x}}}(t_0), Q_{\bar{\boldsymbol{x}}\bar{\boldsymbol{u}}}(t_0), Q_{\bar{\boldsymbol{u}}\bar{\boldsymbol{u}}}(t_0)] \\
= \mathtt{ODESolve}([\boldsymbol{x}_{t_1}, \Phi_{\bar{\boldsymbol{x}}}, \mathbf{0}, \Phi_{\bar{\boldsymbol{x}}\bar{\boldsymbol{x}}}, \mathbf{0}, \mathbf{0}, \mathbf{0}], t_1, t_0, \tilde{G}),
\end{aligned}
\tag{10}
$$

where $\tilde{G}$ augments the original dynamics $F$ in (1) with all 6 ODEs presented in (9). Despite that this OCP-theoretic backward pass (10) retains the same $\mathcal{O}(1)$ memory complexity as in (5), the dimension of the new augmented state, which now carries second-order matrices, can grow to an unfavorable size that dramatically slows down the numerical integration. Hence, we must consider other representations of (9), if any, in order to proceed. In the following proposition, we present one of which that transforms (9) into a set of vector ODEs, so that we can compute them much efficiently.

**Proposition 2** (Low-rank representation of (9)). *Suppose $\ell := 0$ in (6) and let $Q_{\bar{\boldsymbol{x}}\bar{\boldsymbol{x}}}(t_1) = \sum_{i=1}^{R} \boldsymbol{y}_i \otimes \boldsymbol{y}_i$ be a symmetric matrix of rank $R \le n$, where $\boldsymbol{y}_i \in \mathbb{R}^m$ and $\otimes$ is the Kronecker product. Then, for all $t \in [t_0, t_1]$, the second-order matrices appeared in (9b, 9c) can be decomposed into*

$$
Q_{\bar{\boldsymbol{x}}\bar{\boldsymbol{x}}}(t) = \sum_{i=1}^{R} \mathbf{q}_i(t) \otimes \mathbf{q}_i(t), \quad Q_{\bar{\boldsymbol{x}}\bar{\boldsymbol{u}}}(t) = \sum_{i=1}^{R} \mathbf{q}_i(t) \otimes \mathbf{p}_i(t), \quad Q_{\bar{\boldsymbol{u}}\bar{\boldsymbol{u}}}(t) = \sum_{i=1}^{R} \mathbf{p}_i(t) \otimes \mathbf{p}_i(t),
$$

*where the vectors $\mathbf{q}_i(t) \in \mathbb{R}^m$ and $\mathbf{p}_i(t) \in \mathbb{R}^n$ obey the following backward ODEs:*

$$
-\frac{\mathrm{d}\mathbf{q}_i(t)}{\mathrm{d}t} = F_{\bar{\boldsymbol{x}}}(t)^\mathsf{T} \mathbf{q}_i(t), \quad -\frac{\mathrm{d}\mathbf{p}_i(t)}{\mathrm{d}t} = F_{\bar{\boldsymbol{u}}}(t)^\mathsf{T} \mathbf{q}_i(t),
\tag{11}
$$

*with the terminal condition given by $(\mathbf{q}_i(t_1), \mathbf{p}_i(t_1)) := (\boldsymbol{y}_i, \mathbf{0})$.*

The proof is left in Appendix A.2. Proposition 2 gives a nontrivial conversion. It indicates that the *coupled matrix* ODEs presented in (9b, 9c) can be disentangled into a set of *independent vector* ODEs where each of them follows its own dynamics (11). As the rank $R$ determines the number of these vector ODEs, this conversion will be particularly useful if the second-order matrices exhibit low-rank structures. Fortunately, this is indeed the case for many Neural-ODE applications which often propagate $\mathbf{x}_t$ in a latent space of higher dimension (Chen et al., 2018; Grathwohl et al., 2018; Kidger et al., 2020b).

Based on Proposition 2, the second-order precondition matrix $\mathcal{L}_{\theta\theta}$ is given by[2]

$$
\mathcal{L}_{\theta\theta} \equiv Q_{\bar{\boldsymbol{u}}\bar{\boldsymbol{u}}}(t_0) = \sum_{i=1}^{R} \left( \int_{t_1}^{t_0} F_{\bar{\boldsymbol{u}}}{}^\mathsf{T} \mathbf{q}_i \, \mathrm{d}t \right) \otimes \left( \int_{t_1}^{t_0} F_{\bar{\boldsymbol{u}}}{}^\mathsf{T} \mathbf{q}_i \, \mathrm{d}t \right),
\tag{12}
$$

where $\mathbf{q}_i \equiv \mathbf{q}_i(t)$ follows (11). Our final step is to facilitate efficient computation of (12) with Kronecker-based factorization, which underlines many popular second-order methods for discrete DNNs (Grosse & Martens, 2016; Martens et al., 2018). Recall that the vector field $F$ is represented

---

[2] We drop the dependence on $t$ for brevity, yet all terms inside the integrations of (12, 13) are time-varying.

**Algorithm 1** SNOpt: Second-order Neural ODE Optimizer
---
1: **Input:** dataset $\mathcal{D}$, parametrized vector field $F(\cdot, \cdot, \theta)$, integration time $[t_0, t_1]$, black-box ODE solver `ODESolve`, learning rate $\eta$, rank $R$, interval of the time grid $\Delta t$
2: **repeat**
3:     Solve $\mathbf{x}(t_1) = $ `ODESolve`$(\mathbf{x}(t_0), t_0, t_1, F)$, where $\mathbf{x}(t_0) \sim \mathcal{D}$.          ▷ Forward pass
4:     Initialize $(\bar{\boldsymbol{A}}_n, \bar{\boldsymbol{B}}_n) := (\mathbf{0}, \mathbf{0})$ for each layer $n$ and set $\mathbf{q}_i(t_1) := \boldsymbol{y}_i$.
5:     **for** $t'$ in $\{t_1, t_1 - \Delta t, \cdots, t_0 + \Delta t, t_0\}$ **do**
6:         Set $t := t' - \Delta t$ as the small integration step, then call
           $[\mathbf{x}(t), Q_{\bar{\boldsymbol{x}}}(t), Q_{\bar{\boldsymbol{u}}}(t), \{\mathbf{q}_i(t)\}_{i=1}^R]$
              $=$ `ODESolve`$([\mathbf{x}(t'), Q_{\bar{\boldsymbol{x}}}(t'), Q_{\bar{\boldsymbol{u}}}(t'), \{\mathbf{q}_i(t')\}_{i=1}^R], t', t, \widehat{G})$,     ▷ Backward pass
        where $\widehat{G}$ augments the ODEs of state (1), first and second-order derivatives (9a, 11).
7:         Evaluate $\mathbf{z}^n(t), \mathbf{h}^n(t), F(t, \mathbf{x}_t, \theta)$, then compute $\boldsymbol{A}_n(t), \boldsymbol{B}_n(t)$ in (13).
8:         Update $\bar{\boldsymbol{A}}_n \leftarrow \bar{\boldsymbol{A}}_n + \boldsymbol{A}_n(t) \cdot \Delta t$ and $\bar{\boldsymbol{B}}_n \leftarrow \bar{\boldsymbol{B}}_n + \boldsymbol{B}_n(t) \cdot \Delta t$.
9:     **end for**
10:    $\forall n$, apply $\theta^n \leftarrow \theta^n - \eta \cdot \text{vec}(\bar{\boldsymbol{B}}_n^{-1} Q_{\bar{\boldsymbol{u}}^n}(t_0) \bar{\boldsymbol{A}}_n^{-\mathsf{T}})$.      ▷ Second-order parameter update
11: **until** converges
---

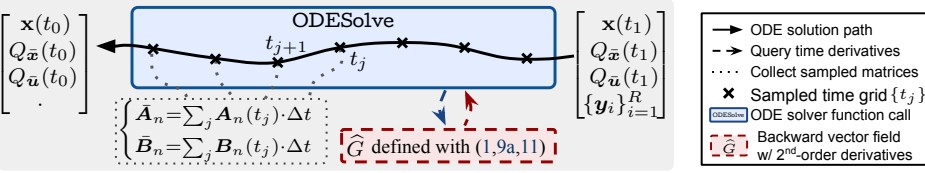

Figure 4: Our second-order method, SNOpt, solves a new backward ODE, *i.e.* the $\widehat{G}$ appeared in line 6 of Alg. 1, which augments second-order derivatives, while simultaneously collecting the matrices $\boldsymbol{A}_n(t_j)$ and $\boldsymbol{B}_n(t_j)$ on a sampled time grid $\{t_j\}$ for computing the preconditioned update in (14).

by a DNN. Let $\mathbf{z}^n(t), \mathbf{h}^n(t)$, and $\mathbf{u}^n(t)$ denote the activation vector, pre-activation vector, and the parameter of layer $n$ when evaluating $\frac{d\mathbf{x}}{dt}$ at time $t$ (see Fig. 3), then the integration in (12) can be broken down into each layer $n$,

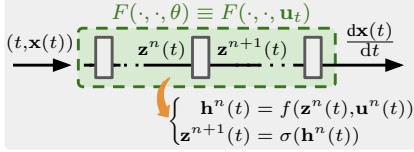

$$\int_{t_1}^{t_0} \left( F_{\bar{\boldsymbol{u}}}^{\mathsf{T}} \mathbf{q}_i \right) dt = [\cdots, \int_{t_1}^{t_0} \left( F_{\bar{\boldsymbol{u}}^n}^{\mathsf{T}} \mathbf{q}_i \right) dt, \cdots]$$

$$= [\cdots, \int_{t_1}^{t_0} \left( \mathbf{z}^n \otimes \left( \frac{\partial F}{\partial \mathbf{h}^n}^{\mathsf{T}} \mathbf{q}_i \right) \right) dt, \cdots],$$

Figure 3: The layer propagation inside the vector field $F$, where $f$ and $\sigma$ denote affine and nonlinear activation functions.

where the second equality holds by $F_{\bar{\boldsymbol{u}}^n}^{\mathsf{T}} \mathbf{q}_i = \left( \frac{\partial F}{\partial \mathbf{h}^n} \frac{\partial \mathbf{h}^n}{\partial \mathbf{u}^n} \right)^{\mathsf{T}} \mathbf{q}_i = \mathbf{z}^n \otimes \left( \frac{\partial F}{\partial \mathbf{h}^n}^{\mathsf{T}} \mathbf{q}_i \right)$. This is an essential step towards the Kronecker approximation of the layer-wise precondition matrix:

$$\mathcal{L}_{\theta^n \theta^n} \equiv Q_{\bar{\boldsymbol{u}}^n \bar{\boldsymbol{u}}^n}(t_0) = \sum_{i=1}^R \left( \int_{t_1}^{t_0} \left( \mathbf{z}^n \otimes \left( \frac{\partial F}{\partial \mathbf{h}^n}^{\mathsf{T}} \mathbf{q}_i \right) \right) dt \right) \otimes \left( \int_{t_1}^{t_0} \left( \mathbf{z}^n \otimes \left( \frac{\partial F}{\partial \mathbf{h}^n}^{\mathsf{T}} \mathbf{q}_i \right) \right) dt \right)$$

$$\approx \underbrace{\int_{t_1}^{t_0} (\mathbf{z}^n \otimes \mathbf{z}^n) \, dt}_{\boldsymbol{A}_n(t)} \otimes \underbrace{\int_{t_1}^{t_0} \sum_{i=1}^R \left( \left( \frac{\partial F}{\partial \mathbf{h}^n}^{\mathsf{T}} \mathbf{q}_i \right) \otimes \left( \frac{\partial F}{\partial \mathbf{h}^n}^{\mathsf{T}} \mathbf{q}_i \right) \right) dt}_{\boldsymbol{B}_n(t)}. \quad (13)$$

We discuss the approximation behind (13), and also the one for (14), in Appendix A.2. Note that $\boldsymbol{A}_n(t)$ and $\boldsymbol{B}_n(t)$ are much smaller matrices in $\mathbb{R}^{m \times m}$ compared to the ones in (9), and they can be efficiently computed with automatic differentiation packages (Paszke et al., 2017). Now, let $\{t_j\}$ be a time grid uniformly distributed over $[t_0, t_1]$ so that $\bar{\boldsymbol{A}}_n = \sum_j \boldsymbol{A}_n(t_j) \Delta t$ and $\bar{\boldsymbol{B}}_n = \sum_j \boldsymbol{B}_n(t_j) \Delta t$ approximate the integrations in (13), then our final preconditioned update law is given by

$$\forall n, \quad \mathcal{L}_{\theta^n \theta^n}^{-1} \mathcal{L}_{\theta^n} \approx \text{vec}\left( \bar{\boldsymbol{B}}_n^{-1} Q_{\bar{\boldsymbol{u}}^n}(t_0) \bar{\boldsymbol{A}}_n^{-\mathsf{T}} \right), \quad (14)$$

where vec denotes vectorization. Our second-order method – named **SNOpt** – is summarized in Alg. 1, with the backward computation (*i.e.* line 4-9 in Alg. 1) illustrated in Fig. 4. In practice, we also adopt eigen-based amortization with Tikhonov regularization (George et al. (2018); see Alg. 2 in Appendix A.4), which stabilizes the updates over stochastic training.

**Remark.** The fact that Proposition 2 holds only for degenerate $\ell$ can be easily circumvented in practice. As $\ell$ typically represents weight decay, $\ell := \frac{1}{t_1 - t_0} \|\theta\|_2$, which is time-independent, it can be separated from the backward ODEs (9) and added after solving the backward integration, *i.e.*

$$Q_{\bar{\mathbf{u}}}(t_0) \leftarrow \gamma\theta + Q_{\bar{\mathbf{u}}}(t_0), \quad Q_{\bar{\mathbf{u}}\bar{\mathbf{u}}}(t_0) \leftarrow \gamma\mathbf{I} + Q_{\bar{\mathbf{u}}\bar{\mathbf{u}}}(t_0),$$

where $\gamma$ is the regularization factor. Finally, we find that using the scaled *Gaussian-Newton* matrix, *i.e.* $Q_{\bar{\mathbf{x}}\bar{\mathbf{x}}}(t_1) \approx \frac{1}{t_1 - t_0}\Phi_{\bar{\mathbf{x}}} \otimes \Phi_{\bar{\mathbf{x}}}$, generally provides a good trade-off between the performance and runtime complexity. As such, we adopt this approximation to Proposition 2 for all experiments.

### 3.3 Memory Complexity Analysis

Table 2: Memory complexity at different stages of our derivation in terms of $\mathbf{x}_t \in \mathbb{R}^m$, $\theta \in \mathbb{R}^n$, and the rank $R$. Note that *all* methods have $\mathcal{O}(1)$ in terms of depth.

|  | Theorem 1 Eqs. (9,10) | Proposition 2 Eqs. (11,12) | **SNOpt** (Alg. 1) Eqs. (13,14) | first-order adjoint Eqs. (3,4) |
|---|---|---|---|---|
| backward storage | $\mathcal{O}((m+n)^2)$ | $\mathcal{O}(Rm + Rn)$ | $\mathcal{O}(Rm + 2n)$ | $\mathcal{O}(m+n)$ |
| parameter update | $\mathcal{O}(n^2)$ | $\mathcal{O}(n^2)$ | $\mathcal{O}(2n)$ | $\mathcal{O}(n)$ |

Table 2 summarizes the memory complexity of different computational methods that appeared along our derivation in Section 3.1 and 3.2. Despite that *all* methods retain $\mathcal{O}(1)$ memory as with the first-order adjoint method, their complexity differs in terms of the state and parameter dimension. Starting from our encouraging result in Theorem 1, which allows one to compute all derivatives with a single backward pass, we first exploit their low-rank representation in Proposition 2. This reduces the storage to $\mathcal{O}(Rm + Rn)$ and paves a way toward adopting Kronecker factorization, which further facilitates efficient preconditioning. With all these, our SNOpt is capable of performing efficient second-order updates while enjoying similar memory complexity (up to some constant) compared to first-order adjoint methods. Lastly, for image applications where Neural ODEs often consist of convolution layers, we adopt convolution-based Kronecker factorization (Grosse & Martens, 2016; Gao et al., 2020), which effectively makes the complexity to scale w.r.t. the number of feature maps (*i.e.* number of channels) rather than the full size of feature maps.

### 3.4 Extension to Architecture Optimization

Let us discuss an intriguing extension of our OCP framework to optimizing the architecture of Neural ODEs, specifically the integration bound $t_1$. In practice, when problems contain no prior information on the integration, $[t_0, t_1]$ is typically set to some trivial values (usually $[0, 1]$) without further justification. However, these values can greatly affect both the performance and runtime. Take CIFAR10 for instance (see Fig. 5), the required training time decreases linearly as we drop $t_1$ from 1, yet the accuracy retains mostly the same unless $t_1$ becomes too small. Similar results also appear on MNIST (see Fig. 12 in Appendix A.5). In other words, we may interpret the integration bound $t_1$ as an *architectural parameter* that needs to be jointly optimized during training.

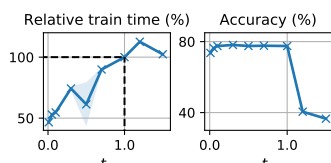

Figure 5: Training performance of CIFAR10 with Adam when using different $t_1$, which motivates joint optimization of $t_1$. Experiment setup is left in Appendix A.4.

The aforementioned interpretation fits naturally into our OCP framework. Specifically, we can consider the following extension of $Q$, which introduces the terminal time T as a new variable:

$$\widetilde{Q}(t, \mathbf{x}_t, \mathbf{u}_t, \mathrm{T}) := \widetilde{\Phi}(\mathrm{T}, \mathbf{x}(\mathrm{T})) + \int_t^{\mathrm{T}} \ell(\tau, \mathbf{x}_\tau, \mathbf{u}_\tau) \, \mathrm{d}\tau, \tag{15}$$

where $\widetilde{\Phi}(\mathrm{T}, \mathbf{x}(\mathrm{T}))$ explicitly imposes the penalty for longer integration time, *e.g.* $\widetilde{\Phi} := \Phi(\mathbf{x}(\mathrm{T})) + \frac{c}{2}\mathrm{T}^2$. Following a similar procedure presented in Section 3.1, we can transform (15) into its ODE form (as in (8)) then characterize its local behavior (as in (9)) along a solution path $(\bar{\mathbf{x}}_t, \bar{\mathbf{u}}_t, \bar{T})$. After some tedious derivations, which are left in Appendix A.3, we will arrive at the update rule below,

$$\mathrm{T} \leftarrow \bar{T} - \eta \cdot \delta\mathrm{T}(\delta\theta), \quad \text{where} \quad \delta\mathrm{T}(\delta\theta) = [\widetilde{Q}_{\bar{T}\bar{T}}(t_0)]^{-1}\left(\widetilde{Q}_{\bar{T}}(t_0) + \widetilde{Q}_{\bar{T}\bar{\mathbf{u}}}(t_0)\delta\theta\right). \tag{16}$$

Similar to what we have discussed in Section 3.1, one shall view $\widetilde{Q}_{\bar{T}}(t_0) \equiv \frac{\partial \mathcal{L}}{\partial \mathrm{T}}$ as the first-order derivative w.r.t. the terminal time T. Likewise, $\widetilde{Q}_{\bar{T}\bar{T}}(t_0) \equiv \frac{\partial^2 \mathcal{L}}{\partial \mathrm{T}\partial \mathrm{T}}$, and etc. Equation (16) is a second-order *feedback* policy that adjusts its updates based on the change of the parameter $\theta$. Intuitively, it moves in the descending direction of the preconditioned gradient (*i.e.* $\widetilde{Q}_{\bar{T}\bar{T}}^{-1}\widetilde{Q}_{\bar{T}}$), while accounting for the fact that $\theta$ *is also progressing during training* (via the feedback $\widetilde{Q}_{\bar{T}\bar{u}}\delta\theta$). The latter is a distinct feature arising from the OCP principle. As we will show later, this update (16) leads to distinct behavior with superior convergence compared to first-order baselines (Massaroli et al., 2020).

## 4 Experiments

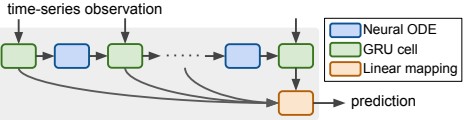

Figure 6: Hybrid model for time-series prediction.

Table 3: Sample size of time-series datasets (input dimension, class label, series length)

| SpoAD | ArtWR | CharT |
|---|---|---|
| (27, 10, 93) | (19, 25, 144) | (7, 20, 187) |

**Dataset.** We select 9 datasets from 3 distinct applications where N-ODEs have been applied, including image classification (●), time-series prediction (●), and continuous normalizing flow (●; CNF):

- **MNIST**, **SVHN**, **CIFAR10**: MNIST consists of 28×28 gray-scale images, while SVHN and CIFAR10 consist of 3×32×32 colour images. All 3 image datasets have 10 label classes.

- **SpoAD**, **ArtWR**, **CharT**: We consider UEA time series archive (Bagnall et al., 2018). Spoken-ArabicDigits (SpoAD) is a speech dataset, whereas ArticularyWordRecognition (ArtWR) and CharacterTrajectories (CharT) are motion-related datasets. Table 3 details their sample sizes.

- **Circle**, **Gas**, **Miniboone**: Circle is a 2-dim synthetic dataset adopted from Chen et al. (2018). Gas and Miniboone are 8 and 43-dim tabular datasets commonly used in CNF (Grathwohl et al., 2018; Onken et al., 2020). All 3 datasets transform a multivariate Gaussian to the target distributions.

**Models.** The models for image datasets and CNF resemble standard feedforward networks, except now consisting of Neural ODEs as continuous transformation layers. Specifically, the models for image classification consist of convolution-based feature extraction, followed by a Neural ODE and linear mapping. Meanwhile, the CNF models are identical to the ones in Grathwohl et al. (2018), which consist of 1-5 Neural ODEs, depending on the size of the dataset. As for the time-series models, we adopt the hybrid models from Rubanova et al. (2019), which consist of a Neural ODE for hidden state propagation, standard recurrent cell (*e.g.* GRU (Cho et al., 2014)) to incorporate incoming time-series observation, and a linear prediction layer. Figure 6 illustrates this process. We detail other configurations in Appendix A.4.

**ODE solver.** We use standard Runge-Kutta 4(5) adaptive solver (dopri5; Dormand & Prince (1980)) implemented by the torchdiffeq package. The numerical tolerance is set to 1e-6 for CNF and 1e-3 for the rest. We fix the integration time to $[0, 1]$ whenever it appears as a hyper-parameter (*e.g.* for image and CNF datasets[3]); otherwise we adopt the problem-specific setup (*e.g.* for time series).

**Training setup.** We consider Adam and SGD (with momentum) as the first-order baselines since they are default training methods for most Neural-ODE applications. As for our second-order SNOpt, we set up the time grid $\{t_j\}$ such that it collects roughly 100 samples along the backward integration to estimate the precondition matrices (see Fig. 4). The hyper-parameters (*e.g.* learning rate) are tuned for each method on each dataset, and we detail the tuning process in Appendix A.4. We also employ practical acceleration techniques, including the semi-norm (Kidger et al., 2020a) for speeding up ODESolve, and the Jacobian-free estimator (FFJORD; Grathwohl et al. (2018)) for accelerating CNF models. The batch size is set to 256, 512, and 1000 respectively for ArtWord, CharTraj, and Gas. The rest of the datasets use 128 as the batch size. All experiments are conducted on a TITAN RTX.

### 4.1 Results

**Convergence and computation efficiency.** Figures 1 and 7 report the training curves of each method measured by wall-clock time. It is obvious that our SNOpt admits a superior convergence

---

[3] except for Circle where we set $[t_0, t_1]:=[0, 10]$ in order to match the original setup in Chen et al. (2018).

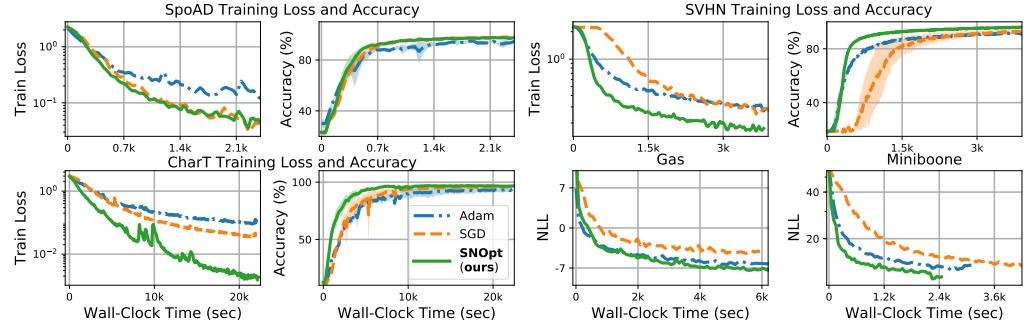

Figure 7: Training performance in *wall-clock* runtime, averaged over 3 trials. Our SNOpt achieves faster convergence against first-order baselines. See Fig. 14 in Appendix A.5 for MNIST and Circle.

Table 4: Test-time performance: accuracies for image and time-series datasets; NLL for CNF datasets

|       | MNIST | SVHN | CIFAR10 | SpoAD | ArtWR | CharT | Circle | Gas | Miniboone |
|-------|-------|------|---------|-------|-------|-------|--------|------|-----------|
| Adam  | 98.83 | 91.92 | 77.41  | 94.64 | 84.14 | 93.29 | 0.90   | -6.42 | 13.10    |
| SGD   | 98.68 | 93.34 | 76.42  | **97.70** | 85.82 | 95.93 | 0.94 | -4.58 | 13.75 |
| **SNOpt** | **98.99** | **95.77** | **79.11** | 97.41 | **90.23** | **96.63** | **0.86** | **-7.55** | **12.50** |

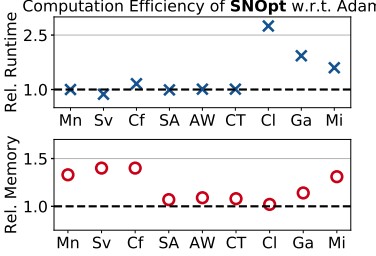

Figure 8: Relative runtime and memory of our SNOpt compared to Adam (denoted by the dashed black lines) on all 9 datasets, where 'Mn' is the shorthand for MNIST, and *etc.*

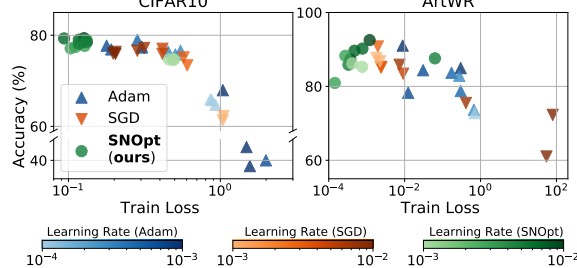

Figure 9: Sensitivity analysis where each sample represents a training result using different optimizer and learning rate (annotated by different symbol and color). Our SNOpt achieves higher accuracies and is insensitive to hyperparameter changes. Note that x-axes are in $\log$ scale.

rate compared to the first-order baselines, and in many cases exceeds their performances by a large margin. In Fig. 8, we report the computation efficiency of our SNOpt compared to Adam on each dataset, and leave their numerical values in Appendix A.4 (Table 9 and 10). For image and time-series datasets (*i.e.* Mn~CT), our SNOpt runs nearly as fast as first-order methods. This is made possible through a rigorous OCP analysis in Section 3, where we showed that second-order matrices can be constructed along with the *same* backward integration when we compute the gradient. Hence, only a minimal overhead is introduced. As for CNF, which propagates the probability density additional to the vanilla state dynamics, our SNOpt is roughly 1.5 to 2.5 times slower, yet it still converges faster in the overall wall-clock time (see Fig. 7). On the other hand, the use of second-order matrices increases the memory consumption of SNOpt by 10-40%, depending on the model and dataset. However, the actual increase in memory (less than 1GB for all datasets; see Table 10) remains affordable on standard GPU machines. More importantly, our SNOpt retains the $\mathcal{O}(1)$ memory throughout training.

**Test-time performance and hyper-parameter sensitivity.** Table 4 reports the test-time performance, including the accuracies (%) for image and time-series classification, and the negative log-likelihood (NLL) for CNF. On most datasets, our method achieves competitive results against standard baselines. In practice, we also find that using the preconditioned updates greatly reduce the sensitivity to hyper-parameters (*e.g.* learning rate). This is demonstrated in Fig. 9, where we sample distinct learning rates from a proper interval for each method (shown with different color bars) and record their training results after convergence. It is clear that our method not only converges to higher

Table 5: Performance of jointly optimizing the integration bound $t_1$ on CIFAR10

| Method | Train time (%) w.r.t. $t_1=1.0$ | Accuracy (%) |
|---|---|---|
| ASM baseline | 96 | 76.61 |
| **SNOpt (ours)** | **81** | **77.82** |

Table 6: Measure of implicit regularization on SVHN

| | # of function evaluation (NFE) | Regularization ($\int \|\nabla_{\mathbf{x}} F\|^2 + \int \|F\|^2$) |
|---|---|---|
| Adam | 42.1 | 323.88 |
| **SNOpt** | **32.6** | **199.1** |

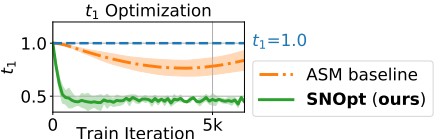

Figure 10: Dynamics of $t_1$ over CIFAR10 training using different methods.

Figure 11: Comparison between SNOpt and second-order recursive adjoint. SNOpt is at least 2 times faster and improves the accuracies of baselines by 5-15%.

accuracies with lower losses, these values are also more concentrated on the plots. In other words, our method achieves better convergence in a more consistent manner across different hyper-parameters.

**Joint optimization of the integration bound** $t_1$**.** Table 5 and Fig. 10 report the performance of optimizing $t_1$ along with its convergence dynamics. Specifically, we compare our second-order feedback policy (16) derived in Section 3.4 to the first-order ASM baseline proposed in Massaroli et al. (2020). It is clear that our OCP-theoretic method leads to substantially faster convergence, and the optimized $t_1$ stably hovers around $0.5$ without deviation (as appeared for the baseline). This drops the training time by nearly 20% compared to the vanilla training, where we fix $t_1$ to $1.0$, yet without sacrificing the test-time accuracy. A similar experiment for MNIST (see Fig. 13 in Appendix A.5) shows a consistent result. We highlight these improvements as the benefit gained from introducing the well-established OCP principle to these emerging deep continuous-time models.

**Comparison with recursive adjoint.** Finally, Fig. 11 reports the comparison between our SNOpt and the recursive adjoint baseline (see Section 2 and Table 1). It is clear that our method outperforms this second-order baseline by a large margin in both runtime efficiency and test-time performance. Note that we omit the comparison on CNF datasets since the recursive adjoint simply fails to converge.

**Remark** (Implicit regularization)**.** In some cases (*e.g.* SVHN in Fig. 8), our method may run slightly faster than first-order methods. This is a distinct phenomenon arising exclusively from training these continuous-time models. Since their forward and backward passes involve solving *parameterized* ODEs (see Fig. 2), the computation graphs are *parameter-dependent*; hence adaptive throughout training. In this vein, we conjecture that the preconditioned updates in these cases may have guided the parameter to regions that are numerically stabler (hence faster) for integration.[4] With this in mind, we report in Table 6 the value of Jacobian, $\int \|\nabla_{\mathbf{x}} F\|^2$, and Kinetic, $\int \|F\|^2$, regularization (Finlay et al., 2020) in SVHN training. Interestingly, the parameter found by our SNOpt indeed has a substantially lower value (hence stronger regularization and better-conditioned ODE dynamics) compared to the one found by Adam. This provides a plausible explanation of the reduction in the NFE when using our method, yet without hindering the test-time performance (see Table 4).

## 5    Conclusion

We present an efficient higher-order optimization framework for training Neural ODEs. Our method – named **SNOpt** – differs from existing second-order methods in various aspects. While it leverages similar factorization inherited in Kronecker-based methods (Martens & Grosse, 2015), the two methodologies differ fundamentally in that we construct analytic ODE expressions for higher-order derivatives (Theorem 1) and compute them through `ODESolve`. This retains the favorable $\mathcal{O}(1)$ memory as opposed to their $\mathcal{O}(T)$. It also enables a flexible rank-based factorization in Proposition 2. Meanwhile, our method extends the recent trend of OCP-inspired methods (Li et al., 2017; Liu et al., 2021b) to deep continuous-time models, yet using a rather straightforward framework without imposing additional assumptions, such as Markovian or game transformation. To summarize, our work advances several methodologies to the emerging deep continuous-time models, achieving strong empirical results and opening up new opportunities for analyzing models such as Neural SDEs/PDEs.

---

[4] In Appendix A.4, we provide some theoretical discussions (see Corollary 9) in this regard.

## Acknowledgments and Disclosure of Funding

The authors would like to thank Chia-Wen Kuo and Chen-Hsuan Lin for the meticulous proofreading, and Keuntaek Lee for providing additional computational resources. Guan-Horng Liu was supported by CPS NSF Award #1932068, and Tianrong Chen was supported by ARO Award #W911NF2010151.

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
