# A  Appendix

## A.1   Review of Optimal Control Programming (OCP) Perspective of Training Discrete DNNs and Continuous-time OCP

Here, we review the OCP perspective of training discrete DNNs and discuss how the continuous-time OCP can be connected to the training process of Neural ODEs. For a complete treatment, we refer readers to *e.g.* Weinan (2017); Li et al. (2017); Weinan et al. (2018); Liu & Theodorou (2019); Liu et al. (2021a), and their references therein.

Abuse the notation and let the layer propagation rule in standard feedforward DNNs with depth $T$ be

$$\boldsymbol{z}_{t+1} = f(\boldsymbol{z}_t, \boldsymbol{u}_t), \quad t \in \{0, 1, \cdots, T\}. \tag{17}$$

Here, $\boldsymbol{z}_t$ and $\boldsymbol{u}_t$ represent the (vectorized) hidden state and parameter of layer $t$. For instance, consider the propagation of a fully-connected layer, *i.e.* $\boldsymbol{z}_{t+1} = \sigma(\boldsymbol{W}_t \boldsymbol{z}_t + \boldsymbol{b}_t)$, where $\boldsymbol{W}_t$, $\boldsymbol{b}_t$, and $\sigma(\cdot)$ are respectively the weight, bias, and nonlinear activation function. Then, (17) treats $\boldsymbol{u}_t := \mathrm{vec}([\boldsymbol{W}_t, \boldsymbol{b}_t])$ as the vectorized parameter and $f$ as the composition of $\sigma(\cdot)$ and the affine transformation (Do not confuse with Fig. 3 which denotes $f$ as the affine transformation).

The OCP perspective notices that (17) can also be interpreted as a discrete-time dynamical system that propagates the state $\boldsymbol{z}_t$ with the control variable $\boldsymbol{u}_t$. In this vein, computing the forward pass of a DNN can be seen as propagating a nonlinear dynamical system from time $t = 0$ to $T$. Furthermore, the training process, *i.e.* finding optimal parameters $\{\boldsymbol{u}_t : \forall t\}$ for all layers, can be seen as a discrete-time Optimal Control Programming (OCP), which searches for an optimal control sequence $\{\boldsymbol{u}_t : \forall t\}$ that minimizes some objective.

In the case of Neural ODEs, the discrete-time layer propagation rule in (17) is replaced with the ODE in (1). However, as we have shown in Section 3.1, the interpretation between the trainable parameter $\theta$ and control variable (hence the connection between the training process and OCP) remains valid. In fact, consider the vanilla form of continuous-time OCP,

$$\min_{\mathbf{u}(t): t \in [t_0, t_1]} \left[ \Phi(\mathbf{x}_{t_1}) + \int_{t_0}^{t_1} \ell(t, \mathbf{x}_t, \mathbf{u}_t) \mathrm{d}t \right], \quad \dot{\mathbf{x}}_t = F(t, \mathbf{x}_t, \mathbf{u}_t), \quad \mathbf{x}_{t_0} = \boldsymbol{x}_{t_0}, \tag{18}$$

which resembles the one we used in (6) except considering a time-varying control process $\mathbf{u}(t)$. The necessary condition to the programming (18) can be characterized by the celebrated Pontryagin's maximum principle (Pontryagin et al., 1962).

**Theorem 3** (Pontryagin's maximum principle). *Let $\mathbf{u}_t^* \equiv \mathbf{u}^*(t)$ be a solution that achieved the minimum of (18). Then, there exists continuous processes, $\mathbf{x}_t^*$ and $\mathbf{a}_t^*$, such that*

$$\dot{\mathbf{x}}_t^* = \nabla_{\mathbf{a}} H(t, \mathbf{x}_t^*, \mathbf{a}_t^*, \mathbf{u}_t^*) \qquad\qquad \mathbf{x}_0^* = \mathbf{x}_0, \tag{19a}$$

$$\dot{\mathbf{a}}_t^* = -\nabla_{\mathbf{x}} H(t, \mathbf{x}_t^*, \mathbf{a}_t^*, \mathbf{u}_t^*), \qquad\qquad \mathbf{a}_{t_1}^* = \nabla_{\mathbf{x}} \Phi\left(\mathbf{x}_{t_1}^*\right), \tag{19b}$$

$$H(t, \mathbf{x}_t^*, \mathbf{a}_t^*, \mathbf{u}_t^*) \le H(t, \mathbf{x}_t^*, \mathbf{a}_t^*, \mathbf{u}_t), \qquad \forall \mathbf{u}_t \in \mathbb{R}^m, \quad t \in [t_0, t_1], \tag{19c}$$

*where the Hamiltonian function is defined as*

$$H(t, \mathbf{x}_t, \mathbf{a}_t, \mathbf{u}_t) := \mathbf{a}_t \cdot F(t, \mathbf{x}_t, \mathbf{u}_t) + \ell(t, \mathbf{x}_t, \mathbf{u}_t).$$

It can be readily verified that (19b) gives the same backward ODE in (4). In other words, the Adjoint Sensitivity Method used for deriving (3, 4) is a direct consequence arising from the OCP optimization theory. In this work, we provide a full treatment of continuous-time OCP theory and show that it opens up new algorithmic opportunities to higher-order training methods for Neural ODEs.

## A.2   Missing Derivations and Discussions in Section 3.1 and 3.2

**Proof of Theorem 1.** Rewrite the backward ODE of the accumulated loss $Q$ in (8) below

$$0 = \ell(t, \mathbf{x}_t, \mathbf{u}_t) + \frac{\mathrm{d}Q(t, \mathbf{x}_t, \mathbf{u}_t)}{\mathrm{d}t}, \quad Q(t_1, \mathbf{x}_{t_1}) = \Phi(\mathbf{x}_{t_1}).$$

Given a solution path $(\bar{\boldsymbol{x}}_t, \bar{\boldsymbol{u}}_t)$ of the ODEs in (6), define the differential state and control variables $(\delta \mathbf{x}_t, \delta \mathbf{u}_t)$ by

$$\delta \mathbf{x}_t := \mathbf{x}_t - \bar{\boldsymbol{x}}_t \quad \text{and} \quad \delta \mathbf{u}_t := \mathbf{u}_t - \bar{\boldsymbol{u}}_t.$$

We first perform second-order expansions for $\ell$ and $Q$ along the solution path, which are given by

$$\ell \approx \ell(t, \bar{\boldsymbol{x}}_t, \bar{\boldsymbol{u}}_t) + \ell_{\bar{\boldsymbol{x}}}^{\mathsf{T}} \delta\mathbf{x}_t + \ell_{\bar{\boldsymbol{u}}}^{\mathsf{T}} \delta\mathbf{u}_t + \frac{1}{2} \begin{bmatrix} \delta\mathbf{x}_t \\ \delta\mathbf{u}_t \end{bmatrix}^{\mathsf{T}} \begin{bmatrix} \ell_{\bar{\boldsymbol{x}}\bar{\boldsymbol{x}}} & \ell_{\bar{\boldsymbol{x}}\bar{\boldsymbol{u}}} \\ \ell_{\bar{\boldsymbol{u}}\bar{\boldsymbol{x}}} & \ell_{\bar{\boldsymbol{u}}\bar{\boldsymbol{u}}} \end{bmatrix} \begin{bmatrix} \delta\mathbf{x}_t \\ \delta\mathbf{u}_t \end{bmatrix}, \tag{20a}$$

$$Q \approx Q(t, \bar{\boldsymbol{x}}_t, \bar{\boldsymbol{u}}_t) + Q_{\bar{\boldsymbol{x}}}^{\mathsf{T}} \delta\mathbf{x}_t + Q_{\bar{\boldsymbol{u}}}^{\mathsf{T}} \delta\mathbf{u}_t + \frac{1}{2} \begin{bmatrix} \delta\mathbf{x}_t \\ \delta\mathbf{u}_t \end{bmatrix}^{\mathsf{T}} \begin{bmatrix} Q_{\bar{\boldsymbol{x}}\bar{\boldsymbol{x}}} & Q_{\bar{\boldsymbol{x}}\bar{\boldsymbol{u}}} \\ Q_{\bar{\boldsymbol{u}}\bar{\boldsymbol{x}}} & Q_{\bar{\boldsymbol{u}}\bar{\boldsymbol{u}}} \end{bmatrix} \begin{bmatrix} \delta\mathbf{x}_t \\ \delta\mathbf{u}_t \end{bmatrix}, \tag{20b}$$

where all derivatives, *i.e.* $\ell_{\bar{\boldsymbol{x}}}, \ell_{\bar{\boldsymbol{u}}}, Q_{\bar{\boldsymbol{x}}\bar{\boldsymbol{x}}}, Q_{\bar{\boldsymbol{u}}\bar{\boldsymbol{u}}}$, and etc, are time-varying. We can thereby obtain the time derivative of the second-order approximated $Q$ in (20b) following standard ordinary calculus.

$$\begin{aligned}
\frac{\mathrm{d}Q}{\mathrm{d}t} \approx & \frac{\mathrm{d}Q(t, \bar{\boldsymbol{x}}_t, \bar{\boldsymbol{u}}_t)}{\mathrm{d}t} + \left( \frac{\mathrm{d}Q_{\bar{\boldsymbol{x}}}}{\mathrm{d}t}^{\mathsf{T}} \delta\mathbf{x}_t + Q_{\bar{\boldsymbol{x}}}^{\mathsf{T}} \frac{\mathrm{d}\delta\mathbf{x}_t}{\mathrm{d}t} \right) + \left( \frac{\mathrm{d}Q_{\bar{\boldsymbol{u}}}}{\mathrm{d}t}^{\mathsf{T}} \delta\mathbf{u}_t + Q_{\bar{\boldsymbol{u}}}^{\mathsf{T}} \frac{\mathrm{d}\delta\mathbf{u}_t}{\mathrm{d}t} \right) \\
& + \frac{1}{2} \left( \delta\mathbf{x}_t^{\mathsf{T}} \frac{\mathrm{d}Q_{\bar{\boldsymbol{x}}\bar{\boldsymbol{x}}}}{\mathrm{d}t} \delta\mathbf{x}_t + \frac{\mathrm{d}\delta\mathbf{x}_t}{\mathrm{d}t}^{\mathsf{T}} Q_{\bar{\boldsymbol{x}}\bar{\boldsymbol{x}}} \delta\mathbf{x}_t + \delta\mathbf{x}_t^{\mathsf{T}} Q_{\bar{\boldsymbol{x}}\bar{\boldsymbol{x}}} \frac{\mathrm{d}\delta\mathbf{x}_t}{\mathrm{d}t} \right) \\
& + \frac{1}{2} \left( \delta\mathbf{u}_t^{\mathsf{T}} \frac{\mathrm{d}Q_{\bar{\boldsymbol{u}}\bar{\boldsymbol{u}}}}{\mathrm{d}t} \delta\mathbf{u}_t + \frac{\mathrm{d}\delta\mathbf{u}_t}{\mathrm{d}t}^{\mathsf{T}} Q_{\bar{\boldsymbol{u}}\bar{\boldsymbol{u}}} \delta\mathbf{u}_t + \delta\mathbf{u}_t^{\mathsf{T}} Q_{\bar{\boldsymbol{u}}\bar{\boldsymbol{u}}} \frac{\mathrm{d}\delta\mathbf{u}_t}{\mathrm{d}t} \right) \\
& + \frac{1}{2} \left( \delta\mathbf{x}_t^{\mathsf{T}} \frac{\mathrm{d}Q_{\bar{\boldsymbol{x}}\bar{\boldsymbol{u}}}}{\mathrm{d}t} \delta\mathbf{u}_t + \frac{\mathrm{d}\delta\mathbf{x}_t}{\mathrm{d}t}^{\mathsf{T}} Q_{\bar{\boldsymbol{x}}\bar{\boldsymbol{u}}} \delta\mathbf{u}_t + \delta\mathbf{x}_t^{\mathsf{T}} Q_{\bar{\boldsymbol{x}}\bar{\boldsymbol{u}}} \frac{\mathrm{d}\delta\mathbf{u}_t}{\mathrm{d}t} \right) \\
& + \frac{1}{2} \left( \delta\mathbf{u}_t^{\mathsf{T}} \frac{\mathrm{d}Q_{\bar{\boldsymbol{u}}\bar{\boldsymbol{x}}}}{\mathrm{d}t} \delta\mathbf{x}_t + \frac{\mathrm{d}\delta\mathbf{u}_t}{\mathrm{d}t}^{\mathsf{T}} Q_{\bar{\boldsymbol{u}}\bar{\boldsymbol{x}}} \delta\mathbf{x}_t + \delta\mathbf{u}_t^{\mathsf{T}} Q_{\bar{\boldsymbol{u}}\bar{\boldsymbol{x}}} \frac{\mathrm{d}\delta\mathbf{x}_t}{\mathrm{d}t} \right).
\end{aligned} \tag{21}$$

Next, we need to compute $\frac{\mathrm{d}\delta\mathbf{x}_t}{\mathrm{d}t}$ and $\frac{\mathrm{d}\delta\mathbf{u}_t}{\mathrm{d}t}$, *i.e.* the dynamics of the differential state and control. This can be achieved by linearizing the ODE dynamics along $(\bar{\boldsymbol{x}}_t, \bar{\boldsymbol{u}}_t)$.

$$\frac{\mathrm{d}}{\mathrm{d}t}(\bar{\boldsymbol{x}}_t + \delta\mathbf{x}_t) = F(t, \bar{\boldsymbol{x}}_t, \bar{\boldsymbol{u}}_t) + F_{\bar{\boldsymbol{x}}}(t)^{\mathsf{T}} \delta\mathbf{x}_t + F_{\bar{\boldsymbol{u}}}(t)^{\mathsf{T}} \delta\mathbf{u}_t \Rightarrow \frac{\mathrm{d}\delta\mathbf{x}_t}{\mathrm{d}t} = F_{\bar{\boldsymbol{x}}}(t)^{\mathsf{T}} \delta\mathbf{x}_t + F_{\bar{\boldsymbol{u}}}(t)^{\mathsf{T}} \delta\mathbf{u}_t,$$

$$\frac{\mathrm{d}}{\mathrm{d}t}(\bar{\boldsymbol{u}}_t + \delta\mathbf{u}_t) = \mathbf{0} \Rightarrow \frac{\mathrm{d}\delta\mathbf{u}_t}{\mathrm{d}t} = \mathbf{0}, \tag{22}$$

since $\frac{\mathrm{d}\bar{\boldsymbol{x}}_t}{\mathrm{d}t} = F(t, \bar{\boldsymbol{x}}_t, \bar{\boldsymbol{u}}_t)$. Finally, substituting (20a) and (21) back to (8) and replacing all $\left( \frac{\mathrm{d}\delta\mathbf{x}_t}{\mathrm{d}t}, \frac{\mathrm{d}\delta\mathbf{u}_t}{\mathrm{d}t} \right)$ with (22) yield the following set of backward ODEs.

$$\begin{aligned}
-\frac{\mathrm{d}Q_{\bar{\boldsymbol{x}}}}{\mathrm{d}t} &= \ell_{\bar{\boldsymbol{x}}} + F_{\bar{\boldsymbol{x}}}^{\mathsf{T}} Q_{\bar{\boldsymbol{x}}}, & -\frac{\mathrm{d}Q_{\bar{\boldsymbol{u}}}}{\mathrm{d}t} &= \ell_{\bar{\boldsymbol{u}}} + F_{\bar{\boldsymbol{u}}}^{\mathsf{T}} Q_{\bar{\boldsymbol{x}}}, \\
-\frac{\mathrm{d}Q_{\bar{\boldsymbol{x}}\bar{\boldsymbol{x}}}}{\mathrm{d}t} &= \ell_{\bar{\boldsymbol{x}}\bar{\boldsymbol{x}}} + F_{\bar{\boldsymbol{x}}}^{\mathsf{T}} Q_{\bar{\boldsymbol{x}}\bar{\boldsymbol{x}}} + Q_{\bar{\boldsymbol{x}}\bar{\boldsymbol{x}}} F_{\bar{\boldsymbol{x}}}, & -\frac{\mathrm{d}Q_{\bar{\boldsymbol{x}}\bar{\boldsymbol{u}}}}{\mathrm{d}t} &= \ell_{\bar{\boldsymbol{x}}\bar{\boldsymbol{u}}} + Q_{\bar{\boldsymbol{x}}\bar{\boldsymbol{x}}} F_{\bar{\boldsymbol{u}}} + F_{\bar{\boldsymbol{x}}}^{\mathsf{T}} Q_{\bar{\boldsymbol{x}}\bar{\boldsymbol{u}}}, \\
-\frac{\mathrm{d}Q_{\bar{\boldsymbol{u}}\bar{\boldsymbol{u}}}}{\mathrm{d}t} &= \ell_{\bar{\boldsymbol{u}}\bar{\boldsymbol{u}}} + F_{\bar{\boldsymbol{u}}}^{\mathsf{T}} Q_{\bar{\boldsymbol{x}}\bar{\boldsymbol{u}}} + Q_{\bar{\boldsymbol{u}}\bar{\boldsymbol{x}}} F_{\bar{\boldsymbol{u}}}, & -\frac{\mathrm{d}Q_{\bar{\boldsymbol{u}}\bar{\boldsymbol{x}}}}{\mathrm{d}t} &= \ell_{\bar{\boldsymbol{u}}\bar{\boldsymbol{x}}} + F_{\bar{\boldsymbol{u}}}^{\mathsf{T}} Q_{\bar{\boldsymbol{x}}\bar{\boldsymbol{x}}} + Q_{\bar{\boldsymbol{u}}\bar{\boldsymbol{x}}} F_{\bar{\boldsymbol{x}}}.
\end{aligned}$$

$\square$

**Remark 4** (Relation to continuous-time OCP algorithm). The proof of Theorem 1 resembles standard derivation of continuous-time Differential Dynamic Programming (DDP), a second-order OCP method that has shown great successes in modern autonomous systems (Tassa et al., 2014). However, our derivation was modified accordingly to account for the particular OCP proposed in (6), which concerns only the initial condition of the time-invariant control. As this equivalently leaves out the "dynamic" aspect of DDP, we shorthand our methodology by *Differential Programming*.

**Remark 5** (Computing higher-order derivatives). The proof of Theorem 1 can be summarized by

*Step 1*. Expand $Q$ and $\ell$ up to second-order, *i.e.* (20).

*Step 2*. Derive the dynamics of differential variables. In our case, we consider the linear ODE presented in (22).

*Step 3*. Substitute the approximations from Step 1 and 2 back to (8), expand all terms using ordinary calculus (21), then collect the dynamics of each derivative.

For higher-order derivatives, we simply need to consider a higher-order expansion of $Q$ and $\ell$ in Step 1 (see *e.g.* Almubarak et al. (2019) and their reference therein). It is also possible to consider higher-order expression of the linear differential ODEs in Step 2, which may further improve the convergence at the cost of extra overhead (Theodorou et al., 2010).

**Remark 6** (Complexity of Remark 5). Let $k$ be the optimization order. Development of higher-order ($k \geq 3$) optimization based on Theorem 1 certainly has few computational obstacles, just like what we have identified and resolved in the case of $k = 2$ (see Section 3.2). In terms of memory, while the number of backward ODEs suggested by Theorem 1 can grow exponentially w.r.t. $k$, Kelly et al. (2020) has developed an efficient truncated method that reduces the number to $\mathcal{O}(k^2)$ or $\mathcal{O}(k \log k)$. In terms of runtime, analogous to the Kronecker approximation that we use to factorize second-order matrices, Gupta et al. (2018) provided an extension to generic higher-order tensor programming. Hence, it may still be plausible to avoid impractical training.

**Proof of Proposition 2.** We will proceed the proof by induction. Recall that when $\ell$ degenerates, the matrix ODEs presented in (9b, 9c) from Theorem 1 take the form,

$$-\frac{\mathrm{d}Q_{\bar{x}\bar{x}}}{\mathrm{d}t} = F_{\bar{x}}^{\mathsf{T}}Q_{\bar{x}\bar{x}} + Q_{\bar{x}\bar{x}}F_{\bar{x}}, \qquad Q_{\bar{x}\bar{x}}(t_1) = \Phi_{\bar{x}\bar{x}}, \tag{24a}$$

$$-\frac{\mathrm{d}Q_{\bar{u}\bar{u}}}{\mathrm{d}t} = F_{\bar{u}}^{\mathsf{T}}Q_{\bar{x}\bar{u}} + Q_{\bar{u}\bar{x}}F_{\bar{u}}, \qquad Q_{\bar{u}\bar{u}}(t_1) = \mathbf{0}, \tag{24b}$$

$$-\frac{\mathrm{d}Q_{\bar{x}\bar{u}}}{\mathrm{d}t} = Q_{\bar{x}\bar{x}}F_{\bar{u}} + F_{\bar{x}}^{\mathsf{T}}Q_{\bar{x}\bar{u}}, \qquad Q_{\bar{x}\bar{u}}(t_1) = \mathbf{0}, \tag{24c}$$

where we leave out the ODE of $Q_{\bar{u}\bar{x}}$ since $Q_{\bar{u}\bar{x}}(t) = Q_{\bar{x}\bar{u}}^{\mathsf{T}}(t)$ for all $t \in [t_0, t_1]$.

From (24), it is obvious that the decomposition given in Proposition 2 holds at the terminal stage $t_1$. Now, suppose it also holds at $t \in (t_0, t_1)$, then the backward dynamics of second-order matrices at this specific time step $t$, take $\frac{\mathrm{d}Q_{\bar{x}\bar{x}}(t)}{\mathrm{d}t}$ for instance, become

$$
\begin{aligned}
-\frac{\mathrm{d}Q_{\bar{x}\bar{x}}}{\mathrm{d}t} &= F_{\bar{x}}^{\mathsf{T}}Q_{\bar{x}\bar{x}} + Q_{\bar{x}\bar{x}}F_{\bar{x}} \\
&= F_{\bar{x}}^{\mathsf{T}}\left(\sum_{i=1}^{R} \mathbf{q}_i \otimes \mathbf{q}_i\right) + \left(\sum_{i=1}^{R} \mathbf{q}_i \otimes \mathbf{q}_i\right) F_{\bar{x}} \\
&= \sum_{i=1}^{R} \left[\left(F_{\bar{x}}^{\mathsf{T}}\mathbf{q}_i\right) \otimes \mathbf{q}_i + \mathbf{q}_i \otimes \left(F_{\bar{x}}^{\mathsf{T}}\mathbf{q}_i\right)\right],
\end{aligned} \tag{25}
$$

where $\mathbf{q}_i \equiv \mathbf{q}_i(t)$ for brevity. On the other hand, the LHS of (25) can be expanded as

$$-\frac{\mathrm{d}Q_{\bar{x}\bar{x}}}{\mathrm{d}t} = -\frac{\mathrm{d}}{\mathrm{d}t}\left(\sum_{i=1}^{R} \mathbf{q}_i \otimes \mathbf{q}_i\right) = -\sum_{i=1}^{R}\left[\frac{\mathrm{d}\mathbf{q}_i}{\mathrm{d}t} \otimes \mathbf{q}_i + \mathbf{q}_i \otimes \frac{\mathrm{d}\mathbf{q}_i}{\mathrm{d}t}\right], \tag{26}$$

which follows by standard ordinary calculus. Equating (25) and (26) implies that following relation should hold at time $t$,

$$-\frac{\mathrm{d}\mathbf{q}_i}{\mathrm{d}t} = F_{\bar{x}}^{\mathsf{T}}\mathbf{q}_i,$$

which yields the first ODE appeared in (11). Similarly, we can repeat the same process (25, 26) for the matrices $Q_{\bar{x}\bar{u}}$ and $Q_{\bar{u}\bar{u}}$. This will give us

$$
\begin{aligned}
-\frac{\mathrm{d}Q_{\bar{u}\bar{u}}}{\mathrm{d}t} &= F_{\bar{u}}^{\mathsf{T}}Q_{\bar{x}\bar{u}} + Q_{\bar{u}\bar{x}}F_{\bar{u}} \\
&\Rightarrow -\sum_{i=1}^{R}\left[\frac{\mathrm{d}\mathbf{p}_i}{\mathrm{d}t} \otimes \mathbf{p}_i + \mathbf{p}_i \otimes \frac{\mathrm{d}\mathbf{p}_i}{\mathrm{d}t}\right] = \sum_{i=1}^{R}\left[\left(F_{\bar{u}}^{\mathsf{T}}\mathbf{q}_i\right) \otimes \mathbf{p}_i + \mathbf{p}_i \otimes \left(F_{\bar{u}}^{\mathsf{T}}\mathbf{q}_i\right)\right]
\end{aligned}
$$

$$
\begin{aligned}
-\frac{\mathrm{d}Q_{\bar{x}\bar{u}}}{\mathrm{d}t} &= Q_{\bar{x}\bar{x}}F_{\bar{u}} + F_{\bar{x}}^{\mathsf{T}}Q_{\bar{x}\bar{u}} \\
&\Rightarrow -\sum_{i=1}^{R}\left[\frac{\mathrm{d}\mathbf{q}_i}{\mathrm{d}t} \otimes \mathbf{p}_i + \mathbf{q}_i \otimes \frac{\mathrm{d}\mathbf{p}_i}{\mathrm{d}t}\right] = \sum_{i=1}^{R}\left[\left(F_{\bar{x}}^{\mathsf{T}}\mathbf{q}_i\right) \otimes \mathbf{p}_i + \mathbf{q}_i \otimes \left(F_{\bar{u}}^{\mathsf{T}}\mathbf{q}_i\right)\right],
\end{aligned}
$$

which implies that following relation should also hold at time $t$,

$$-\frac{\mathrm{d}\mathbf{p}_i}{\mathrm{d}t} = F_{\bar{\boldsymbol{u}}}^\mathsf{T}\mathbf{q}_i.$$

Hence, we conclude the proof. $\qquad\square$

**Derivation and approximation in (13, 14).** We first recall two formulas related to the Kronecker product that will be shown useful in deriving (13, 14).

$$(\boldsymbol{A}\otimes\boldsymbol{B})(\boldsymbol{C}\otimes\boldsymbol{D})^\mathsf{T} = \boldsymbol{A}\boldsymbol{C}^\mathsf{T}\otimes\boldsymbol{B}\boldsymbol{D}^\mathsf{T}, \tag{27}$$

$$(\boldsymbol{A}\otimes\boldsymbol{B})^{-1}\mathrm{vec}(\boldsymbol{W}) = \mathrm{vec}(\boldsymbol{B}^{-1}\boldsymbol{W}\boldsymbol{A}^{-\mathsf{T}}), \tag{28}$$

where $\boldsymbol{W}\in\mathbb{R}^{l\times p}$, $\boldsymbol{A},\boldsymbol{C}\in\mathbb{R}^{p\times p}$, and $\boldsymbol{B},\boldsymbol{D}\in\mathbb{R}^{l\times l}$. Further, $\boldsymbol{A},\boldsymbol{B}$ are invertible.

Now, we provide a step-by-step derivation of (13). For brevity, we will denote $\mathbf{g}_i^n \equiv \frac{\partial F}{\partial\mathbf{h}^n}^\mathsf{T}\mathbf{q}_i$.

$$
\begin{aligned}
\mathcal{L}_{\theta^n\theta^n} \equiv Q_{\bar{\boldsymbol{u}}^n\bar{\boldsymbol{u}}^n}(t_0) &= \sum_{i=1}^R \left(\int_{t_1}^{t_0}(\mathbf{z}^n\otimes\mathbf{g}_i^n)\,\mathrm{d}t\right)\left(\int_{t_1}^{t_0}(\mathbf{z}^n\otimes\mathbf{g}_i^n)\,\mathrm{d}t\right)^\mathsf{T} \\
&\approx \sum_{i=1}^R \int_{t_1}^{t_0}(\mathbf{z}^n\otimes\mathbf{g}_i^n)(\mathbf{z}^n\otimes\mathbf{g}_i^n)^\mathsf{T}\,\mathrm{d}t \\
&= \sum_{i=1}^R \int_{t_1}^{t_0}(\mathbf{z}^n\mathbf{z}^{n\mathsf{T}})\otimes(\mathbf{g}_i^n\mathbf{g}_i^{n\mathsf{T}})\,\mathrm{d}t \qquad\text{by (27)} \\
&\approx \sum_{i=1}^R \int_{t_1}^{t_0}(\mathbf{z}^n\mathbf{z}^{n\mathsf{T}})\,\mathrm{d}t \otimes \int_{t_1}^{t_0}(\mathbf{g}_i^n\mathbf{g}_i^{n\mathsf{T}})\,\mathrm{d}t \\
&= \int_{t_1}^{t_0}(\mathbf{z}^n\otimes\mathbf{z}^n)\,\mathrm{d}t \otimes \int_{t_1}^{t_0}\sum_{i=1}^R \mathbf{g}_i^n\otimes\mathbf{g}_i^n\,\mathrm{d}t. \qquad\text{by Fubini's Theorem}
\end{aligned}
$$

There are two approximations in the above derivation. The first one assumes that the contributions of the quantity "$\mathbf{z}^n(t)\otimes\mathbf{g}_i^n(t)$" are uncorrelated across time, whereas the second one assumes $\mathbf{z}^n$ and $\mathbf{g}_i^n$ are pair-wise independents. We stress that both are widely adopted assumptions for deriving practical Kronecker-based methods (Grosse & Martens, 2016; Martens et al., 2018). While the first assumption can be rather strong, the second approximation has been verified in some empirical study (Wu et al., 2020) and can be made exact under certain conditions (Martens & Grosse, 2015). Finally, (14) follows readily from (28) by noticing that $\mathcal{L}_{\theta^n\theta^n} = \bar{\boldsymbol{A}}_n\otimes\bar{\boldsymbol{B}}_n$ under our computation.

**Remark 7** (Uncorrelated assumption of $\mathbf{z}^n\otimes\mathbf{g}_i^n$)**.** This assumption is indeed strong yet almost necessary to yield tractable Kronecker matrices for efficient second-order operation. Tracing back to the development of Kronecker-based methods, similar assumptions also appear in convolution layers (*e.g.* uncorrelated between spatial-wise derivatives (Grosse & Martens, 2016)) and recurrent units (*e.g.* uncorrelated between temporal-wise derivatives (Martens et al., 2018)). The latter may be thought of as the discretization of Neural ODEs. We note, however, that it is possible to relax this assumption by considering tractable graphical models (*e.g.* linear Gaussian (Martens et al., 2018)) at the cost of 2-3 times more operations per iteration. In terms of the performance difference, perhaps surprisingly, adopting tractable temporal models provides only minor improvement in test-time performance (see Fig. 4 in Martens et al. (2018)). In some cases, it has been empirically observed that methods adopting the uncorrelated assumption yields better performance (Laurent et al., 2018).

**Remark 8** (Relation to Fisher Information Matrix)**.** Recall that for all experiments we apply Gaussian-Newton approximation to the terminal Hessian $Q_{\bar{\boldsymbol{x}}\bar{\boldsymbol{x}}}(t_1)$. This specific choice is partially based on empirical performance and computational purpose, yet it turns out that the resulting precondition matrices (12, 13) can be interpreted as Fisher information matrix (FIM). In other words, under this specific setup, (12, 13) can be equivalently viewed as the FIM of Neural ODEs. This implies SNOpt may be thought of as following Natural Gradient Descent (NGD), which is well-known for taking the steepest descent direction in the space of model distributions (Amari & Nagaoka, 2000; Martens, 2014). Indeed, it has been observed that NGD-based methods converge to equally good accuracies, even though its learning rate varies across 1-2 orders (see Fig 10 in Ma et al. (2019) and Fig 4 in George et al. (2018)). These observations coincide with our results (Fig. 9) for Neural ODEs.

## A.3 Discussion on the Free-Horizon Optimization in Section 3.4

**Derivation of (16).** Here we present an extension of our OCP framework to jointly optimizing the architecture of Neural ODEs, specifically the integration bound $t_1$. The proceeding derivation, despite being rather tedious, follows a similar procedure in Section 3.1 and the proof of Theorem 1.

Recall the modified cost-to-go function that we consider for free-horizon optimization,

$$\widetilde{Q}(t, \mathbf{x}_t, \mathbf{u}_t, \mathrm{T}) := \widetilde{\Phi}(\mathrm{T}, \mathbf{x}(\mathrm{T})) + \int_t^\mathrm{T} \ell(\tau, \mathbf{x}_\tau, \mathbf{u}_\tau) \, \mathrm{d}\tau,$$

where we introduce a new variable, *i.e.* the terminal horizon $\mathrm{T}$, that shall be jointly optimized. We use the expression $\mathbf{x}(\mathrm{T})$ to highlight the fact that the terminal state is now a function of $\mathrm{T}$.

Similar to what we have explored in Section 3.1, our goal is to derive an analytic expression for the derivatives of $\widetilde{Q}$ at the integration start time $t_0$ w.r.t. this new variable $\mathrm{T}$. This can be achieved by characterizing the local behavior of the following ODE,

$$0 = \ell(t, \mathbf{x}_t, \mathbf{u}_t) + \frac{\mathrm{d}\widetilde{Q}(t, \mathbf{x}_t, \mathbf{u}_t, \mathrm{T})}{\mathrm{d}t}, \quad \widetilde{Q}(\mathrm{T}, \mathbf{x}_\mathrm{T}) = \widetilde{\Phi}(\mathrm{T}, \mathbf{x}(\mathrm{T})), \tag{29}$$

expanded on some nominal solution path $(\bar{\boldsymbol{x}}_t, \bar{\boldsymbol{u}}_t, \bar{T})$.

Let us start from the terminal condition in (29). Given $\widetilde{Q}(\bar{T}, \bar{\boldsymbol{x}}_{\bar{T}}) = \widetilde{\Phi}(\bar{T}, \bar{\boldsymbol{x}}(\bar{T}))$, perturbing the terminal horizon $\bar{T}$ by an infinitesimal amount $\delta\mathrm{T}$ yields

$$\widetilde{Q}(\bar{T} + \delta\mathrm{T}, \bar{\boldsymbol{x}}_{\bar{T}+\delta\mathrm{T}}) = \ell(\bar{\boldsymbol{x}}_{\bar{T}}, \bar{\boldsymbol{u}}_{\bar{T}})\delta\mathrm{T} + \widetilde{\Phi}(\bar{T} + \delta\mathrm{T}, \bar{\boldsymbol{x}}(\bar{T} + \delta\mathrm{T})). \tag{30}$$

It can be shown that the second-order expansion of the last term in (30) takes the form,

$$
\begin{aligned}
\widetilde{\Phi}\left(\bar{T} + \delta\mathrm{T}, \bar{\boldsymbol{x}}(\bar{T} + \delta\mathrm{T})\right) \approx {} & \widetilde{\Phi}\left(\bar{T}, \bar{\boldsymbol{x}}(\bar{T})\right) + \widetilde{\Phi}_{\bar{\boldsymbol{x}}}^\mathsf{T} \delta\mathbf{x}_{\bar{T}} + \left(\widetilde{\Phi}_{\bar{T}} + \widetilde{\Phi}_{\bar{\boldsymbol{x}}}^\mathsf{T} \bar{F}\right) \delta\mathrm{T} + \frac{1}{2}\delta\mathbf{x}_{\bar{T}}^\mathsf{T} \widetilde{\Phi}_{\bar{\boldsymbol{x}}\bar{\boldsymbol{x}}} \delta\mathbf{x}_{\bar{T}} \\
& + \frac{1}{2}\delta\mathbf{x}_{\bar{T}}^\mathsf{T} \left(\widetilde{\Phi}_{\bar{\boldsymbol{x}}\bar{T}} + \widetilde{\Phi}_{\bar{\boldsymbol{x}}\bar{\boldsymbol{x}}}\bar{F}\right) \delta\mathrm{T} + \frac{1}{2}\delta\mathrm{T}\left(\widetilde{\Phi}_{\bar{T}\bar{\boldsymbol{x}}} + \bar{F}^\mathsf{T}\widetilde{\Phi}_{\bar{\boldsymbol{x}}\bar{\boldsymbol{x}}}\right) \delta\mathbf{x}_{\bar{T}} \\
& + \frac{1}{2}\delta\mathrm{T}\left(\widetilde{\Phi}_{\bar{T}\bar{T}} + \widetilde{\Phi}_{\bar{T}\bar{\boldsymbol{x}}}\bar{F} + \bar{F}^\mathsf{T}\widetilde{\Phi}_{\bar{\boldsymbol{x}}\bar{T}} + \bar{F}^\mathsf{T}\widetilde{\Phi}_{\bar{\boldsymbol{x}}\bar{\boldsymbol{x}}}\bar{F}\right) \delta\mathrm{T},
\end{aligned}
\tag{31}
$$

which relies on the fact that the following formula holds for any generic function that takes $t$ and $\mathbf{x}(t)$ as its arguments:

$$\frac{\mathrm{d}}{\mathrm{d}t}(\cdot) = \frac{\partial}{\partial t}(\cdot) + \frac{\partial}{\partial \mathbf{x}}(\cdot)^\mathsf{T} \bar{F}, \quad \text{where } \bar{F} = F(t, \bar{\boldsymbol{x}}_t, \bar{\boldsymbol{u}}_t).$$

Substituting (31) to (30) gives us the local expressions of the terminal condition up to second-order,

$$\widetilde{Q}_{\bar{\boldsymbol{x}}}(\bar{T}) = \widetilde{\Phi}_{\bar{\boldsymbol{x}}}, \qquad\qquad \widetilde{Q}_{\bar{T}}(\bar{T}) = \ell(\bar{\boldsymbol{x}}_{\bar{T}}, \bar{\boldsymbol{u}}_{\bar{T}}) + \widetilde{\Phi}_{\bar{T}} + \widetilde{\Phi}_{\bar{\boldsymbol{x}}}^\mathsf{T}\bar{F}, \tag{32a}$$

$$\widetilde{Q}_{\bar{T}\bar{\boldsymbol{x}}}(\bar{T}) = \widetilde{\Phi}_{\bar{T}\bar{\boldsymbol{x}}} + \bar{F}^\mathsf{T}\widetilde{\Phi}_{\bar{\boldsymbol{x}}\bar{\boldsymbol{x}}}, \quad \widetilde{Q}_{\bar{\boldsymbol{x}}\bar{T}}(\bar{T}) = \widetilde{\Phi}_{\bar{\boldsymbol{x}}\bar{T}} + \widetilde{\Phi}_{\bar{\boldsymbol{x}}\bar{\boldsymbol{x}}}\bar{F}, \tag{32b}$$

$$\widetilde{Q}_{\bar{\boldsymbol{x}}\bar{\boldsymbol{x}}}(\bar{T}) = \widetilde{\Phi}_{\bar{\boldsymbol{x}}\bar{\boldsymbol{x}}}, \qquad\qquad \widetilde{Q}_{\bar{T}\bar{T}}(\bar{T}) = \widetilde{\Phi}_{\bar{T}\bar{T}} + \widetilde{\Phi}_{\bar{T}\bar{\boldsymbol{x}}}^\mathsf{T}\bar{F} + \bar{F}^\mathsf{T}\widetilde{\Phi}_{\bar{\boldsymbol{x}}\bar{T}} + \bar{F}^\mathsf{T}\widetilde{\Phi}_{\bar{\boldsymbol{x}}\bar{\boldsymbol{x}}}\bar{F}, \tag{32c}$$

where $\widetilde{Q}_{\bar{\boldsymbol{x}}}(\bar{T}) \equiv \frac{\delta\widetilde{Q}}{\delta\mathbf{x}_{\bar{T}}} = \frac{\widetilde{Q}(\bar{T}+\delta\mathrm{T}, \bar{\boldsymbol{x}}_{\bar{T}+\delta\mathrm{T}}) - \widetilde{Q}(\bar{T}, \bar{\boldsymbol{x}}_{\bar{T}})}{\delta\mathbf{x}_{\bar{T}}}$, and etc.

Next, consider the ODE dynamics in (29). Similar to (20b), we can expand $\widetilde{Q}$ w.r.t. all optimizing variables, *i.e.* $(\mathbf{x}_t, \mathbf{u}_t, \mathrm{T})$, up to second-order. In this case, the approximation is given by

$$\widetilde{Q}(t, \bar{\boldsymbol{x}}_t, \bar{\boldsymbol{u}}_t, \bar{T}) + \widetilde{Q}_{\bar{\boldsymbol{x}}}^\mathsf{T}\delta\mathbf{x}_t + \widetilde{Q}_{\bar{\boldsymbol{u}}}^\mathsf{T}\delta\mathbf{u}_t + \widetilde{Q}_{\bar{T}}\delta\mathrm{T} + \frac{1}{2}\begin{bmatrix}\delta\mathbf{x}_t \\ \delta\mathbf{u}_t \\ \delta\mathrm{T}\end{bmatrix}^\mathsf{T} \begin{bmatrix} \widetilde{Q}_{\bar{\boldsymbol{x}}\bar{\boldsymbol{x}}} & \widetilde{Q}_{\bar{\boldsymbol{x}}\bar{\boldsymbol{u}}} & \widetilde{Q}_{\bar{\boldsymbol{x}}\bar{T}} \\ \widetilde{Q}_{\bar{\boldsymbol{u}}\bar{\boldsymbol{x}}} & \widetilde{Q}_{\bar{\boldsymbol{u}}\bar{\boldsymbol{u}}} & \widetilde{Q}_{\bar{\boldsymbol{u}}\bar{T}} \\ \widetilde{Q}_{\bar{T}\bar{\boldsymbol{x}}} & \widetilde{Q}_{\bar{T}\bar{\boldsymbol{u}}} & \widetilde{Q}_{\bar{T}\bar{T}} \end{bmatrix}\begin{bmatrix}\delta\mathbf{x}_t \\ \delta\mathbf{u}_t \\ \delta\mathrm{T}\end{bmatrix}, \tag{33}$$

which shares the same form as (20b) except having additional terms that account for the derivatives related to $\mathrm{T}$ (marked as green). Substitute (33) to the ODE dynamics in (29), then expand the time derivatives $\frac{\mathrm{d}}{\mathrm{d}t}$ as in (21), and finally replace $\frac{\mathrm{d}\delta\mathbf{x}_t}{\mathrm{d}t}$, $\frac{\mathrm{d}\delta\mathbf{u}_t}{\mathrm{d}t}$, and $\frac{\mathrm{d}\delta\mathrm{T}}{\mathrm{d}t}$ with

$$\frac{\mathrm{d}\delta\mathbf{x}_t}{\mathrm{d}t} = F_{\bar{\boldsymbol{x}}}^\mathsf{T}\delta\mathbf{x}_t + F_{\bar{\boldsymbol{u}}}^\mathsf{T}\delta\mathbf{u}_t, \quad \frac{\mathrm{d}\delta\mathbf{u}_t}{\mathrm{d}t} = \mathbf{0}, \quad \text{and} \quad \frac{\mathrm{d}\delta\mathrm{T}}{\mathrm{d}t} = 0.$$

Then, it can be shown that the first and second-order derivatives of $\widetilde{Q}$ w.r.t. T obey the following backward ODEs:

$$-\frac{\mathrm{d}\widetilde{Q}_{\bar{T}}}{\mathrm{d}t} = 0, \quad -\frac{\mathrm{d}\widetilde{Q}_{\bar{T}\bar{T}}}{\mathrm{d}t} = 0, \quad -\frac{\mathrm{d}\widetilde{Q}_{\bar{T}\bar{x}}}{\mathrm{d}t} = \widetilde{Q}_{\bar{T}\bar{x}}F_{\bar{x}}, \quad -\frac{\mathrm{d}\widetilde{Q}_{\bar{T}\bar{u}}}{\mathrm{d}t} = \widetilde{Q}_{\bar{T}\bar{x}}F_{\bar{u}},$$

with the terminal condition given by (32). As for the derivatives that do not involve T, *e.g.* $\widetilde{Q}_{\bar{x}\bar{x}}$ and $\widetilde{Q}_{\bar{u}\bar{u}}$, one can verify that they follow the same backward structures given in (9) except changing the terminal condition from $\Phi$ to $\widetilde{\Phi}$.

To summarize, solving the following ODEs gives us the derivatives of $\widetilde{Q}$ related to T at $t_0$:

$$-\frac{\mathrm{d}}{\mathrm{d}t}\widetilde{Q}_{\bar{T}}(t) = 0, \qquad \widetilde{Q}_{\bar{T}}(\bar{T}) = \ell(\bar{x}_{\bar{T}}, \bar{u}_{\bar{T}}) + \widetilde{\Phi}_{\bar{T}} + \widetilde{\Phi}_{\bar{x}}^{\mathsf{T}}\bar{F} \tag{34a}$$

$$-\frac{\mathrm{d}}{\mathrm{d}t}\widetilde{Q}_{\bar{T}\bar{T}}(t) = 0, \qquad \widetilde{Q}_{\bar{T}\bar{T}}(\bar{T}) = \widetilde{\Phi}_{\bar{T}\bar{T}} + \widetilde{\Phi}_{\bar{T}\bar{x}}^{\mathsf{T}}\bar{F} + \bar{F}^{\mathsf{T}}\widetilde{\Phi}_{\bar{x}\bar{T}} + \bar{F}^{\mathsf{T}}\widetilde{\Phi}_{\bar{x}\bar{x}}\bar{F} \tag{34b}$$

$$-\frac{\mathrm{d}}{\mathrm{d}t}\widetilde{Q}_{\bar{T}\bar{x}}(t) = \widetilde{Q}_{\bar{T}\bar{x}}F_{\bar{x}}, \quad \widetilde{Q}_{\bar{T}\bar{x}}(\bar{T}) = \widetilde{\Phi}_{\bar{T}\bar{x}} + \bar{F}^{\mathsf{T}}\widetilde{\Phi}_{\bar{x}\bar{x}} \tag{34c}$$

$$-\frac{\mathrm{d}}{\mathrm{d}t}\widetilde{Q}_{\bar{T}\bar{u}}(t) = \widetilde{Q}_{\bar{T}\bar{x}}F_{\bar{u}}, \quad \widetilde{Q}_{\bar{T}\bar{u}}(\bar{T}) = \mathbf{0} \tag{34d}$$

Then, we can consider the following quadratic programming for the optimal perturbation $\delta\mathrm{T}^*$,

$$\min_{\delta\mathrm{T}} \ Q_{\bar{x}}(t_0)^{\mathsf{T}}\delta\mathbf{x}_{t_0} + Q_{\bar{u}}(t_0)^{\mathsf{T}}\delta\mathbf{u}_{t_0} + \widetilde{Q}_{\bar{T}}(t_0)\delta\mathrm{T}$$

$$+ \frac{1}{2}\begin{bmatrix}\delta\mathbf{x}_{t_0} \\ \delta\mathbf{u}_{t_0} \\ \delta\mathrm{T}\end{bmatrix}^{\mathsf{T}}\begin{bmatrix}\widetilde{Q}_{\bar{x}\bar{x}}(t_0) & \widetilde{Q}_{\bar{x}\bar{u}}(t_0) & \widetilde{Q}_{\bar{x}\bar{T}}(t_0) \\ \widetilde{Q}_{\bar{u}\bar{x}}(t_0) & \widetilde{Q}_{\bar{u}\bar{u}}(t_0) & \widetilde{Q}_{\bar{u}\bar{T}}(t_0) \\ \widetilde{Q}_{\bar{T}\bar{x}}(t_0) & \widetilde{Q}_{\bar{T}\bar{u}}(t_0) & \widetilde{Q}_{\bar{T}\bar{T}}(t_0)\end{bmatrix}\begin{bmatrix}\delta\mathbf{x}_{t_0} \\ \delta\mathbf{u}_{t_0} \\ \delta\mathrm{T}\end{bmatrix},$$

which has an analytic feedback solution given by

$$\delta\mathrm{T}^*(\delta\mathbf{x}_{t_0}, \delta\mathbf{u}_{t_0}) = [\widetilde{Q}_{\bar{T}\bar{T}}(t_0)]^{-1}\left(\widetilde{Q}_{\bar{T}}(t_0) + \widetilde{Q}_{\bar{T}\bar{x}}(t_0)\delta\mathbf{x}_{t_0} + \widetilde{Q}_{\bar{T}\bar{u}}(t_0)\delta\mathbf{u}_{t_0}\right).$$

In practice, we drop the state differential $\delta\mathbf{x}_{t_0}$ and only keep the control differential $\delta\mathbf{u}_{t_0}$, which can be viewed as the parameter update $\delta\theta$ by recalling (6). With these, we arrive at the second-order feedback policy presented in (16).

**Practical implementation.** We consider a vanilla quadratic cost, $\widetilde{\Phi}(\mathrm{T}, \mathbf{x}(\mathrm{T})) := \Phi(\mathbf{x}(\mathrm{T})) + \frac{c}{2}\mathrm{T}^2$, which penalizes longer integration time, and impose Gaussian-Newton approximation for the terminal cost, *i.e.* $\Phi_{\bar{x}\bar{x}} \approx \Phi_{\bar{x}}\Phi_{\bar{x}}^{\mathsf{T}}$. With these, the terminal conditions in (34) can be simplified to

$$\widetilde{Q}_{\bar{T}}(\bar{T}) = c\bar{T} + \Phi_{\bar{x}}^{\mathsf{T}}\bar{F}, \quad \widetilde{Q}_{\bar{T}\bar{T}}(\bar{T}) = c + \left(\Phi_{\bar{x}}^{\mathsf{T}}\bar{F}\right)^2, \quad \widetilde{Q}_{\bar{T}\bar{x}}(\bar{T}) = \left(\Phi_{\bar{x}}^{\mathsf{T}}\bar{F}\right)\Phi_{\bar{x}}^{\mathsf{T}}.$$

Since $\widetilde{Q}_{\bar{T}}(t)$ and $\widetilde{Q}_{\bar{T}\bar{T}}(t)$ are time-invariant (see (34a, 34b)), we know the values of $\widetilde{Q}_{\bar{T}}(t_0)$ and $\widetilde{Q}_{\bar{T}\bar{T}}(t_0)$ at the terminal stage. Further, one can verify that $\forall t \in [t_0, \bar{T}], \quad \widetilde{Q}_{\bar{T}\bar{u}}(t) = \left(\Phi_{\bar{x}}^{\mathsf{T}}\bar{F}\right)Q_{\bar{u}}(t)^{\mathsf{T}}$. In other words, the feedback term $\widetilde{Q}_{\bar{T}\bar{u}}$ simply rescales the first-order derivative $\widetilde{Q}_{\bar{u}}$ by $\Phi_{\bar{x}}^{\mathsf{T}}\bar{F}$. These reasonings suggest that we can evaluate the second-order feedback policy (16) almost at no cost without augmenting any additional state to `ODESolve`. Finally, to adopt the stochastic training, we keep the moving averages of all terms and update T with (16) every 50-100 training iterations.

### A.4 Experiment Details

All experiments are conducted on the same GPU machine (TITAN RTX) and implemented with `pytorch`. Below we provide full discussions on topics that are deferred from Section 4.

**Model configuration.** Here, we specify the model for each dataset. We will adopt the following syntax to describe the layer configuration.

- `Linear(input_dim, output_dim)`
- `Conv(output_channel, kernel, stride)`

Table 7: Configuration of the vector field $F(t, \mathbf{x}_t, \theta)$ of Neural ODEs used for each dataset
([‡]MIT License; [§]Apache License)

| Dataset | DNN architecture as $F(t, \mathbf{x}_t, \theta)$ | Model reference |
|---|---|---|
| MNIST SVHN CIFAR10 | `Conv(64,3,1)` $\to$ `ReLU` $\to$ `Conv(64,3,1)` | Chen et al. (2018)[‡] |
| SpoAD CharT | `Linear(32,32)` $\to$ `Tanh` $\to$ `Linear(32,32)` $\to$ `Tanh` $\to$ `Linear(32,32)` $\to$ `Tanh` $\to$ `Linear(32,32)` | Kidger et al. (2020b)[§] |
| ArtWR | `Linear(64,64)` $\to$ `Tanh` $\to$ `Linear(64,64)` $\to$ `Tanh` $\to$ `Linear(64,64)` $\to$ `Tanh` $\to$ `Linear(64,64)` | Kidger et al. (2020b)[§] |
| Circle | `Linear(2,64)`[??] $\to$ `Tanh` $\to$ `Linear(64,2)`[6] | Chen et al. (2018)[‡] |
| Gas | `ConcatSquashLinear(8,160)` $\to$ `Tanh` $\to$ `ConcatSquashLinear(160,160)` $\to$ `Tanh` $\to$ `ConcatSquashLinear(160,160)` $\to$ `Tanh` $\to$ `ConcatSquashLinear(160,8)` | Grathwohl et al. (2018)[‡] |
| Miniboone | `ConcatSquashLinear(43,860)` $\to$ `SoftPlus` $\to$ `ConcatSquashLinear(860,860)` $\to$ `SoftPlus` $\to$ `ConcatSquashLinear(860,43)` | Grathwohl et al. (2018)[‡] |

Table 8: Hyper-parameter grid search considered for each method

| Method | Learning rate | Weight decay |
|---|---|---|
| Adam | { 1e-4, 3e-4, 5e-4, 7e-4, 1e-3, 3e-3, 5e-3, 7e-3, 1e-2, 3e-2, 5e-2 } | {0.0, 1e-4, 1e-3 } |
| SGD | { 1e-3, 3e-3, 5e-3, 7e-3, 1e-2, 3e-2, 5e-2, 7e-2, 1e-1, 3e-1, 5e-1 } | {0.0, 1e-4, 1e-3 } |
| **Ours** | { 1e-3, 3e-3, 5e-3, 7e-3, 1e-2, 3e-2, 5e-2, 7e-2, 1e-1, 3e-1, 5e-1 } | {0.0, 1e-4, 1e-3 } |

- `ConcatSquashLinear(input_dim, output_dim)`[5]

- `GRUCell(input_dim, hidden_dim)`

Table 7 details the vector field $F(t, \mathbf{x}_t, \theta)$ of Neural ODEs for each dataset. All vector fields are represented by some DNNs, and their architectures are adopted from previous references as listed. The convolution-based feature extraction of image-classification models consists of 3 convolution layers connected through `ReLU`, *i.e.* `Conv(64,3,1)` $\to$ `ReLU` $\to$ `Conv(64,4,2)` $\to$ `ReLU` $\to$ `Conv(64,4,2)`. For time-series models, We set the dimension of the hidden space to 32, 64, and 32 respectively for SpoAD, ArtWR, and CharT. Hence, their GRU cells are configured by `GRUCell(27,32)`, `GRUCell(19,64)`, and `GRUCell(7,32)`. Since these models take regular time-series with the interval of 1 second, the integration intervals of their Neural ODEs are set to $\{0, 1, \cdots, K\}$, where $K$ is the series length listed in Table 3. Finally, we find that using 1 Neural ODE is sufficient to achieve good performance on Circle and Miniboone, whereas for Gas, we use 5 Neural ODEs stacked in sequence.

**Tuning process.** We perform a grid search on tuning the hyper-parameters (*e.g.* learning rate, weight decay) for each method on each dataset. The search grid for each method is detailed in Table 8. All figures and tables mentioned in Section 4 report the best-tuned results. For time-series models, we employ standard learning rate decay and note that without this annealing mechanism, we are unable to have first-order baselines converge stably. We also observe that the magnitude of the gradients of the GRU cells is typically 10-50 larger than the one of the Neural ODEs. This can make training unstable when the same configured optimizer is used to train all modules. Hence, in practice we fix Adam to train the GRUs while varying the optimizer for training Neural ODEs. Lastly, for image classification models, we deploy our method together with the standard Kronecker-based method (Grosse & Martens, 2016) for training the convolution layers. This enables full second-order training

---

[5] https://github.com/rtqichen/ffjord/blob/master/lib/layers/diffeq_layers/basic.py#L76

for the entire model, where the Neural ODE, as a continuous-time layer, is trained using our method proposed in Alg. 1. Finally, the momentum value for SGD is set to 0.9.

**Dataset.** All image datasets are preprocessed with standardization. To accelerate training, we utilize 10% of the samples in Gas, which still contains 85,217 training samples and 10,520 test samples. In general, the relative performance among training methods remains consistent for larger dataset ratios.

**Setup and motivation of Fig. 5.** We initialize all Neural ODEs with the *same* parameters while only varying the integration bound $t_1$. By manually grid-searching over $t_1$, Fig. 5 implies that despite initializing from the same parameter, different $t_1$ can yield distinct training time and accuracy; in other words, different $t_1$ can lead to distinct ODE solution. As an ideal Neural ODE model should keep the training time as small as possible without sacrificing the accuracy, there is a clear motivation to adaptive/optimize $t_1$ throughout training. Additional comparison w.r.t. standard (*i.e.* static) residual models can be founded in Appendix A.5.

**Generating Fig. 8.** The numerical values of the per-iteration runtime are reported in Table 9, whereas the ones for the memory consumption are given in Table 10. We use the last rows (*i.e.* $\frac{\text{SNOpt}}{\text{Adam}}$) of these two tables to generate Fig. 8.

Table 9: Per-iteration runtime (seconds) of different optimizers on each dataset

| | Image Classification | | | Time-series Prediction | | | Continuous NF | | |
|---|---|---|---|---|---|---|---|---|---|
| | MNIST | SVHN | CIFAR10 | SpoAD | ArtWR | CharT | Circle | Gas | Minib. |
| Adam | 0.15 | 0.78 | 0.17 | 5.24 | 9.95 | 14.79 | 0.34 | 2.25 | 0.65 |
| SGD | 0.15 | 0.81 | 0.17 | 5.23 | 10.00 | 14.77 | 0.33 | 2.28 | 0.74 |
| **SNOpt** | 0.15 | 0.68 | 0.20 | 5.18 | 10.05 | 14.89 | 0.94 | 4.34 | 1.04 |
| $\frac{\text{SNOpt}}{\text{Adam}}$ | 1.00 | 0.87 | 1.16 | 0.99 | 1.01 | 1.01 | 2.75 | 1.93 | 1.60 |

Table 10: Memory Consumption (GBs) of different optimizers on each dataset

| | Image Classification | | | Time-series Prediction | | | Continuous NF | | |
|---|---|---|---|---|---|---|---|---|---|
| | MNIST | SVHN | CIFAR10 | SpoAD | ArtWR | CharT | Circle | Gas | Minib. |
| Adam | 1.23 | 1.29 | 1.29 | 1.39 | 1.18 | 1.24 | 1.13 | 1.17 | 1.28 |
| SGD | 1.23 | 1.28 | 1.28 | 1.39 | 1.18 | 1.24 | 1.13 | 1.17 | 1.28 |
| **SNOpt** | 1.64 | 1.81 | 1.81 | 1.49 | 1.28 | 1.34 | 1.15 | 1.34 | 1.68 |
| $\frac{\text{SNOpt}}{\text{Adam}}$ | 1.33 | 1.40 | 1.40 | 1.07 | 1.09 | 1.08 | 1.02 | 1.14 | 1.31 |

**Tikhonov regularization in line 10 of Alg. 1.** In practice, we apply Tikhonov regularization to the precondition matrix, *i.e.* $\mathcal{L}_{\theta^n \theta^n} + \epsilon \mathbf{I}$, where $\theta^n$ is the parameter of layer $n$ (see Fig. 3 and (13)) and $\epsilon$ is the Tikhonov regularization widely used for stabilizing second-order training (Botev et al., 2017; Zhang et al., 2019). To efficiently compute this $\epsilon$-regularized Kronecker precondition matrix without additional factorization or approximation (*e.g.* Section 6.3 in Martens & Grosse (2015)), we instead follow George et al. (2018) and

---

**Algorithm 2** $\epsilon$-regularized Kronecker Update

1: **Input:** Tikhonov regularization $\epsilon$, amortization $\alpha$,
      Kronecker matrices $\bar{\mathbf{A}}_n$ $\bar{\mathbf{B}}_n$
2: $\mathbf{U}_A, \Sigma_A = \texttt{EigenDecomposition}(\bar{\mathbf{A}}_n)$
3: $\mathbf{U}_B, \Sigma_B = \texttt{EigenDecomposition}(\bar{\mathbf{B}}_n)$
4: $\mathbf{X} := \texttt{vec}^{-1}((\mathbf{U}_A \otimes \mathbf{U}_B)^\top \mathcal{L}_{\theta^n}) = \mathbf{U}_B^\top \widetilde{\mathcal{L}}_{\theta^n} \mathbf{U}_A$
5: $\mathbf{S}^* := \alpha \mathbf{S}^* + (1-\alpha)\mathbf{X}^2$
6: $\mathbf{X} := \mathbf{X}/(\mathbf{S}^* + \epsilon)$
7: $\delta\theta := (\mathbf{U}_A \otimes \mathbf{U}_B)\texttt{vec}(\mathbf{X}) = \texttt{vec}(\mathbf{U}_B \mathbf{X} \mathbf{U}_A^\top)$
8: $\theta \leftarrow \theta - \eta \delta\theta$

---

perform eigen-decompositions, *i.e.* $\bar{\mathbf{A}}_n = \mathbf{U}_A \Sigma_A \mathbf{U}_A^\top$ and $\bar{\mathbf{B}}_n = \mathbf{U}_B \Sigma_B \mathbf{U}_B^\top$, so that we can utilize the property of Kronecker product (Schacke, 2004) to obtain

$$(\bar{\mathbf{A}}_n + \bar{\mathbf{B}}_n + \epsilon \mathbf{I})^{-1} = (\mathbf{U}_A \otimes \mathbf{U}_B)(\Sigma_A \otimes \Sigma_B + \epsilon)^{-1}(\mathbf{U}_A \otimes \mathbf{U}_B)^\top. \tag{35}$$

This, together with the eigen-based amortization which substitutes the original diagonal matrix $\mathbf{S} := \Sigma_A \otimes \Sigma_B$ in (35) with $\mathbf{S}^* := ((\mathbf{U}_A \otimes \mathbf{U}_B)^\top \mathcal{L}_{\theta^n})^2$, leads to the computation in Alg. 2. Note that vec is the shorthand for vectorization, and we denote $\mathcal{L}_{\theta^n} = \texttt{vec}(\widetilde{\mathcal{L}}_{\theta^n})$. Finally, $\alpha$ is the

amortizing coefficient, which we set to 0.75 for all experiments. As for $\epsilon$, we test 3 different values from {0.1, 0.05, 0.03} and report the best result.

**Error bar in Table 4.** Table 11 reports the standard derivations of Table 4, indicating that our result remains statistically sound with comparatively lower variance.

Table 11: Test-time performance: accuracies for image and time-series datasets; NLL for CNF datasets

|  | MNIST | SVHN | CIFAR10 | SpoAD | ArtWR | CharT | Circle | Gas | Minib. |
|---|---|---|---|---|---|---|---|---|---|
| Adam | 98.83±0.18 | 91.92±0.33 | 77.41±0.51 | 94.64±1.12 | 84.14±2.53 | 93.29±1.59 | 0.90±**0.02** | -6.42±**0.18** | 13.10±0.33 |
| SGD | 98.68±0.22 | 93.34±1.17 | 76.42±0.51 | **97.70**±0.69 | 85.82±3.83 | 95.93±0.22 | 0.94±0.03 | -4.58±0.23 | 13.75±0.19 |
| **SNOpt** | **98.99**±0.15 | **95.77**±0.18 | **79.11**±0.48 | 97.41±0.46 | **90.23**±1.49 | **96.63**±0.19 | **0.86**±0.04 | **-7.55**±0.46 | **12.50**±0.12 |

**Discussion on Footnote 4.** Here, we provide some reasoning on why the preconditioned updates may lead the parameter to regions that are stabler for integration. We first adopt the theoretical results in Martens & Grosse (2015), particularly their Theorem 1 and Corollary 3, to our setup.

**Corollary 9** (Preconditioned Neural ODEs). *Updating the parameter of a Neural ODE, $F(\cdot, \cdot, \theta)$, with the preconditioned updates in (14) is equivalent to updating the parameter $\theta^\dagger \in \mathbb{R}^n$ of a "preconditioned" Neural ODE, $F^\dagger(\cdot, \cdot, \theta^\dagger)$, with gradient descent. This preconditioned Neural ODE has all the activations $\mathbf{z}^n$ and derivatives $F_{\mathbf{h}^n}^\mathsf{T} \mathbf{q}_i$ (see Fig. 3) centered and whitened.*

These centering and whitening mechanisms are known to enhance convergence (Desjardins et al., 2015) and closely relate to Batch Normalization (Ioffe & Szegedy, 2015), which effectively smoothens the optimization landscape (Santurkar et al., 2018). Hence, one shall expect it also smoothens the diffeomorphism of both the forward and backward ODEs (1, 5) of Neural ODEs.

### A.5 Additional Experiments

$t_1$ **optimization.** Fig. 12 shows that a similar behavior (as in Fig. 5) can be found when training MNIST: while the accuracy remains almost stationary as we decrease $t_1$ from 1.0, the required training time can drop by 20-35%. Finally, we provide additional experiments for $t_1$ optimization in Fig. 13. Specifically, Fig. 13a repeats the same experiment (as in Fig. 10) on training MNIST, showing that our method (green curve) converges faster than the baseline. Meanwhile, Fig. 13b and 13c suggest that our approach is also more effective in recovering from an unstable initialization of $t_1$. Note that both Fig. 10 and 13 use Adam to optimize the parameter $\theta$.

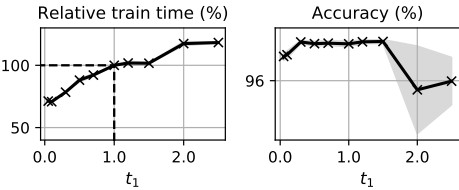

Figure 12: Training performance of MNIST with Adam when using different $t_1$.

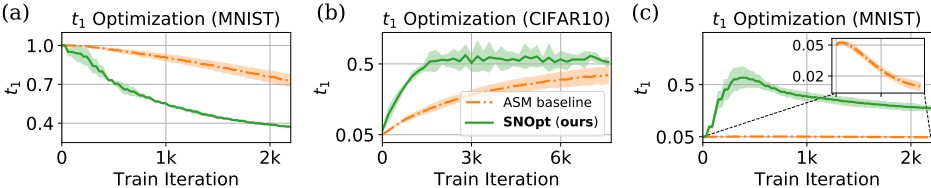

Figure 13: Dynamics of $t_1$ over training using different methods, where we consider (a) MNIST training with $t_1$ initialized to 1.0, and (b, c) CIFAR10 and MNIST training with $t_1$ initialized to some unstable small values (*e.g.* 0.05).

**Convergence on all datasets.** Figures 14 and 15 report the training curves of all datasets measured either by the wall-clock time or training iteration.

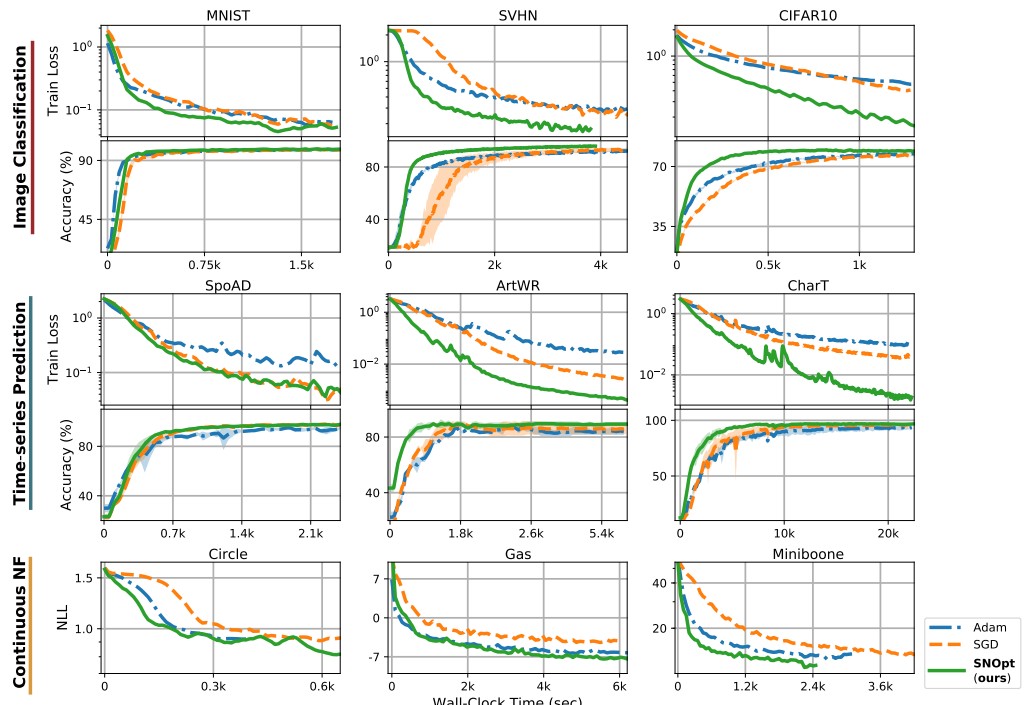

Figure 14: Optimization performance measured by *wall-clock* time across 9 datasets, including image (1st-2nd rows) and time-series (3rd-4th rows) classification, and continuous NF (5th row). We repeat the same figure with update iterations as x-axes in Fig 15. Our method (green) achieves faster convergence rate compared to first-order baselines. Each curve is averaged over 3 random trials.

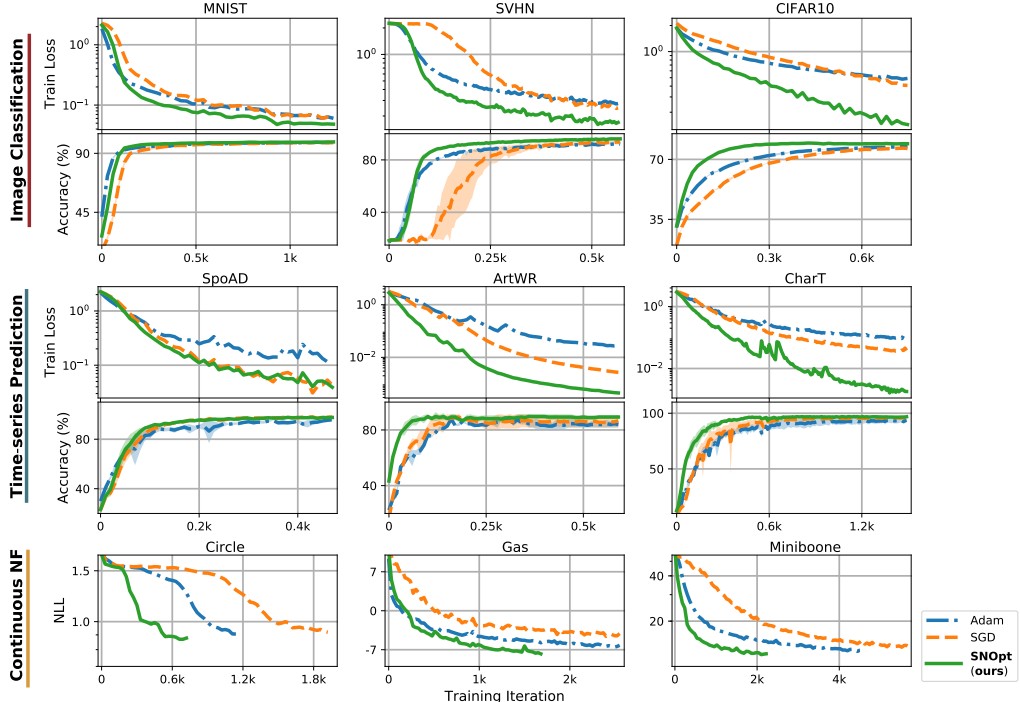

Figure 15: Optimization performance measured by *iteration updates* across 9 datasets, including image (1st-2nd rows) and time-series (3rd-4th rows) classification, and continuous NF (5th row). Each curve is averaged over 3 random trials.

**Comparison with first-order methods that handle numerical errors.** Table 12 and 13 report the performance difference between vanilla first-order methods (*e.g.* Adam, SGD), first-order methods equipped with error-handling modules (specifically MALI (Zhuang et al., 2021)), and our SNOpt. While MALI does improve the accuracies of vanilla first-order methods at the cost of extra per-iteration runtime (roughly 3 times longer), our method achieves highest accuracy among all optimization methods and retains a comparable runtime compared to *e.g.* vanilla Adam.

Table 12: Test-time performance (accuracies %) w.r.t. different optimization methods

|  | Adam | Adam + MALI | SGD | SGD + MALI | **SNOpt** |
|---|---|---|---|---|---|
| SVHN | 91.92 | 91.98 | 93.34 | 94.33 | **95.77** |
| CIFAR10 | 77.41 | 77.70 | 76.42 | 76.41 | **79.11** |

Table 13: Per-iteration runtime (seconds) w.r.t. different optimization methods

|  | Adam | Adam + MALI | SGD | SGD + MALI | **SNOpt** |
|---|---|---|---|---|---|
| SVHN | 0.78 | 2.31 | 0.81 | 1.28 | **0.68** |
| CIFAR10 | **0.17** | 0.55 | **0.17** | 0.23 | 0.20 |

**Comparison with LBFGS.** Table 14 reports various evaluational metrics between LBFGS and our SNOpt on training MNIST. First, notice that our method achieves superior final accuracy compared to LBFGS. Secondly, while both methods are able to converge to a reasonable accuracy (90%) within similar iterations, our method runs 5 times faster than LBFGS per iteration; hence converges much faster in wall-clock time. In practice, we observe that LBFGS can exhibit unstable training without careful tuning on the hyper-parameter of Neural ODEs, *e.g.* the type of ODE solver and tolerance.

Table 14: Comparison between LBFGS and our SNOpt on training MNIST

|  | Accuracy (%) | Runtime (sec/itr) | Iterations to Accu. 90% | Time to Accu. 90% |
|---|---|---|---|---|
| LBFGS | 92.76 | 0.75 | 111 steps | 2 min 57 s |
| **SNOpt** | 98.99 | 0.15 | 105 steps | 18 s |

**Results with different ODE solver (`implicit adams`).** Table 15 reports the test-time performance when we switch the ODE solver from `dopri5` to `implicit adams`. The result shows that our method retains the same leading position as appeared in Table 4, and the relative performance between optimizers also remains unchanged.

Table 15: Test-time performance using "`implicit adams`" ODE solver: accuracies for image and time-series datasets; NLL for CNF datasets

|  | MNIST | SVHN | CIFAR10 | SpoAD | ArtWR | CharT | Circle | Gas | Miniboone |
|---|---|---|---|---|---|---|---|---|---|
| Adam | 98.86 | 91.76 | 77.22 | 95.33 | 86.28 | 88.83 | 0.90 | -6.51 | 13.29 |
| SGD | 98.71 | 94.19 | 76.48 | **97.80** | 87.05 | 95.38 | 0.93 | -4.69 | 13.77 |
| **SNOpt** | **98.95** | **95.76** | **79.00** | 97.45 | **89.50** | **97.17** | **0.86** | **-7.41** | **12.37** |

**Comparison with discrete-time residual networks.** Table 16 reports the training results where we replace the Neural ODEs with standard (*i.e.* discrete-time) residual layers, $\mathbf{x}_{k+1} = \mathbf{x}_k + F(\mathbf{x}_k, \theta)$. Since ODE systems can be made invariant w.r.t. time rescaling (*e.g.* consider $\frac{dx}{dt} = F(t, x, \theta)$ and $\tau = ct$, then $\frac{dx}{d\tau} = \frac{1}{c} F(\frac{\tau}{c}, x, \theta)$ will give the same trajectory $x(t) = x(\frac{\tau}{c})$), the results of these residual networks provide a performance validation for our joint optimization of $t_1$ and $\theta$. Comparing Table 16 and 5 on training CIFAR10, we indeed find that SNOpt is able to reach the similar performance (77.82% *vs.* 77.87%) of the residual network, whereas the ASM baseline gives only 76.61%, which is 1% lower.

Table 16: Accuracies (%) of residual networks trained with Adam or SGD

|  | MNIST | SVHN | CIFAR10 |
|---|---|---|---|
| resnet + Adam | $98.75 \pm 0.21$ | $97.28 \pm 0.37$ | $77.87 \pm 0.44$ |

**Batch size analysis.** Table 17 provides results on image classification when we enlarge the batch size by the factor of 4 (*i.e.* $128 \rightarrow 512$). It is clear that our method retains the same leading position with a comparatively smaller variance. We also note that while enlarging batch size increases the memory for all methods, the ratio between our method and first-order baselines does not scale w.r.t. this hyper-parameter. Hence, just as enlarging batch size may accelerate first-order training, it can equally improve our second-order training. In fact, a (reasonably) larger batch size has a side benefit for second-order methods as it helps stabilize the preconditioned matrices, *i.e.* $\bar{A}_n$ and $\bar{B}_n$ in (14), throughout the stochastic training (note that too large batch size can still hinder training (Keskar et al., 2016)).

Table 17: Accuracies (%) when using larger ($128 \rightarrow 512$) batch sizes

|  | MNIST | SVHN | CIFAR10 |
|---|---|---|---|
| Adam | $99.14 \pm 0.12$ | $94.19 \pm 0.18$ | $77.57 \pm 0.30$ |
| SGD | $98.92 \pm 0.08$ | $95.67 \pm 0.48$ | $76.66 \pm 0.29$ |
| **SNOpt** | $\mathbf{99.18 \pm 0.07}$ | $\mathbf{98.00 \pm 0.12}$ | $\mathbf{80.03 \pm 0.10}$ |