# OpenReview forum: "Second-Order Neural ODE Optimizer"
_NeurIPS.cc/2021/Conference — NeurIPS 2021 Spotlight_

### Official Review · Reviewer_w3BT · 2021-07-04

**Rating:** 8
**Confidence:** 5

**Summary:**

This paper presents a novel second-order optimization technique for neural ordinary differential equations (NODEs), derived from the optimal control perspective of NODEs. In a nutshell, the authors consider the second-order Taylor expansion of the loss term, leading to matrix-valued ODEs that give second-order derivatives. A series of approximations are proposed to compute a low-rank representation of the second-order preconditioning matrices. The experiments demonstrate that the proposed optimization method converges faster than a first-order optimization routine in all neural ODE applications at the cost of slightly higher memory usage. The method also allows optimizing the integration time in continuous-time normalizing flows.

**Limitations And Societal Impact:**

The limitations are clearly described. No potentially negative societal impact I can think of.

**Main Review:**

The integration operation in continuous-time models can be overwhelmingly time-consuming, which oftentimes makes it difficult to apply neural ODEs to even simple, low-dimensional problems. Therefore, the proposed approach addresses an overlooked but fundamental restriction of a rather recent model family. The paper is well-written. Despite the heavy technical details, the presentation is very clear and I had no difficulty grasping the motivation and ideas. The experiments and the discussion of the results are exhaustive. Experiments indeed show that the presented optimization scheme improves the NODE training in all three applications of NODEs.

A few concrete suggestions and questions:
- Since the approximations are very involved, having a broader picture of them would be nice. For instance, a cartoon illustration of the series of approximations and what they aim to address could help the reader gain insights more easily. Also, additional (simple) ablation studies that compare approximations with the corresponding exact expressions might reveal how good/tight the approximations are.
- A direct comparison against a Quasi-Newton method (I suggest LBFGS) on a simple problem would be very interesting to see as such methods can scale to low/mid-size neural nets.
- Line 243: Which tolerance values (absolute/relative) does this refer to? Also, any justification for not using the default values?
- To support the claims in Line 272, it would be nice to test NODE with a regularization scheme, e.g., Finlay et al., 2020. Also, one may further check the number of function evaluations.
- Looking at Fig13 (train time), I wonder why it first drops and then goes up. Maybe because of a change in the stiffness of the vector field? Can this be experimentally demonstrated?

Minor details:
- I suggest writing "back-propagation" instead of "Back-propagation".
- Line117: ',and etc' seems unnecessary. Also, what is w.l.o.g?
- An important reference to [1] is missing.

[1] Stapor, Paul, Fabian Fröhlich, and Jan Hasenauer. "Optimization and profile calculation of ODE models using second order adjoint sensitivity analysis." Bioinformatics 34.13 (2018): i151-i159.

**Post-rebuttal comment:** Thanks authors for the rebuttal. As noted in my original review, this is a very nice paper and should be definitely accepted.

**Time Spent Reviewing:**

5

---

> ### Author Response · Authors · 2021-08-10
> **Author Response to Reviewer w3BT**
>
>
> **1. Comparison to LBFGS on MNIST**
>
> - In the table below we report various evaluational metrics between our method and LBFGS on training MNIST. First, notice that our method achieves superior final accuracy compared to LBFGS (99% vs 92%). Secondly, while both methods are able to converge to a reasonable accuracy (90%) within similar iterations (100~110 steps), our method runs 5 times faster than LBFGS per iteration (0.15 sec vs 0.75 sec); hence converges much faster in wall-clock time (18 sec vs 3 min).
>
> 	|  | Accuracy (%) | Runtime (sec/itr) | Itr to Accu. 90% | Time to Accu. 90% |
> 	|---|---|---|---|---|
> 	| LBFGS | 92.76 | 0.75 | 111 steps | 2 min 57 s |
> 	| Ours | 98.99 | 0.15 | 105 steps | 18 s |
>
> - In practice, we observe that LBFGS can exhibit unstable training without careful tuning on the hyper-parameter of NODEs, e.g. type of ODE solver and tolerance. This is in contrast to our method, which is designed specifically for deep continuous-time models (e.g. NODEs) by drawing inspiration from optimal control theory to perform much efficient and stable second-order training.
>
>
> ---
>
>
> **2. Clarification on tolerance**
>
> - The reported tolerance is used for both absolute and relative tolerances. We note that the default tolerance can vary on the application basis. In our case, we adopt the _same_ tolerance value from prior works that apply NODEs to either classification [2] or CNF [3]. In practice, CNF often requires smaller tolerance to achieve comparable results to other generative models [4], as its training often relies on propagating the instantaneous change of the log-density with an unbiased (yet noisy) trace estimator.
>
>
> ---
>
>
> **3. Support L272 with regularization (Finlay et al., 2020) and NFE**
>
> - We thank the reviewer for the great suggestion. To support our conjecture that our preconditioned updates may guide the parameter to regions that are numerically stabler (hence faster) for integration, we follow reviewer's suggestion and
> record the values of the Jacobian, $\int \lVert \nabla_x F \rVert^2$,  and Kinetic, $\int \lVert F \rVert^2$, regularization (Finlay et al., 2020) throughout the SVHN training. The table below reports these regularization values ($R_J$, $R_K$), together with the number of function evaluations of both forward (NFE-F) and backward (NFE-B) processes.
>
> 	|  | NFE-F | NFE-B | $R_J$ | $R_K$ | $R_J {+} R_K$ |
> 	|---|---|---|---|---|---|
> 	| Adam | 32.0 | 42.1 | 16.65 | 307.23 | 323.88 |
> 	| Ours | 26.0 | 32.6 | 21.09 | 178.01 | 199.10 |
>
> - Since $R_J$ and $R_K$ are typically augmented to the original objective with the same coefficient (Finlay et al., 2020), i.e. $\tilde{\mathcal{L}} {=} \mathcal{L} {+} \lambda (R_J {+} R_K)$, the summation $R_J {+} R_K$ reflects the _implicit regularization_ from the optimizer to accelerate the training, where lower value indicates stronger regularization. Interestingly, the ODE parameter found by our method indeed has a substantially lower $R_J {+} R_K$ value (hence stronger regularization and better-conditioned ODE dynamics) compared to Adam. This provides a possible explanation for the 20% reduction on both NFE-F and NFE-B when using our method, yet without hindering the test-time performance. We will include these results in the revision.
>
>
> ---
>
>
> **4. Correction of Fig 13 in Appendix**
>
> - We thank the reviewer for the careful examination. We identify errors when generating this specific subplot and provide the correct value (relative train time w.r.t. $t_1$=1.0) in the below table. The corrected result remains consistent with Fig 5 (page 7 in the main paper), showing that the required training time grows linearly (without any absurd drop) as $t_1$ increases. We stress that all figures come from the same submission package and all of our claims remain valid.
>
> 	| $t_1$ | 0.05 | 0.1 | 0.3 | 0.5 | 0.7 | 1.0 | 1.3 | 1.5 | 2.0 | 2.5 |
> 	|:---:|:---:|:---:|:---:|:---:|:---:|:---:|:---:|:---:|:---:|:---:|
> 	| Rel. train time (%) | 71.3 | 70.8 | 78.4 | 88.1 | 92.2 | 100 | 101.8 | 101.7 | 117.4 | 118.3 |
>
> ---
>
>
> **5. Other clarifications**
>
> - We will include additional pictorial illustrations (and analysis) to further improve the readability, for instance a comparison of memory complexity between different formula/approximation (see the third response to Reviewer HLDZ) with some cartoon illustration. "w.l.o.g" refers to "without loss of generality". We will use the full expression and include the reference [1] in the revision!
>
>
> ---
>
>
> [2] https://github.com/rtqichen/torchdiffeq/blob/master/examples/odenet_mnist.py#L14
> [3] https://github.com/rtqichen/ffjord/blob/master/train_tabular.py#L36-L37
> [4] Grathwohl et al., 2018; FFJORD

---

### Official Review · Reviewer_ikHp · 2021-07-14

**Rating:** 7
**Confidence:** 4

**Summary:**

This work proposes an alternative to the recursive adjoint method for computing higher order gradients of neural ODEs based on optimal control theory that promises to be more efficient and more accurate.

**Ethical Concerns:**

There are no ethical concerns.

**Limitations And Societal Impact:**

Impact is adequately addressed.

**Main Review:**

My main concern is that the authors do not take sufficient care in statements about the memory and time costs. The adjoint method is O(1) memory *in depth* i.e. in time, but it is O(n) in the number of parameters, i.e. $dim_R(\theta) = n$. Naively computing higher-order derivatives by recursively applying the adjoint method has O(n) memory footprint and O(n) time, but these calculations are independent and can be parallelised to give O(n^2) memory and O(1) time (as described by the authors L98). The proposed method maintains O(1) in depth, but has a footprint of O(n^2) ($Q_{uu}$ is matrix valued $\in R^{n\times n}$), and is O(1) in time.

So there isn’t actually any change in the memory or time costs as compared to recursively applying the adjoint method. There may be a difference in the error, as stated in L41-43, but there is no evidence presented in favour of this claim nor evidence that this is of practical consequence in the training of neural ODEs, which is the evaluation criteria used in this work.

L97-101 make some arguments against the recursive method, stating that the method scales unfavourably due to recursion and errors. That k=2 is apparently the useful limit of this recursion (‘*second*-order neural ODE optimizer') diminishes the significance of the first complaint and it is diminished further when we note that the time cost is linear in the order of the optimization, which is a fixed hyperparameter. Evidence is not presented to support the second complaint, and this is a missed opportunity as whether the accumulation of errors is of any significance in training is an interesting scientific question. Recursively applying the adjoint method is the essential baseline to compare the method against, particularly given the proposals from section 3.2 onwards to approximate the optimisation and L194 which introduces a further approximation explicitly to trade off complexity for performance, such that both methods run with some error.

This is a shame as the work is otherwise interesting and highly creative, and seeing this comparison seems essential.

I’d also be interested in some discussion about how the number of terms in equation 9 grows in the order of the optimisation. I followed the discussion in the appendix to Almubarak 2019, though the appendix indicates that there is a further reference within Almubarak 2019 that is to be followed next, though doesn’t specify which. Combinatorially, we expect to get 2^(n+1) - 2 (i.e. 2 for first order, 6 for second order, 14 for third order…) This seems to suggest that there is a further term missing from the memory footprint. Including the optimisation order, k, the costs seem to be
Recursive adjoint: memory O(n^k) time O(k)
Proposal: memory O(2^k n^k) time O(1)
Given that k is generally a fixed hyperparameter and ≤ 2 I doubt this is a pressing problem, but some comment would be appreciated.

Section 3.3 would benefit from including the choice of initialisation in the discussion. We can freely vary the integration time without modifying the output by making corresponding changes to the magnitude of the weights in the derivative function i.e. if you want to double the integration time, half the weights in the final layer of the derivative network. So discussions about the integration interval are missing important context. Furthermore, given that the model uses a CNN feature extractor (Appendix A.4 gives this as a 3 layer conv net) prior to the NODE, and that output of the integration is passed through a linear layer to get the prediction, the results in Figure 5 are explained equally as the ODE being superfluous. The correct baselines for comparison here would be 0 integration, which shows 50% reduction in train time for what appears to be a 2-3 percentage point reduction in accuracy, and a model that has the same capacity as the NODE added. In this case the ODE function is a 2 layer conv net, so a 5 layer conv net in total would be appropriate. Figure 13 in the appendix, showing the same experiment for MNIST, presents similarly inconclusive results, in that case the change in accuracy is even less significant across the range of values tested.

The results in Figure 7 are mixed, which isn’t accurately reflected in the discussion in the **Convergence and computational efficiency** paragraph. On SpoAD, SGD trains as quickly and to the same performance. On CharT the three methods reach similar levels of performance. The absence of errors in Table 2 makes it difficult to compare the significance of the bolded results (and appropriate baselines discussed above are absent).

Table 3 does not select the best training time model from Figure 5. There is a higher-performing and faster-training model on that plot, something in the region of t = 0.4 or 0.5 looks to have performance at least as good as that of the t = 1.0 model at a train time percentage of 50-75%.

Figure 9 is a useful plot, though the caption doesn’t seem to match the data: the model is sensitive to the choice of learning rate, with the training loss varying by approximately ~2x between high and low learning rates on CIFAR10. A higher learning rate is not investigated for Adam, which would appear to be on trend to match the performance of the proposal if a higher learning rate were used (perhaps it wouldn’t, the evidence isn’t provided.) Better would be to provide training plots as in Figure 7 for varying learning rates for the different methods.

Also absent is a discussion of the effect of batch size on the training plots. With the first order methods having a lower memory overhead, there should be more scope for increasing the batch size to accelerate training. Has this been investigated?

Overall, this is an exciting project that is let down by its evaluation. The absence of obvious baselines, most critically the recursive adjoint method, and seemingly cherry picked examples in the comparisons that are present, make it difficult to evaluate the impact of this work.

--- minor points
The presentation is clear and of high quality.
L91 should be ‘the backpropgation algorithm’
L122 seemly should be seeming
L160 ‘much more efficiently’


### Update
Following the rebuttal and back and forth with the authors, my concerns have been addressed and I'm happy to raise my score (4->7) to reflect this.

**Time Spent Reviewing:**

6

---

> ### Author Response · Authors · 2021-08-10
> **Author Response to Reviewer ikHp (Part 1/2)**
>
>
>
> **1. Misunderstanding on recursive adjoint baseline**
>
> **1.1. Recursive dependency of adjoint baseline**
>
> - We first clarify the recursive adjoint baseline mentioned in L40-41 and L97-98. The baseline refers to the suggestion from the official implementation of Neural ODE [1,2] on extending first-order adjoint method to higher-order derivatives by _recursively_ and _sequentially_ calling the adjoint-inherited autograd function. This has been later implemented in other work [3]. Algorithmically, it shares the following syntax.
>
> 	```
> 	1    class AdjointMethod(autograd.function):
> 	2        def forward(ODE, x0):
> 	3            return solve(ODE, x0) # standard ode solve and store forward info
> 	4
> 	5        def backward(...): # automatically invoke when calling "grad"
> 	6            adjoint_ode = AdjointODE(...) # construct adjoint ode using forward info
> 	7            return AdjointMethod.forward(adjoint_ode, ...) # apply adjoint method "recursively"
> 	8
> 	9    x1 = AdjointMethod.forward(ODE, x0) # invoke 1st forward call
> 	10   L, θ = loss(x1), ODE.parameters()
> 	11   dL_dθ = grad(L, θ) # invoke 1st backward call and 2nd forward call
> 	12   d2L_dθ2 = grad(dL_dθ, θ) # invoke 2nd backward call
> 	```
>
> - Similar to vanilla backpropagation, the adjoint method (despite being memory efficient in depth) still requires wrapping information from the forward pass (line 6). Hence, for the second adjoint-grad in line 12, its forward pass info won't be available until the first adjoint-grad (line 11) is called. In other words, this procedure exhibits _recursive dependency_ and therefore _cannot_ be parallelized. Our recursive formula in L98 should also indicate this recursive dependency (hence _not_ independent).
>
> - It is certainly plausible to parallelize the higher-order computation by better exploring the optimization structure, which is exactly what we have derived in Theorem 1. However, to our best knowledge this parallelization has _never_ been mentioned _nor_ appeared in prior Neural-ODEs works. Hence, it is unfair to compare (our) Theorem 1 against our proposed method (in fact, the complexity of the latter is incorrect either; see 1.4). We urge the reviewer to recognize the distinction.
>
>
> ---
>
>
> **1.2 Accumulated error of recursive adjoint baseline**
>
> - Regarding the effect of accumulated errors on hindering Neural ODEs training, the reviewer surmised that insufficient evidence is provided for this claim. We respectfully disagree, as we have stated (L34) and cited prior efforts [4-6] that identify the numerical issues of the adjoint method (also recognized by Reviewer HLDZ). Complaints against higher-order recursive adjoint also appeared in [7] (see page 6; _"... double adjoint ... this produces gradients that are sufficiently inaccurate as to prevent models from training"_) for training Neural SDE, which generalizes Neural ODE with additional diffusion.
>
>
> - To further solidify our claim, we implement the recursive adjoint baseline and compute its errors w.r.t. the ground-truth derivatives using backpropagation. The below table reports the L2 norm on 2 datasets (upper: MNIST, bottom: CIFAR10) with 3 ODE solvers (RK4, implicit Adams, Dopri5). For the reference, we also include the error of 1st-order derivative.
>
> 	|  | recursive adjoint+RK4 | recursive adjoint+Implicit Adams  | recursive adjoint+Dopri5 |
> 	|---|---|---|---|
> 	| 1st-order derivative error | 5.25e-4 | 1.41e-2 | 2.51e-4 |
> 	| 2nd-order derivative error | 0.12 | 3.08 | 7.94 |
>
>
>
> 	|  | recursive adjoint+RK4 | recursive adjoint+Implicit Adams  | recursive adjoint+Dopri5 |
> 	|---|---|---|---|
> 	| 1st-order derivative error | 7.63e-05 | 2.11e-3 | 3.43e-4 |
> 	| 2nd-order derivative error | 6.84e-3 | 0.25 | 41.10 |
>
> - It is clear that recursive adjoint accumulates derivative errors: typically 2-4 orders of magnitude larger than the 1st-order error. In fact, the accumulated error becomes notoriously large that prevents any reasonable training when using Dopri5, which is oftentime the default adaptive solver for training Neural ODEs [1,8,9]. We thereby validate that the recursive adjoint baseline can suffer from both recursive dependency and accumulated errors.
>
>
> ---
>
>
> **1.3 Additional experiment and comparison to our method**
>
> - One of the main criticisms from the reviewer is the missing comparison between recursive adjoint baseline and our method. In the below tables, we provide the test-time accuracy (%; upper table) and runtime (sec/itr; bottom table) on all 9 datasets. All values are averaged over 3 trails. Since recursive adjoint is quite unstable with adaptive solvers (due to the accumulated error; see table in 1.2), we have to consider fixed-step solvers (implicit adams gives best performance) instead. We try our best to tune this baseline on each dataset w.r.t. the type of solver, tolerance, and other hyper-parameters.
>
> |  | mnist | svhn | cifar10 | spoAD | ArtWR | CharT | Circle | Gas | Minib |
> |---|---|---|---|---|---|---|---|---|---|
> | recursive adjoint | 92.99±1.34 | 81.32± 2.49 | 66.55± 2.72 | 82.63±7.81 | 81.61±2.30 | 88.11±7.29 | 1.36 ± 0.07 | 5.27 ± 0.68 | 41.31 ± 1.32 |
> | Ours | 98.99 ± 0.15 | 95.77± 0.18 | 79.11±0.48 | 97.41± 0.46 | 90.23± 2.07 | 96.63±0.69 | 0.86±0.04 | -7.55±0.46 | 12.50±0.12 |
>
>
>
> |  | mnist | svhn | cifar10 | spoAD | ArtWR | CharT | Circle | Gas | Minib |
> |---|---|---|---|---|---|---|---|---|---|
> | recursive adjoint | 0.39 | 0.91 | 0.36 | 27.46 | 48.83 | 29.86 | 0.48 | 6.59 | 0.70 |
> | Ours | 0.15 | 0.68 | 0.20 | 5.18 | 10.05 | 14.89 | 0.94 | 4.34 | 1.04 |
> | $\frac{\text{recursive adjoint}}{\text{ours}}$ | 2.6 | 1.34 | 1.80 | 5.30 | 4.86 | 2.01 | 0.51 | 1.52 | 0.67 |
>
> - It is clear that our method outperforms the recursive adjoint baseline by a _large_ margin, and exhibits a much faster per-iteration runtime. On image and time-series datasets, our method leads the accuracies over baseline by 6-15% and can run up to 5 times faster. We note that while the baseline seems to run faster on CNF datasets, their losses remain close to initial values (1.6/8/60 for Circle/Gas/Minib), suggesting that the baseline has failed to yield any meaningful convergence.
>
> - In practice, we also observe that recursive adjoint can exhibit unstable training without careful tuning on the hyper-parameter of NODEs. This is in contrast to our method, which draws inspiration from both continuous-time optimal control theory and Kronecker-based methods to perform much efficient and stable second-order training.
>
>
> ---
>
>
> **1.4 Clarification on our method**
>
> - The reviewer surmised that the recursive adjoint baseline and our method share the same memory complexity $O(n^2)$, where $n$ is the number of parameters. We stress that _neither is correct_. While our clarification in 1.1 suggests that the baseline should stay in $O(n)$, our proposed method, which should be referred to Eq 13 & 14 (Alg 1), has the memory complexity $O(n+m^2)$, where $m^2$ is the dimension of matrices $\bar{A}$ and $\bar{B}$ (L185).
>
>
> - Eq 9 & 10 (Theorem 1) indeed have the memory footprint $O(n^2)$, as conjectured by the reviewer. However, we have _never_ claimed that Eq 9 & 10 are the "proposed" method used for training; on the contrary, we have particularly stated (L155-159) that Eq 9 & 10 are, despite inspiring, still inefficient for practical usage. We have also explicitly mentioned (L187-189) that Alg 1, which instead implements Eq 13 & 14, as our final proposed method. We urge the reviewer to recognize the distinction.
>
>
> ---
>
>
> **1.5 Summary**
>
> - We clarify the recursive adjoint method mentioned in L40-41/L97-98 and provide its implementation details **[1.1]**, which are grounded and supported by numerous references [1-3,7]. We provide strong evidence showing _(i)_ this recursive adjoint baseline can suffer from recursive runtime dependency and accumulated errors **[1.2]**, and _(ii)_ our method outperforms this baseline on both test-time performance and per-iteration runtime **[1.3]**. Finally, we clarify the difference between the memory complexity of two methods ($O(n)$ vs $O(n+m^2)$) **[1.4]**. In the revision, we will include discussions on memory analysis (in terms of $n$ and $m$ rather than depth), the reference [7], and additional experiments. We sincerely hope the reviewer will re-evaluate our contribution in its entirety.
>
>
> ---
>
>
> [1] Chen et al 2019, Neural ODE
> [2] https://github.com/rtqichen/torchdiffeq/blob/master/torchdiffeq/_impl/adjoint.py#L72
> [3] https://github.com/google-research/torchsde/blob/master/torchsde/_core/adjoint.py#L98
> [4] Gholami et al 2019, "Anode: Unconditionally accurate memory-efficient gradients for neural odes"
> [5] Zhuang et al 2020, "Adaptive checkpoint adjoint method for gradient estimation in neural ode"
> [6] Zhuang et al 2021, "MALI: A memory efficient and reverse accurate integrator for Neural ODEs"
> [7] Kidger et al 2020, https://arxiv.org/pdf/2102.03657.pdf
> [8] Grathwohl et al., 2018; "FFJORD: Free-form Continuous Dynamics for Scalable Reversible Generative Models"
> [9] Kidger et al., 2020b; "Neural Controlled Differential Equations for Irregular Time Series"

---

> > ### Author Response · Authors · 2021-08-10
> > **Author Response to Reviewer ikHp (Part 2/2)**
> >
> > **2. Misinterpretation on architecture optimization (Sec 3.3, Fig 5, Table 3)**
> >
> > - The reviewer suspected the equivalence between Neural ODEs with different integration time ($t_1$) by stating _"... we can freely vary $t_1$ without modifying output ..."_, which led to a misinterpretation on Sec 3.3. We stress that this surmise is _incorrect_ for Neural ODEs, which consider _time-varying_ systems $F(t,x, \theta)$ rather than time-invariant. Halving the derivative function of time-varying system does _not_ correspond to doubling $t_1$. Consider two simple systems $\frac{\mathrm{d}x}{\mathrm{d}t} {=} (1-2t)x$ and $\frac{\mathrm{d}y}{\mathrm{d}t} {=} 0.5(1-2t)y$ with the same initial condition $x(0) {=} y(0) {=} 1$. Their solutions are respectively $x(t) {=} \exp(t-t^2)$ and $y(t) {=} \exp(0.5(t-t^2))$, hence implying $x(t) {\neq} y(2t)$. As all of our derivations and experiments are based on standard _time-varying_ Neural ODEs [10], this misinterpretation can lead to ill-founded doubts on the credibility of our experiments. We urge the reviewer to recognize the difference.
> >
> >
> > - The purpose of Fig 5 is to empirically demonstrate (by _manually_ grid-searching over $t_1$) that despite initializing from the same parameter, different $t_1$ can yield different training time and accuracy. We have _never_ claimed the equivalence of the Neural ODE models in Fig 5; on the contrary, since different $t_1$ can lead to distinct ODE solution, and an ideal Neural ODE model should keep the training time as small as possible without sacrificing the accuracy, there is a clear motivation to adaptive/optimize $t_1$ throughout training. This is also recognized by Reviewer 1V2i. We will include the setup of Fig 5 in the revision.
> >
> >
> > - On the other hand, Table 3 presents the results of _adaptively training_ $t_1$ using our second-order feedback policy (Eq 16). This policy aims to optimize $t_1$ given a prior-designed penalty on larger $t_1$ (L211-212, L567) rather than selecting the model appeared in Fig 5, which was found by _exhaustive search_ and presented only for motivation purposes. While it is possible to encourage lower $t_1$ by properly tuning the penalty, we emphasize that our method has already achieved superior performance compared to first-order baseline (Table 3 &Fig 10).
> >
> > ---
> >
> > **3. Discussion on higher-order ($\ge$3) optimization in Appendix**
> >
> > - The number of backward ODEs in Eq 9, as conjectured by the reviewer, indeed scales exponentially for higher order optimization. However, we stress again that Eq 9 does _not_ correspond to our proposed method. Rather, it serves as an initial attempt that opens up a previously unexplored optimization framework from optimal control theory, and from which we must proceed with further handling on both memory (Proposition 2; Eq 11 & 12) and runtime (Kronecker approximation; Eq 13 & 14) in order to derive an efficient and stable second-order method.
> >
> >
> > - In this vein, we believe topics such as tensor factorization and advanced automatic differentiation (AD) are worth investigating for those pursuing in this direction. For instance, [11] extended the Kronecker-factorized matrices to general higher-order tensors, and [12] seems to explore a similar factorization for optimizing high-dimensional PDE. Alternatively, [13] develops a recursive Taylor-mode AD that promises to compute higher-order derivatives at $O(k^2)$ rather than $O(\exp(k))$. Exploring these factorizations, as we have provided strong results in this work, has great potential to facilitate practical training.
> >
> >
> > ---
> >
> >
> > **4. Correction of Fig 13 in Appendix**
> >
> > - We thank the reviewer for the careful examination. We identify errors when generating this specific subplot and
> > provide the correct value (relative train time w.r.t. $t_1$=1.0) in the below table. The corrected result remains consistent with Fig 5 (page 7 in the main paper), showing that the required training time grows linearly (without any absurd drop) as $t_1$ increases. We stress that all figures come from the same submission package and all of our claims remain valid.
> >
> > 	| $t_1$ | 0.05 | 0.1 | 0.3 | 0.5 | 0.7 | 1.0 | 1.3 | 1.5 | 2.0 | 2.5 |
> > 	|:---:|:---:|:---:|:---:|:---:|:---:|:---:|:---:|:---:|:---:|:---:|
> > 	| Rel. train time (%) | 71.3 | 70.8 | 78.4 | 88.1 | 92.2 | 100 | 101.8 | 101.7 | 117.4 | 118.3 |
> >
> >
> > ---
> >
> >
> > **5. Error bar in Table 2**
> >
> > - In the below table we provide standard deviation for Table 2, and we refer the reviewer to the above 1.3 for the results of recursive adjoint baseline. The table indicates that our result in Table 2 remains statistically sound with comparatively lower variance.
> >
> > |  | mnist | svhn | cifar10 | spoAD | ArtWR | CharT | Circle | Gas | Minib |
> > |---|---|---|---|---|---|---|---|---|---|
> > | Adam | 98.83± 0.18 | 91.92±0.33 | 77.41±0.51 | 94.64± 1.12 | 84.14±2.53 | 93.29±1.59 | 0.90±0.02 | -6.42±0.18 | 13.10±0.33 |
> > | SGD | 98.68± 0.22 | 93.34±1.17 | 76.42±0.51 | 97.70±0.69 | 85.82±3.83 | 95.93±0.22 | 0.94±0.03 | -4.58±0.23 | 13.75±0.19 |
> > | Ours | 98.99 ± 0.15 | 95.77± 0.18 | 79.11±0.48 | 97.41± 0.46 | 90.23± 1.49 | 96.63±0.19 | 0.86±0.04 | -7.55±0.46 | 12.50±0.12 |
> >
> > ---
> >
> >
> > **6. Clarification on Fig 9**
> >
> > - We first notice that both plots in Fig 9 have the x-axes in _log_ scale. Hence, despite spreading across seemingly identical areas, the fact that our method (green) concentrates on the leftmost part of the plot suggests that it produces more _concentrated_ (i.e. _less sensitive_) results compared to Adam or SGD. For CIFAR10, the training loss of our method concentrates on only 2 bulks: 0.12±0.01 and 0.48±0.01, with the latter occurring only for the largest learning rate. On the contrary, the loss of SGD and Adam spreads across 0.18~2.0 with roughly 5 bulks. Hence, it is clear that our method, compared with baselines, generates results that are _less sensitive_ to hyper-parameters. This is also recognized by Reviewer 1V2i.
> >
> >
> > - In practice, Adam achieved its best result with the learning rate that is typically 1-2 orders smaller than the one for SGD. This has been observed in both traditional deep models and Neural ODEs. The below table reports accuracy (%) and training loss (_inside the parentheses_) for Adam with higher learning rates. On CIFAR10, all larger learning rates have failed to converge properly. On ArtWR, higher learning rates either fail to exceed best-tuned result (~85.8%) or exhibit similar convergence failure (lr$\ge$0.007).
> >
> > 	|  | lr=3e-3 | lr=5e-3 | lr=7e-3 | lr=1e-2 |
> > 	|---|---|---|---|---|
> > 	| Cifar10 | 14.24% (1115.8) | 20.15% (4.53) | 31.03% (1.89) | nan |
> > 	| ArtWR | 82.07% (0.248) | 81.15% (0.002) | 44.89% (90.7) | 5.21% (155.0) |
> >
> >
> > ---
> >
> >
> > **7. Batch size analysis**
> >
> > - We first note that while enlarging batch size increases the memory for all methods, the ratio between our method and first-order baselines does not scale w.r.t. this hyper-parameter. Hence, just as enlarging batch size may accelerate first-order training, it can equally improve our second-order training. In all 9 datasets considered in this work, which sufficiently covers the applications of Neural ODEs, we have not observed any memory shortage when using our method with a (reasonably) larger batch size (note that too large batch size can hinder training [14]).
> >
> >
> > - In terms of the actual performance, the below table provides results on image classification when we enlarge the batch size by the factor of 4 (i.e. 128 -> 512). It is clear that our method retains the same leading position with a comparatively smaller variance. In fact, larger batch size has a side benefit for second-order method as it helps stabilize the preconditioned matrix ($\bar{A}_n$ and $\bar{B}_n$ in Eq 13 & 14) throughout the stochastic training.
> >
> > 	|  | MNIST | SVHN | CIFAR10 |
> > 	|---|---|---|---|
> > 	| Adam | 99.14 ± 0.12 | 94.19 ± 0.18 | 77.57 ± 0.30 |
> > 	| SGD | 98.92 ± 0.08 | 95.67 ± 0.48 | 76.66 ± 0.29 |
> > 	| Ours | **99.18** ± 0.07 | **98.00** ± 0.12 | **80.03** ± 0.10 |
> >
> >
> > - Lastly, we note that increasing the batch size does not always accelerate Neural ODE training. Take SVHN for instance, we observed that increasing the batch size from 128 to 512 concurrently increases the number of function evaluation (NFE), which shall be understood as the "depth" of Neural ODE [1]. This effect is absent in traditional discrete-time models whose depths are fixed throughout training. For Adam, its NFE is increased by 34%, whereas the NFE of our method only increases by 7%. This aligns with our conjecture (L271-273) that preconditioned updates may sometimes yield better-conditioned ODE parameter, and we have provided additional experiments (see our 3. response to Reviewer w3BT) to support the claim. We will include these discussions of batch size analysis in the revision.
> >
> >
> > ---
> >
> >
> > **8. Evaluation comparisons seem cherry picked**
> >
> > - We strongly disagree. All of our experiments have been conducted under standard training pipeline of Neural ODEs. We have stated the experiment details thoroughly in Appendix A.4, including the architecture, dataset, and tuning process, and we also provided reference links to publicly available codebases. This has been recognized by Reviewer 1V2i, HLDZ and w3BT. We urge the reviewer to re-evaluate our experiments in their entirety.
> >
> >
> > ---
> >
> >
> > [10] https://github.com/rtqichen/torchdiffeq/blob/master/examples/odenet_mnist.py#L104
> > [11] Gupta et al 2019, "Shampoo: Preconditioned Stochastic Tensor Optimization"
> > [12] Richter et al 2021, "Solving high-dim PDE with tensor train"
> > [13] Kelly et al 2020, "Learning differential equations that are easy to solve"
> > [14] Keskar et al 2017, "On Large-Batch Training for Deep Learning: Generalization Gap and Sharp Minima"

---

> > > ### Comment · Reviewer_ikHp · 2021-08-11
> > > **Response to response (1/2) and (2/2)**
> > >
> > > Thank you for the detailed response. I have attempted to reply to all points below and clarified my complaints where it seems there has been some confusion, but the overview is:
> > > - the additional experimental results sufficiently address my main complaint regarding the recursive adjoint baseline
> > > - there are still questions about the choice of presented experiments, and the choice of baselines in Fig 5, Tab 3, Fig 10, Tab 2 (for image classification) and a minor complaint about the presentation of Fig 9
> > >
> > > As my main complaint has been address with the additional results in 1.2 and 1.3 I am willing to increase my score to recommend acceptance if the other reviewers do not think my complaints about the experimental procedure are significant.
> > >
> > > ## 1. “Misunderstanding”
> > > I’m not sure what misunderstanding the authors are referring to. The logical flow of the paper is
> > > - L26-36 Neural ODEs are slow to train (wall clock)
> > > - L37-40 Higher order optimisation may alleviate the problem of long training times (wall clock)
> > > - L40-41 Chen et al. note on repeatedly applying adjoint
> > > - L41-44 Claim: “this is, unfortunately, impractical as the recursion will accumulate the aforementioned integration errors and scale the per-iteration runtime linearly.”
> > >
> > > In the original manuscript there is neither a reference nor any evidence presented for this claim The response in **1.2** adequately addresses this complaint by providing evidence, but the reference to [7] is irrelevant as that work also does not provide evidence for its claim.
> > >
> > > The recursive dependency here is between orders of derivatives, not parameters, so it is unclear to me what point the authors are making in 1.1. Perhaps the confusion arises from my use of $dim_R(\theta) = n$ and the authors’ use of $n$ for the order in L98? I will switch to $dim_R(\theta) = N$ and $k$ for the order. The second order algorithm in 1.1 has memory footprint $O(N^2)$ and is $O(1)$ in depth. Is that disputed?
> > >
> > > I have not requested Theorem 1 be compared to the proposed method.
> > >
> > > ### 1.2
> > > My complaint is exactly that the statement L34 is unsubstantiated, and so referencing it as evidence for its own veracity is unconvincing. The references [4-6] are concerned only with first-order errors and do not discuss the accumulation of error over the order of the derivative (higher order derivatives are not mentioned over those three papers.) [7] as pointed out above does not provide evidence for the claim either, and so is not evidence.
> > >
> > > The evidence provided by the authors here is sufficient response to this complaint (and it is not clear to me why they would not simply lead with this evidence.)
> > >
> > > ### 1.3
> > > These further experiments are appreciated. Together with the results in 1.2, this evidence substantiates the claims about i) error accumulation and ii) that this leads to worse performance, as is the evaluation criteria used in this work. Do the authors not agree that their claim in L41-44 and therefore the logical argument of the paper is predicated on these results?
> > >
> > > ### 1.4
> > > $\frac{d^2L}{d\theta^2}$ is matrix valued $\in R^{N \times N}$, I would be curious to understand how this has been reduced to $O(N)$ memory (as provided in the response to HLDZ, the “peak memory” is indeed $O(N^2)$.)
> > >
> > > The comparison is to the complexity of Theorem 1 as this is the point of the paper at which the baseline is rejected:
> > > - L97-99 recursive scales unfavourably
> > > - L99-100 in the next section we present a novel higher-order method…
> > > - Section 3.1 up to Theorem 1 proposed
> > >
> > > That section 3 ends with an efficient approximation to Theorem 1 is not relevant to the way that the baseline is rejected for its memory complexity. The difference between Theorem 1 and the proposed approximation is appreciated, but the paper does not present the rejection of the recursive baseline as being due to the existence of that approximation.
> > >
> > > 1.5
> > > I suggest that the authors focus on the evidence they have gathered in favour of L34, which is compelling. The commitments to include further discussion of these points and the evidence provided in the response, sufficiently addresses my complaint.
> > >
> > > ## 2.
> > > My apologies, I should have been clearer on this point. With $\frac{dx}{dt} = F(t,x,\theta)$ let $\tau = ct$, $d\tau = c dt$ then $\frac{dx}{d\tau} = \frac{1}{c}\frac{dx}{dt} = \frac{1}{c}F(t,x,\theta) =  \frac{1}{c}F(\frac{\tau}{c},x,\theta)$. With the example provided by the authors, $\frac{dx}{dt} = (1-2t)x$ we have $\frac{dx}{d\tau} = \frac{1}{c} (1-\frac{2t}{c})x$ and $x = \text{exp}\Big(\frac{\tau}{c} - \frac{\tau^2}{c}\Big)$ which, of course, evaluates correctly. The correct statement is: rescale the weights in the final layer and rescale the weights on $t$ in the first layer. The choice of initialisation changes these things and the point about including it in the discussion stands. Thank you for agreeing to include the setup for the experiment in the revision.
> > >
> > > Are the authors disputing my point about Fig 5 being equally well explained as the ODE being superfluous in this setting, and therefore not a basis for motivating optimising $t_1$  as an architectural parameter? That is the argument given in L197-208 and is the only discussion of Fig. 5 and 13 in the work, and that is the point I am contesting in my complaint. The ASM baseline and the proposed method are obviously not competitive at SoTA but I’m not asking for that comparison, only a comparison with the equivalent non-ODE model with the same parameters, i.e. the CNN feature extractor as in Appendix A.4 with the 2-layer conv net from the ODE function and the linear layer to produce a ‘static’ model with 5 convolutional layers.
> > >
> > > That is the relevant baseline for the proposed joint optimisation of $t_1$ and $\theta$: it has exactly the same capacity, and is in the same model class, so can it be reached with the proposed method?
> > >
> > > ## 3.
> > > I appreciate and do not dispute the authors’ comments on Eq 9 not being their approximation in Eq 11 and 12, and that this is an “initial attempt”. An honest discussion about how the method scales to higher-order derivatives would not detract from the work and would motivate future work, as described in the second bullet in this response.
> > >
> > > I would encourage the authors to include this discussion in their work as I agree with their assessment that the directions they have highlighted have great practical potential.
> > >
> > > ## 4.
> > > Thank you for the correction.
> > >
> > > ## 5.
> > > Thank you for providing these errors. As presented, the result on MNIST is not significant. Again, I would ask that the authors consider including the 'static' baseline in the comparison on image classification.
> > >
> > > ## 6.
> > > The loss of the proposed method spans nearly 2 orders of magnitude on ArtWR and nearly an order of magnitude on CIFAR10, whilst I agree with the authors that this is _less sensitive_ than Adam and SGD on CIFAR10, Adam seems to be the least sensitive in this regard on ArtWR. I appreciate that the proposed method is the least sensitive in terms of accuracy. I stress again that this information would be more clearly presented in the format used in Figures 5 and 13.
> > >
> > > I'd encourage the authors to include their results for Adam at higher learning rates showing the catastrophic decrease in performance as suggested in the response. Again these results would be much clearer if they were presented differently, please reconsider the choice of representation. (On a more basic level, simulate how Figure 9 appears to a colour blind person e.g. using [SimDaltonism](https://apps.apple.com/us/app/sim-daltonism/id693112260?mt=12) as the green and orange are effectively indistinguishable).
> > >
> > > ## 7.
> > > I appreciate the note that all of the methods scale linearly with batch size, the point is that the proposal starts from a higher initial complexity and so will run out of memory first. Thank you for the discussion and reference to [14].
> > >
> > > The discussion about an enlarged batch size stabilising Eq 13, 14 is valuable and I would encourage the authors to include this, and the additional results, in their revision.
> > >
> > > ## 8.
> > > The point about cherry picking relates to baselines and the choice of experiments presented in the main work vs. the appendix (the comment does not relate to whatever implementation the authors have used, which I have assumed to be a good faith effort). In the original manuscript there was no evidence to suggest the recursive adjoint performed less well than the proposal and there is still no comparison to the ‘static’ baseline I have suggested for image classification. Furthermore, Figure 7 shows training plots for 5 tasks (SpoAD, CharT, SVHN, Gas, and Miniboone) but Table 2 has results on 9 (with the addition of MNIST, CIFAR10, ArtWR, and Circle) then Table 3 and Figure 5 focus on only CIFAR10 whilst Figure 9 uses just CIFAR10 and ArtWR. Table 1 similarly only uses SpoAD, ArtWR and CharT. There isn’t a discussion of why these experimental choices are made, which is what gives the appearance of cherry picking the results. In summary
> > >
> > > | Model | Fig 5 | Fig 7 | Fig 8 | Tab 2 | Fig 9 | Tab 3 | Fig 10 |
> > > | --- | --- | --- | --- | --- | --- | --- | --- |
> > > | MNIST     | ❌ | ❌ | ✅ | ✅ | ❌ | ❌ | ❌ |
> > > | SVHN      | ❌ | ✅ | ✅ | ✅ | ❌ | ❌ | ❌ |
> > > | CIFAR10   | ✅ | ❌ | ✅ | ✅ | ✅ | ✅ | ✅ |
> > > | SpoAD     |  N/A  | ✅ | ✅ | ✅ | ❌ |  N/A  |  N/A |
> > > | ArtWR     |  N/A  | ❌ | ✅ | ✅ | ✅ |  N/A  |  N/A |
> > > | CharT     |  N/A  | ✅ | ✅ | ✅ | ❌ |  N/A  |  N/A |
> > > | Circle    | ❌ | ❌ | ✅ | ✅ | ❌ | ❌ | ❌ |
> > > | Gas       | ❌ | ✅ | ✅ | ✅ | ❌ | ❌ | ❌ |
> > > | Miniboone | ❌ | ✅ | ✅ | ✅ | ❌ | ❌ | ❌ |
> > >
> > > For example, can the authors explain why SVHN was the only image classification task that was included in Figure 7? The results in the supplementary material provide the equivalent figures for MNIST, where Adam trains faster and the methods converge to the same performance, and CIFAR10, where SGD trains more consistently than on SVHN.
> > >
> > > I’m not asking that the authors include every combination of these results but the choice of results should be explained.

---

> > > > ### Author Response · Authors · 2021-08-12
> > > > **Author Response to Reviewer ikHp**
> > > >
> > > > We thank the reviewer for the expeditious reply and greatly appreciate the reviewer's willingness to raise the score. Our clarifications (with the additional experiments on static model) are provided below.
> > > >
> > > > ---
> > > >
> > > >
> > > > **1. Recursive adjoint baseline**
> > > >
> > > > - We are pleased that the reviewer acknowledged our additional experiments on showing the recursive adjoint baseline indeed exhibits _(i)_ error accumulation and _(ii)_ worse performance compared to our proposed method. These results will certainly be included in the revision so that the claims in L34,L40-41,L97-100 are compellingly supported.
> > > >
> > > >
> > > > - We now understood that the current transition from Sec 2 to 3.1 has caused confusion. In the revision, we will make changes to the presentation, particularly L99-101, so that recursive adjoint is neither rejected nor compared against immediately when we enter Sec 3.1. In addition, we plan to include a detailed memory complexity table, as suggested by Reviewer HLDZ and w3BT, that keeps track of the complexity of equations and approximations at different stages.
> > > >
> > > >
> > > > - Regarding the memory footprint, this is indeed a good catch that we missed to clarify in the first response. The specific algorithm in 1.1 has the memory of $O(N)$ since it only computes the _diagonal entries of the Hessian_. This is a known consequence in automatic differentiation: one would have to loop over $N$ in order to compute the full Hessian, as appeared in the official package [15-17]. While adding an additional for-loop indeed gives us $O(N^2)$ memory, its runtime soon becomes computationally unaffordable in our experiments ($N$=16k~75k). We acknowledge that the recursive formula in L98 and the "d2L_dθ2" appeared in 1.1 can cause confusion. These notations will be changed in the revision.
> > > >
> > > >   [15] https://github.com/pytorch/pytorch/blob/master/torch/autograd/functional.py#L696-L702
> > > >   [16] https://github.com/pytorch/pytorch/blob/master/torch/autograd/functional.py#L505-L509
> > > >   [17] https://discuss.pytorch.org/t/compute-the-hessian-matrix-of-a-network/15270
> > > >
> > > > ---
> > > >
> > > >
> > > > **2 & 5. Additional experiments with static model**
> > > >
> > > >
> > > > - We thank the reviewer for the clarification. The discussion on time scaling of $\frac{\mathrm{d}x}{\mathrm{d}t}=F(t,x,\theta)$ makes sense to us, and we will certainly include initialization setup (and also these discussions) in the revision.
> > > >
> > > >
> > > > - The below table provides the results on image datasets trained with the "static" model, where we replace the neural ODE block with a residual convolution block, i.e. "$y = x + \int F_\theta(x)$" --> "$y = x + F_\theta(x)$". We note that this conversion has also been used in prior works ([4]; Sec 4.2 in [5,6]) for similar comparison purposes.
> > > >
> > > > 	|  | MNIST | SVHN | CIFAR10 |
> > > > 	|---|---|---|---|
> > > > 	| Static model + Adam | 98.75 ± 0.21 | 97.28 ± 0.37 | 77.87 ± 0.44 |
> > > > 	| Static model + SGD | 98.66 ± 0.16 | 97.28 ± 0.19 | 75.41 ± 0.27 |
> > > >
> > > >
> > > > - Regarding $t_1$ optimization (Fig 5), the above result on CIFAR10 with Adam (77.87%) seems to align with the reviewer's conjecture: that our proposed method (77.82%; see Table 3) is indeed able to reach the performance of the static baseline, whereas ASM gives only 76.61%, which is 1% lower. We will include the result in Sec 3.3 to better motivate optimizing $t_1$, as suggested by the reviewer.
> > > >
> > > > - However, comparing the above table to Table 2 may seem inconclusive. First, examining the performance difference of the same baselines (Adam & SGD) when trained with either NeuralODE + adjoint method (first 2 rows in Table 2) or static model + backpropagation (above table), the latter achieves comparable results on MNIST and CIFAR10 and can exceed the former by 4-5% on SVHN. On the other hand, while the best-tuned static model (given by Adam) leads our method by 1.5% on SVHN, our method overtook it on CIFAR10 by 1.2%.
> > > >
> > > >   We note, however, that similar results have appeared in prior works (see Fig 5(1) & Table 2 in [6]), where the performance of Neural ODEs can either exceed or being suppressed by static baselines, depending on the hyper-parameters (choice of ODE solver, tolerance, step size, etc). While prior efforts attempted to close the gap by developing better error-handling solvers [5,6], our work provides an alternative from second-order optimization that also improves the performance of Neural ODEs yet with a favorable per-iteration runtime (see 5. response for Reviewer HLDZ).
> > > >
> > > >
> > > > ---
> > > >
> > > >
> > > > **3 & 7**
> > > >
> > > > - Additional discussions on higher-order optimization (**3.**) and stabilization from enlarged batch size (**7.**) will certainly be included in the revision.
> > > >
> > > > ---
> > > >
> > > >
> > > > **6. Fig 9**
> > > >
> > > >
> > > > - We first note that interpreting sensitivity using "_how many orders of magnitude the training losses are spanning over_" is fair only when the losses of two methods are roughly on the same scale. In this case, the training losses of Adam and our method are almost disjoint (except the green point located in the middle). Comparing them under the standard scaling, the difference between the max and min training losses for Adam is 0.69, whereas the one for our method is only 0.06, which is in fact an order of magnitude _smaller_ (hence _less sensitive_) than Adam.
> > > >
> > > >
> > > > - The use of log-scale in the x-axes is to accommodate training losses from all methods without hindering visualization (all green dots simply collapse to one point on standard scale), yet we acknowledge that this may cause confusion for interpretation. In the revision, we will consider alternative presentations of Fig 9, as suggested by the reviewer, for better readability. The results for Adam at higher learning rates will also be included, and we will consider different colors (thanks for the advice!).
> > > >
> > > >
> > > > ---
> > > >
> > > >
> > > > **8. Clarification on experiments in main paper vs Appendix**
> > > >
> > > >
> > > > - As the performance such as training convergence, computational efficiency, and test-time performance are most concerned in practice when deciding optimizers, we see them as the key comparison metrics and prioritize the experiments so that these values are reported for all 9 datasets. We acknowledge that ideally it would be better to include Fig 11, which reports training convergence of all 9 datasets, together with Table 2 and Fig 8 in the main paper. Unfortunately, this figure will occupy 1/2 page and force us to discard other sections that can serve equally important to other audiences.
> > > >
> > > >   Hence, we have little choice but to leave relatively smaller datasets (compared to other datasets of the same application) to Appendix. In our case, they are MNIST and Circle. Next, we made a particular decision to split the rest 7 datasets into Fig 1 & 7, with the hope to provide first-glance experiment support and catch attention on the front page. This leads to the current presentation of Fig 1, 7, 11 and Table 2. To ensure results of all datasets are properly mentioned and directed, we have stated that Fig 1 & 7 report training results (L256-257) and direct readers to Fig 11 for MNIST and Circle (caption of Fig 7).
> > > >
> > > >
> > > > - For $t_1$ optimization, we focus on image datasets by following a prior that supervised learning (compared to generative modeling for CNF) resembles closer optimal-control programming [18]. CIFAR10 and MNIST are selected as two representatives for large/small-scale problems, and due to space constraint, only CIFAR10 is presented in the main paper (MNIST is left in Appendix).
> > > >
> > > >   [18] Benning et al 2019, "Deep learning as optimal control problems"
> > > >
> > > >
> > > > - As for sensitivity analysis, we select CIFAR10 and ArtWR since _(i)_ they are trained on the largest model compared to other datasets of the same application (hence can better distinguish between different optimizers) and _(ii)_ they are relatively faster to trained, which allows us to run extensive experiments to swipe over different hyper-parameters. In practice, generative modeling (CNF) typically requires more careful tuning compared to classification (image/time-series); hence all methods on CNF can be sensitive to hyper-parameter selection. This will be mentioned in the revision.
> > > >
> > > >
> > > > - Table 1 only presents sample size of time-series datasets since the dimension of the other two applications have already been stated (L225-226 for images; L230-232 for CNF). We choose to present the sample size of time-series datasets with a table for its compact expression (compared to text) and space constraint.
> > > >
> > > >
> > > > ---
> > > >
> > > >
> > > > We thank the reviewer again for the meticulous reading. If our replies adequately address your concerns, we would like to kindly ask the reviewer to raise the score so that it better reflects the discussion at the current stage.

---

### Official Review · Reviewer_HLDZ · 2021-07-18

**Rating:** 6
**Confidence:** 4

**Summary:**

This paper proposes a method to derive second-order derivatives for Neural ODEs, which requires O(1) memory in theory. For practical feasibility, the authors propose to use a low-rank approximation of the (dynamic) Hessian matrix, together with Kronecker-based factorization to further reduce the memory cost. The authors validated their methods in image classification, time-series analysis and continuous generative modeling.

**Limitations And Societal Impact:**

Yes

**Main Review:**

Pros:

This method, to my knowledge, is novel in terms of an efficient method to estimate the second-order gradient information for Neural ODE models. The paper is in general well-written with solid theoretical analysis and experimental validations. It would be better if the authors could release code for such a complicated method.

Cons:

I have a few comments and concerns rather than critiques for the authors to clarify.

1. I quickly went through the proof, found the key result rely on two important assumptions (line 526 in appendix), a) $z(t) \otimes g(t)$ are uncorrelated in time $t$, b) $z$ and $g_i$ are pairwise independent. Assumption a) looks very strong even unpractical to me, since in the update it's always $z(t+\delta (t))=z(t)+\Delta(t)$, omit the details of $\Delta(t)$ here. It's clear that $z(t)$ and $z(t+\delta (t))$ are highly-correlated.

2. The author did not specify how to choose a proper rank $R$

3. In terms of memory consumption, If I get it correctly, Full rank second-order adjoint > Low-rank second-order adjoint > first-order adjoint, so the memory cost of the proposed method is in-between. Even though it's O(1) in time $t$, I'm concerned the memory cost has a large constant. \
Specifically, suppose the parameter number is $n$ and activation dimension is $m$, first-order adjoint needs $O(n+m)$ memory, while the proposed method needs $O(n+m+Rm+Rn)$, where $R$ is the rank. This memory cost is required to store the first and second-order gradients. \
In terms of update, the method requires a much higher peak memory of $O(n^2+m^2)$, though it's reusable, because Eq.13 requires the inversion of two matrices of size $n^2$ and $m^2$, this inversion could also be very expensive.  \
Table 7 in appendix shows that the proposed method requires 40% more memory than first-order methods. Note that this is for small-size image ($m=32\times 32$) on CIfar, on large images such as ImageNet ($m=224\times 224$), I expect the memory would be even much larger. \
So the scalability to large scale networks especially with high-resolution images is one of my concerns.

Suggestions

It's shown that numerical errors in gradient for Neural ODE is the cause of bad empirical performances, so I wonder how does the proposed method compare with first-order methods that tackle the numerical issues carefully [1,2,3].

[1] Gholami, Amir, Kurt Keutzer, and George Biros. "Anode: Unconditionally accurate memory-efficient gradients for neural odes." arXiv preprint arXiv:1902.10298 (2019).  \
[2] Zhuang, Juntang, et al. "Adaptive checkpoint adjoint method for gradient estimation in neural ode." International Conference on Machine Learning. PMLR, 2020.  \
[3] Zhuang, Juntang, et al. "MALI: A memory efficient and reverse accurate integrator for Neural ODEs." arXiv preprint arXiv:2102.04668 (2021).

**Time Spent Reviewing:**

4

---

> ### Author Response · Authors · 2021-08-10
> **Author Response to Reviewer HLDZ**
>
>
>
> **1. Uncorrelated assumption of $z(t) \otimes g(t)$**
>
> - We acknowledge that this assumption is indeed strong yet (unfortunately) almost necessary along our derivation to yield tractable Kronecker matrices for efficient second-order operation. Referring back to the development of Kronecker-based methods, similar assumptions also appear in convolution layers (e.g. uncorrelated between spatial-wise derivatives [4]) and recurrent units (e.g. uncorrelated between temporal-wise derivatives [5]). The latter may be thought of as the discretization of Neural ODEs.
>
>
> - We note, however, that it is possible to relax this assumption by considering tractable graphical models (e.g. linear Gaussian [5]) at the cost of 2-3 times more operations per iteration. In terms of the performance difference, perhaps surprisingly, adopting tractable temporal models provides only minor improvement in test-time performance (see Fig 4 in Appendix C of [5]). In some cases, it has been empirically observed that methods that adopt the uncorrelated assumption yields better performance [6]. Hence, in our case, we adopt the assumption not only for tractability but to keep our second-order method computationally comparable to first-order methods in runtime. We will include these discussions in the revision.
>
>
> ---
>
> **2. Selecting the rank $R$**
>
> - For all experiments we apply Gaussian-Newton approximation (L193-195) to the terminal Hessian $Q_xx(t_1)$. Hence, the rank $R$ is implicitly set to 1. The decision was made not only for the practical purpose but to enable connection to Fisher information (i.e. natural gradient) and Gaussian-Newton method, which have been shown effective in second-order optimization of traditional DNNs [7,8].
>
>
> - In practice, increasing $R$ may provide minor improvements with the cost of increasing per-iteration runtime. These additional computations come from constructing the precondition matrices (Eq 13) as well as enlarging the state dimension of the adjoint process (line 6 in Alg 1). As such, we observe that $R=1$ provides equally competitive results while keeping our method distinguishable from first-order methods in both performance and runtime.
>
>
> ---
>
> **3. Clarification on Memory Complexity (fully-connected Neural ODEs)**
>
> - In the below table, we provide the memory complexity of both the adjoint storage and parameter update using different computational formula, starting from the full-rank expression (Theorem 1; Eq 9 & 10), to the low-rank expression (Proposition 2; Eq 11 & 12), and finally our "proposed" second-order update rule (Alg 1; Eq 13 & 14). We adopt the same notation from the reviewer's comments ($n$: parameter number, $m$: activation dimension, $R$: rank) and focus on fully-connected-based Neural ODEs, i.e. $ n \approx m^2$ (see the next bullet point for CNN-based analysis).
>
> 	|  | full rank 2nd-order adjoint(Eq 9 & 10) | low rank 2nd-order adjoint(Eq 11 & 12) | Proposed 2nd-order update(Eq 13 & 14; Alg 1) | First order adjoint (Eq 5) |
> 	|---|---|---|---|---|
> 	| Adjoint Storage | $O(n{+}m{+}(m{+}n)^2)$ | $O(n{+}m{+}Rm{+}Rn)$ | $O(n{+}m{+}Rm{+}2m^2)$ | $O(n{+}m)$ |
> 	| Param. Update | $O(n^2)$ | $O(n^2)$ | $O(2m^2)$ | $O(n)$ |
>
>
> - We stress that there is an important distinction in memory consumption between the low-rank 2nd-order adjoint (Eq 11) and our proposed method (Eq 13), which further introduces layer-wise Kronecker approximation. By doing so, the vectors $\mathbf{p}_i$ appeared in Eq 11, whose dimension is the same as the parameter (hence concerned by the reviewer), were _never_ explicitly constructed during training. Rather, we only carry the matrices $\bar{A}$ and $\bar{B}$. _BOTH_ are with the size of $m^2$, since $ \frac{\partial F}{\partial \mathbf{h}} \mathbf{q} $ shares the same size as the pre-activation vector $\mathbf{h}$ (see Fig 3). As a result, our proposed method requires $O(n{+}m{+}2m^2)$, rather than $O(n{+}m{+}Rm{+}Rn)$, for the adjoint storage, and it dramatically reduces the peak memory during parameter update from $O(n^2)$ to $O(2m^2)$. These matrices $\bar{A}$ and $\bar{B}$ are also comparatively cheaper to inverse than the full 2nd-order matrix (size: $n^2$) appeared in full/low rank 2nd-order adjoint. We will include these complexity analysis in the revision.
>
> ---
>
> **4. Memory Scalability for convolution-based Neural ODEs in ImageNet**
>
> - As for convolution-based Neural ODEs that propagate feature maps for image applications, our proposed method adopts Kronecker-based factorization for convolution layers (Theorem 2 in [4]; also Theorem 4.3 in [9]). The memory complexity of the resulting approximation scales w.r.t. the number of feature maps (i.e. number of channels; typically 64 ~ 512 for ImageNet) rather than the size of feature map (as conjectured by the reviewer).
>
>
> - Hence, one should expect a similar memory-increasing ratio despite lifting to large-scale applications. To validate our claim, we run additional experiments and record the highest memory footprint when training the Neural ODE used for ImageNet application [10,11]. While vanilla Adam with 1st order adjoint requires 19.8GB, our method consumes 23.4GB. This indicates a memory-increasing ratio of only 18%. In short, we validate that our method retains a similar (_in fact slightly lower_) memory-increasing ratio for large-scale image applications such as ImageNet.
>
>
> - We acknowledge that Eq 13 & 14 are mainly based for fully-connected layers. While they are indeed the primary building blocks for most applications (see Table 4), the notation may cause confusion when interpreting convolution layers. We will include the discussion for convolution layers in the revision.
>
> ---
>
> **5. Comparison to prior first-order methods that handle numerical errors**
>
> - In the tables below, we report the performance between vanilla Adam, Adam+MALI, and our method on various metrics, including the test-time accuracy (%), per-iteration runtime (sec/itr), and its relative ratio w.r.t. vanilla Adam (_inside the parentheses_). We consider MALI [3] as the representative of the error-handling method since it has been shown to improve ACA [2] on training Neural ODEs. On the other hand, Anode [1] can be viewed as a fixed-step ODE solver (see Sec 6 in [2]); hence not directly applicable to our adaptive solver setup. While Adam+MALII did improve the accuracy of vanilla Adam at the cost of extra per-iteration runtime (roughly 3 times longer), our method achieves highest accuracy among 3 methods and retains a comparable runtime w.r.t. vanilla Adam.
>
> 	|  | Adam | Adam+MALI  | Ours |
> 	|---|---|---|---|
> 	| SVHN | 91.92 % | 91.98 % | **95.77 %** |
> 	| CIFAR10 | 77.41 % | 77.70 % | **79.11 %** |
>
>
> 	|  | Adam | Adam+MALI  | Ours |
> 	|---|---|---|---|
> 	| SVHN | 0.78 sec/itr | 2.31 sec/itr (_2.96_) | **0.68 sec/itr** (**_0.87_**) |
> 	| CIFAR10 | **0.17 sec/itr** | 0.55 sec/itr (_3.24_) | 0.20 sec/itr (**_1.16_**) |
>
> - We note that the topic of error handling stands in parallel to the development of efficient second-order optimization considered in our work. As both methodologies attempt to make influential advances toward practical training of Neural ODEs,
> it will be interesting in future works to investigate error handling in higher-order training scheme, and we thank the reviewer for bringing up the topic.
>
> ---
>
> [4] Grosse & Martens 2016, "K-kfac for Conv"
> [5] Martens et al 2018, "K-kfac for RNN"
> [6] Laurent et al 2018, "Fisher approx beyond K-fac"
> [7] Zhang et al 2019, "Fast Convergence of Natural Gradient Descent for DNNs"
> [8] Botev et al 2017, "Practical Gauss-Newton Optimisation for Deep Learning"
> [9] Gao et al, 2020, "Trace-restricted Kronecker-Factored ..."
> [10] Zhuang et al 2021, "Mali ..."
> [11] https://github.com/juntang-zhuang/TorchDiffEqPack/blob/master/image_classification/models/resnet.py

---

> > ### Comment · Reviewer_HLDZ · 2021-08-17
> > **Updated response**
> >
> > Thanks for your response, most of my concerns have been addressed.
> >
> > After reading authors's response and reading the paper again, I noticed that Adam always perform worse than SGD, and for experiments that deals with numerical issues in gradient estimation, the authors only compared to Adam + MALI but not SGD + MALI. Also it's better to report the rtol, atol, lr and weight decay for better comparison.
> >
> > I wonder what's the exact operation in line 8 and 10 in the algorithm? Did you add anything like epsilon to prevent numerical issues? Did you brutally (but accurately) do the matrix inversion or by some approximation? For weight decay, did you use decoupled decay? From reported experiments, in most cases SGD outperforms Adam, which implies that adaptive optimizers might not be suitable for chosen tasks. This also implies that if you use any operations like adding eps, use a larger eps would make the algorithm behave closer to SGD and you would get better performance for the chosen tasks. So it would be better if the authors could provide an algorithm with exact operation.

---

> > > ### Author Response · Authors · 2021-08-20
> > > **Author Response to Reviewer HLDZ**
> > >
> > >
> > >
> > > We thank the reviewer for the response. Clarifications are provided below.
> > >
> > > ---
> > >
> > >
> > > **6. Operation of $\epsilon$-regularized Kronecker precondition in line 10 of Alg 1**
> > >
> > > - As conjectured by the reviewer, the precondition matrix used for practical training is indeed $\mathcal{L}_{\theta_n\theta_n} + \epsilon I$, where $\theta_n$ is the parameter of layer $n$ (see Fig 3 & Eq 13) and $\epsilon$ is the Tikhonov regularization widely used for stabilizing second-order training [5,7-9]. To efficiently compute this $\epsilon$-regularized Kronecker precondition matrix without additional factorization/approximation (e.g. Sec 6.3 in [12]), we instead perform eigen-decompositions, i.e. $\bar{A}_n = U_A \Sigma_A U_A^{\mathsf{T}}$ and $\bar{B}_n = U_B \Sigma_B U_B^{\mathsf{T}}$, so that we can utilize the property of Kronecker product [13] to obtain $(\bar{A}_n \otimes \bar{B}_n + \epsilon I)^{-1} {=} (U_A \otimes U_B)(\Sigma_A \otimes \Sigma_B + \epsilon)^{-1}(U_A \otimes U_B)^{\mathsf{T}}$.
> > >
> > >
> > > - In practice, we further adopt eigen-based amortization (L188-189) [14], which substitutes the original diagonal matrix $S := \Sigma_A \otimes \Sigma_B$ with $S^* := ((U_A \otimes U_B)^{\mathsf{T}}\mathcal{L}_{\theta_n})^2$. It has been _proven_ that using $S^*$ yields a better approximation (in the sense of Frobenius norm) compared to $S$ (see Fig 2 & Appendix A in [14]), and that averaging $S^*$ over training provides best empirical performance for training discrete DNNs (see Sec 4 in [14]) .
> > >
> > >
> > > - To summarize, the exact operation for $\epsilon$-regularized Kronecker-preconditioned update goes as follows. Note that we omit the subscript $n$ for brevity. Also, we denote $\tilde{\mathcal{L}}_\theta$ as the un-vectorized gradient (i.e. of the same size as layer's weight matrix). Hence, $\mathcal{L}_\theta$ = vec($\tilde{\mathcal{L}}_\theta$) or equivalently $\tilde{\mathcal{L}}_\theta$ = vec$^{-1}$($\mathcal{L}_\theta$).
> > >
> > >     > 10.1 $\quad$ $U_A$, $\Sigma_A$ = EigenDecomposition($\bar{A}$)
> > >     > 10.2 $\quad$ $U_B$, $\Sigma_B$ = EigenDecomposition($\bar{B}$)
> > >     > 10.3 $\quad$ $X$ := vec$^{-1}$($(U_A \otimes U_B)^{\mathsf{T}} \mathcal{L}_\theta$) = $U_B^{\mathsf{T}} \tilde{\mathcal{L}}_\theta U_A$
> > >     > 10.4 $\quad$ $S^*$ := $\alpha S^* + (1-\alpha)X^2$
> > >     > 10.5 $\quad$ $X$ := $X$ / ($S^* + \epsilon$)
> > >     > 10.6 $\quad$ update := $(U_A \otimes U_B)$ vec($X$) = vec($U_B X U_A^{\mathsf{T}}$)
> > >     > 10.7 $\quad$ $\theta \leftarrow \theta - \eta$ $\cdot$ update
> > >
> > >     The eigen-decompositions in Lines 10.1 & 10.2 are computed exactly (by calling `torch.symeig`) without approximation. Lines 10.3 & 10.4 compute the running average of $S^*$, and Line 10.5 performs element-wise division. Line 10.6 computes the $\epsilon$-regularized Kronecker-preconditioned update ("$X$" is intermediate quantity), and finally Line 10.7 applies the update with learning rate $\eta$.
> > >
> > >
> > > - The above computation contains 2 _new_ hyper-parameters, namely $\epsilon$ and $\alpha$. In practice, we fix $\alpha$=0.75 for all experiments. While fine-tuning $\alpha$ may further improve the performance of our method, we find this value provides sufficiently good results. As for $\epsilon$, we tested 3 different values {0.1, 0.05, 0.03} on each dataset and report the best result. The best-tuned $\epsilon$ for each dataset is shown in the below table.
> > >
> > >     |  | mnist | svhn | cifar10 | spoAD | ArtWR | CharT | Circle | Gas | Minib |
> > >     |---|---|---|---|---|---|---|---|---|---|
> > >     | epsilon | 0.05 | 0.05 | 0.03 | 0.05 | 0.1 | 0.05 | 0.03 | 0.05 | 0.1 |
> > >
> > >     As suggested by the reviewer, in the degenerate case when ($\bar{A}$, $\bar{B}$) := (0, 0), $\epsilon$=1 corresponds to vanilla SGD update. From this view, our $\epsilon$-regularized Kronecker updates behave _far_ from SGD, as our $\epsilon$ is typically 10-30 times smaller (note that the learning rates of two methods are roughly on the same scale; see Fig 9). We also note that in practice $\bar{A}$ and $\bar{B}$ are highly non-trivial matrices. Lastly, for all experiments we use traditional weight decay rather than decoupled decay.
> > >
> > > - Alg 1 was originally presented without $\epsilon$ for its relatively compact expression (compared to $\epsilon$-regularized updates) and familiarity for readers with background on traditional Kronecker operation. In the revision, we will provide these detailed discussions by updating Alg 1 & Appendix, and we thank the reviewer for raising these topics.
> > >
> > >
> > > ---
> > >
> > > **7. SGD + MALI and hyper-parameter setup**
> > >
> > > - In the tables below, we report the performance between SGD, SGD+MALI, and our method with the same format as in the response **5** (upper: test-time accuracy, bottom: per-iteration runtime with its relative ratio w.r.t. SGD shown _inside the parentheses_). We observe similar performance difference when using SGD with MALI at the cost of additional computation, and our method retains highest accuracy among the 3 methods with a comparable runtime.
> > >
> > >     |  | SGD | SGD+MALI  | Ours |
> > >     |---|---|---|---|
> > >     | SVHN | 93.34 % | 94.33 % | **95.77 %** |
> > >     | CIFAR10 | 76.42 % | 76.41 % | **79.11 %** |
> > >
> > >     |  | SGD | SGD+MALI  | Ours |
> > >     |---|---|---|---|
> > >     | SVHN | 0.81 sec/itr | 1.28 sec/itr (_1.58_) | **0.68 sec/itr** (**_0.84_**) |
> > >     | CIFAR10 | **0.17 sec/itr** | 0.23 sec/itr (_1.35_) | 0.20 sec/itr (**_1.16_**) |
> > >
> > >
> > > - In the tables below, we report the hyper-parameter for these experiments (upper: SVHN, bottom: CIFAR10). epsilon is the hyper-parameter specifically for our second-order method (see response **6** above). As noted (L243-244), we fix the rtol & atol w.r.t. each dataset for a better comparison among optimizers. In practice, Adam achieved its best result with the learning rate that is typically 1-2 orders smaller than the one for SGD or our method. This has been observed in both traditional deep models and Neural ODEs. For instance, on CIFAR10, Adam with lr > 1e-3 can fail to converge properly (see response 6 for Reviewer ikHp (Part 2/2)).
> > >
> > >     |  | rtol | atol | lr | weight decay | epsilon |
> > >     |---|---|---|---|---|---|
> > >     | Adam & Adam+MALI | 1e-3 | 1e-3 | 5e-4 | 1e-3 | -  |
> > >     | SGD & SGD+MALI | 1e-3 | 1e-3 | 2e-2 | 0 | - |
> > >     | Ours | 1e-3 | 1e-3 | 1e-2 | 0 | 0.05 |
> > >
> > >     |  | rtol | atol | lr | weight decay | epsilon |
> > >     |---|---|---|---|---|---|
> > >     | Adam & Adam+MALI | 1e-3 | 1e-3 | 1e-3 | 1e-3 | -  |
> > >     | SGD & SGD+MALI | 1e-3 | 1e-3 | 1e-2 | 0 | - |
> > >     | Ours | 1e-3 | 1e-3 | 1e-2 | 0 | 0.03 |
> > >
> > >
> > > ---
> > >
> > >
> > > **8. SGD vs Adam**
> > >
> > > - We first note that for CNF, Adam performs better than SGD on all 3 datasets (since lower negative log-likelihood (NLL) is better; see Table 2). For image classification, it has been previously observed that SGD achieves better test-time performance compared to Adam [15-17], despite a more recent study suggested the performance of optimizers can really differ from tasks to tasks [18]. In our experiments, we found SGD is better suited (compared to Adam) on classification problems but not CNF.
> > >
> > >
> > > ---
> > >
> > >
> > > [12] Optimizing Neural Networks with Kronecker-factored Approximate Curvature
> > > [13] On the Kronecker Product (see KRON 15 on page 9)
> > > [14] Fast Approximate Natural Gradient Descent in a Kronecker-factored Eigenbasis
> > > [15] Adaptive Gradient Methods with Dynamic Bound of Learning Rate
> > > [16] Improving Generalization Performance by Switching from Adam to SGD
> > > [17] Towards Theoretically Understanding Why SGD Generalizes Better Than ADAM in Deep Learning
> > > [18] Descending through a Crowded Valley - Benchmarking Deep Learning Optimizers

---

> > > > ### Comment · Reviewer_HLDZ · 2021-08-21
> > > > **Thanks for your response**
> > > >
> > > > Thanks for your detailed response. I keep my score for acceptance. I would suggest the authors to include the detailed algorithm in the paper; for adaptive optimizers, $\epsilon$ is typically very important, and I think second order methods would also be sensitive to dampening factors $\epsilon$ and $\alpha$, it would be very helpful to include these discussion.

---

### Official Review · Reviewer_1V2i · 2021-07-18

**Rating:** 7
**Confidence:** 3

**Summary:**

This method proposes a second-order optimization network for neural ODEs. Inspired by OC, this paper adopts differential programming to derive backward ODE for higher-order derivatives at constant memory cost. This paper also explores using low-rank representation for the second-order optimization to make more efficient updates. The paper shows their method achieves faster convergence than first-order methods across three different tasks. This framework also enables the optimization of the architecture with the example of integration time of neural ODE.

**Limitations And Societal Impact:**

The limitations were adequately discussed.

**Main Review:**

Originality
- This work proposes a novel second-order optimization network specifically for neural ODEs. It adopts ideas from OCP to neural ODE to achieve this goal. It is clear how this work differs from previous work, and the related works are appropriately cited. I like how it introduces the OCP perspective to neural ODE and it can treat integration time as an optimization variable to optimize the architecture directly.

Quality
- This work is technically sound. It makes sense that using a second-order optimization approach can make neural ODE converge faster and this claim is well-supported by three different tasks, which demonstrate faster convergence and higher accuracy using their method than using the first-order method.

- I would like to see the reasoning behind Figure 9, which claims that this method is less sensitive to hyper-parameter changes. It is not very obvious based on the theoretical ground and is worth exploring.

Clarity
- The overall structure is well-organized.

- Different fonts of x are present in lines 19 and 21 before it is clarified in preliminary.

- On line 118: what are “These functions are known as...” Which are the functions here?

Significance
- Other researchers may use their method to run neural ODE, but I doubt the significance of developing even higher-order optimization based on this work. Also, I am not sure if it is worth implementing the method given its complexity and limited improvement on computational time. Please discuss these in Section 5.

Additional Questions
-What is the relationship between step sizes and integration time? Does increasing the time step size have the same effect as decreasing the integration time for image and CNF datasets?

-Why do you choose a Runge-Kutta 4 adaptive solver instead of an implicit solver such as the implicit Adams method? Do the benefits depend on the specific solver you choose?

-For Eq. 14, why do you call this specific formula a preconditioned update law? Why do you choose this to update theta?

-Is there any mathematical formulation that supports how much the second-order optimization improves the convergence rate of Neural ODE?

-Why using second-order optimization can also improve the accuracy of the experiments as demonstrated in figure 9?


**Time Spent Reviewing:**

3 hours

---

> ### Author Response · Authors · 2021-08-10
> **Author Response to Reviewer 1V2i**
>
>
> **1. Insensitivity, convergence, and generalization of second-order method (Fig 9)**
>
> - We recall (L193-195) that for all experiments we apply Gaussian-Newton approximation to the terminal Hessian $Q_{\bar{x}\bar{x}}(t_1)$. This specific choice is partially based on empirical performance and computational purpose, yet it turns out that the resulting precondition matrices (Eq 12 & 13) can be interpreted as Fisher information matrix (FIM) (see Appendix C in [1] for complete derivation). In other words, under this specific setup, Eq 12 & 13 can be equivalently viewed as the FIM of Neural ODEs. Connecting our optimal-control-inspired matrices to FIM, which originates from information geometry, turns out to be useful for revealing theoretical insights.
>
>
> - First, using FIM to precondition the parameter update corresponds to Natural Gradient Descent (NGD), which is well-known for taking the _steepest_ descent direction in the space of model distributions [2,3]. This implies our updates are invariant to smooth transformations (e.g. whitening) of models. This invariance, together with the fact that the optimization dynamics move in the space of distributions, hints that the resulting method may enjoy a certain level of insensitivity. Indeed, it has been observed that second-order method equipped with FIM converges to equally good accuracies, even though its learning rate varies across 1-2 orders (see Fig 10 in [4] & Fig 4 in [5]). These observations coincide with our results (Fig 9) for Neural ODEs.
>
>
> - Next, it has been _proven_ [6] that NGD improves the convergence rate over first-order methods (gradient descent) by $O(\lambda_\min(G))$, where $G$ depends on the dataset and $\lambda_\min(G)$ is its smallest eigenvalue. This analysis is _architecture-agnostic_, providing certain regularity conditions (e.g. stable) on the Jacobian, and it has been _proven_ that it shares the same generalization bound as first-order methods (Theorem 6 in [6]). As for the additional accuracy improvements in Fig 9, we conjecture these are attributed to the invariance (see our Corollary 4 in L613) and potential implicit regularization (see our third response to Reviewer w3BT) from our method. We will include these discussions in the revision, and we thank the reviewer for bringing up the topic.
>
> ---
>
> **2. Discussion on higher-order ($k\ge3$) optimization in Appendix**
>
> - Development of higher-order ($\ge$3) optimization, based on Theorem 1, certainly has few computational obstacles, just like what we have identified (L155-159) and later resolved (Sec 3.2) in the case of $k=2$. In terms of memory, while the number of backward ODEs in Eq 9 can grow exponentially w.r.t. optimization order $k$, prior work [7] has developed an efficient truncated method that reduces the number to $O(k^2)$ or $O(k\log k)$. In terms of runtime, analogous to the Kronecker approximation that we use to factorize 2nd-order matrices, [8] provided an extension to generic higher-order tensor programming. Hence, it may still be plausible to avoid impractical training.
>
>
> - Whether these higher-order schemes can provide additional benefit, compared to our already complicated 2nd-order method, remains unexplored. However, higher-order derivatives have been shown useful in regularizing Neural ODE dynamics [7], and tensor-based factorization has recently achieved promising results on optimizing high-dimensional nonlinear PDEs [9]. This is indeed a largely unexplored area, and our framework may benefit or inspire applications on other aspects, if not directly optimization. These discussions will be added to Sec 5 in the revision.
>
>
> - Regarding the implementation of our second-order method, we will release code upon publication. We also emphasize that our method improves the computational time over baselines by a _large_ margin (see SVHN, CIFAR10, ArtWR, CharT in Fig 12),
> and are able to achieve better test-time performance that are otherwise unachievable using baselines.
>
> ---
>
> **3. Relationship between step size and integration time**
>
> - While increasing the step size seems to have similar effect as decreasing integration time (since both reduce the computation runtime), they result in different accumulated integration errors. Specifically, increasing the step size will typically increase the error incurred during adjoint process. On the other hand, decreasing the integration time has the opposite effect. For this reason, it is often preferable to consider _adaptive solvers_, which self-tune the step size depending on the stiffness of the current dynamics and the pre-defined tolerance. The integration time typically remains as a hyper-parameter, and our method provides an intellectual way to self-tune/adapt this parameter throughout training.
>
> ---
>
> **4. Performance using implicit Adams ODE solver**
>
> - We select RK4 adaptive solver since it is the default solver that appeared in the official Neural ODE implementation [10] and has been widely adopted in the applications [10-12] considered in this work. In the below table we provide the test-time performance when using the implicit Adams solver.
>
>     |  | mnist | svhn | cifar10 | spoAD | ArtWR | CharT | Circle | Gas | Minib |
>     |---|---|---|---|---|---|---|---|---|---|
>     | Adam | 98.86 | 91.76 | 77.22          | 95.33 | 86.28 | 88.83 | 0.90 | -6.51 | 13.29 |
>     | SGD | 98.71 | 94.19 | 76.48  | **97.80** | 87.05 | 95.38 | 0.93 | -4.69 | 13.77 |
>     | Ours | **98.95** | **95.76** | **79.00** | 97.45 | **89.50** | **97.17** | **0.86** | **-7.41** | **12.37** |
>
>
> - The result shows that our method retains the same leading position as appeared in Table 2, and the relative performance between optimizers also remains unchanged. In practice, the train/test curves also resemble Fig 11 & 12. Hence, we believe the superior performance of our method depends less on the choice of the ODE solver, but rather the use of efficient second-order update during training. We will include these results in the revision.
>
> ---
>
> **5. Preconditioned update in Eq 14**
>
> - The term "preconditioned update" is typically used in the optimization literature [13,14] to indicate second-order update, which "precondition" the original update $L_\theta$ with some matrix $L_{\theta\theta}^{-1}$. In our case, this preconditioned matrix is factorized by two smaller matrices $\bar{A}_n$ and $\bar{B}_n$ in Eq 13. This allows us to adopt the formula from Kronecker product (Eq 28), which yields the update law in Eq 14.
>
>
>
> ---
>
> **6. Notation**
>
> - "These functions" (L118) correspond to the terminal cost $\Phi$ and intermediate cost $\ell$ functions, respectively. We will unify the font in L19,21 in the revision.
>
> ---
>
> [1] Botev et al 2017, "Practical Gauss-Newton Optimisation for Deep Learning"
> [2] Amari and Nagaoka 2007, "Methods of information geometry"
> [3] Martens 2020 "New Insights and Perspectives on the Natural Gradient"
> [4] Ma et al 2019, https://arxiv.org/pdf/1903.06237.pdf
> [5] George et al 2018, "Fast NGD in Kfac Eigenbasis"
> [6] Zhang et al 2019, "Fast Convergence of NGD for DNNs"
> [7] Kelly et al 2020, "Learning differential equations that are easy to solve"
> [8] Gupta et al 2019, "Shampoo: Preconditioned Stochastic Tensor Optimization"
> [9] Richter et al 2021, "Solving high-dim PDE with tensor train"
> [10] https://github.com/rtqichen/torchdiffeq/
> [11] Grathwohl et al., 2018; FFJORD
> [12] Kidger et al., 2020b; NCDE
> [13] Li et al, 2015 "Preconditioned SGLD for DNN"
> [14] Siahkoohi et al, 2021, "Preconditioned training of NF..."

---

### Author Response · Authors · 2021-08-10
**Author response to all reviewers**


We thank the reviewers for their valuable comments. We are excited that the reviewers identified the importance of the problem (_Reviewer 1V2i, w3BT_), appreciated the novelty of the contributions (_Reviewer 1V2i, HLDZ, ikHp_), acknowledged our comprehensive experimental validations (_Reviewer 1V2i, HLDZ, w3BT_), and found the paper well-written (_Reviewer 1V2i, HLDZ, w3BT_). We believe our method takes a significant step toward principled algorithmic design inspired from optimal control theory for the emerging deep continuous-time models.

Unfortunately, Reviewer ikHp misunderstood the recursive adjoint baseline and few experiment setups, which led to ill-founded doubts of our contributions and credibility of our experiments. We have provided strong additional results (1.2, 1.3 response to Reviewer ikHp) showing our method outperforms the concerning baseline by a _large_ margin in both test-time performance and runtime.

All raised concerns are addressed in individual response below. We try our best to resolve the misunderstandings and sincerely hope Reviewer ikHp will reconsider the rating and re-evaluate at an entirety.

---

### Decision · Program_Chairs · 2021-09-27

**Decision:**

Accept (Spotlight)

**Comment:**

This paper proposes a second-order optimization algorithm for neural ODEs. Inspired by optimal control theory, the paper derives backward ODEs for higher-order derivatives at constant memory cost. The authors shows their method achieves faster convergence than first-order methods across three different tasks.

The contribution is novel and relevant to NeurIPS, and is appreciated by the reviewers. The main criticism that surfaced during review is the absence of certain baselines, raised by reviewer ikHp: After back and forth discussion between the authors and reviewer this concern was addressed sufficiently for the reviewer to raise their review score. The authors also replied extensively to comments by the other reviewers, and no further serious concerns were identified. The reviewers formed a consensus recommendation to accept the paper.